# SurvHTE-Bench: A Benchmark for Heterogeneous Treatment Effect Estimation in Survival Analysis

**Shahriar Noroozizadeh**[*]
Machine Learning Department & Heinz College
Carnegie Mellon University
snoroozi@cs.cmu.edu

**Xiaobin Shen**[*]
Heinz College
Carnegie Mellon University
xiaobins@andrew.cmu.edu

**Jeremy C. Weiss**
National Library of Medicine
National Institutes of Health
jeremy.weiss@nih.gov

**George H. Chen**
Heinz College
Carnegie Mellon University
georgechen@cmu.edu

## Abstract

Estimating heterogeneous treatment effects (HTEs) from right-censored survival data is critical in high-stakes applications such as precision medicine and individualized policy-making. Yet, the survival analysis setting poses unique challenges for HTE estimation due to censoring, unobserved counterfactuals, and complex identification assumptions. Despite recent advances, from Causal Survival Forests to survival meta-learners and outcome imputation approaches, evaluation practices remain fragmented and inconsistent. We introduce SurvHTE-Bench, the first comprehensive benchmark for HTE estimation with censored outcomes. The benchmark spans (i) a modular suite of synthetic datasets with known ground truth, systematically varying causal assumptions and survival dynamics, (ii) semi-synthetic datasets that pair real-world covariates with simulated treatments and outcomes, and (iii) real-world datasets from a twin study (with known ground truth) and from an HIV clinical trial. Across synthetic, semi-synthetic, and real-world settings, we provide the first rigorous comparison of survival HTE methods under diverse conditions and realistic assumption violations. SurvHTE-Bench establishes a foundation for fair, reproducible, and extensible evaluation of causal survival methods. The data and code of our benchmark are available at: https://github.com/Shahriarnz14/SurvHTE-Bench.

## 1 Introduction

In many causal inference applications where we aim to quantify how well a treatment works, estimating *heterogeneous treatment effects* (HTEs) could be more useful than only estimating population-level *average treatment effects* (ATEs), building on the intuition that the same treatment can vary in effectiveness when given to different individuals. In survival analysis with right-censored outcomes (common in clinical trials and electronic health records), estimating HTEs can be especially challenging. In addition to the standard difficulties of causal inference (unobserved counterfactuals, confounding), the analyst must account for censoring, where the event of interest is only observed for a subset of subjects. These features complicate identification and estimation, yet they are central in high-stakes applications such as precision medicine and individualized policy-making (Zhu & Gallego, 2020; Chapfuwa et al., 2021; Curth et al., 2021a).

Recent years have seen a growing set of causal survival methods (Chapfuwa et al., 2021; Curth et al., 2021a; Cui et al., 2023; Bo et al., 2024; Noroozizadeh et al., 2025; Xu et al., 2024; Meir et al., 2025). Despite methodological advancement, no standardized benchmark exists, limiting reproducibility and fair comparisons. Most studies rely on bespoke simulations or limited real datasets with unknown ground truth, with differing levels of censoring, survival distributions, and causal

---

[*]These authors contributed equally to this work and are listed alphabetically.

assumptions. As a result, comparisons are not standardized, the robustness of different proposed methods is unclear, and progress is difficult to measure.

While there is a growing benchmarking literature for treatment-effect heterogeneity in fully observed outcomes (e.g., Crabbé et al. (2022); Shimoni et al. (2018); Kapkiç et al. (2024)) and recent benchmarks for survival ATE estimation (e.g., Voinot et al. (2025)), to our knowledge, there is not yet any benchmark for survival HTE estimation under right-censoring. This missing piece motivates our focus on heterogeneous effects in censored time-to-event data.

We introduce SURVHTE-BENCH, the first comprehensive benchmark for HTE estimation in right-censored survival data. Our contributions are as follows:

- **Method unification:** We categorize existing survival HTE methods (and natural extensions of such existing methods that technically have not previously been published) into three broad families: outcome imputation methods, direct-survival causal methods, and survival meta-learners. We provide a modular implementation of 53 methods among these families. This is the first systematic framework that unifies survival HTE methods, facilitating reproducibility and extensibility.
- **Synthetic benchmark design:** We present a curated suite of 40 synthetic datasets spanning eight causal configurations (with different combinations of randomization, unobserved confounding, overlap violation, informative censoring) crossed with five survival scenarios (with different survival and censoring distributions), yielding controlled settings with known ground-truth HTEs under realistic assumption violations.
- **Semi-synthetic and real data:** We also include 10 semi-synthetic datasets adapted from existing literature (real covariates with simulated treatments and outcomes) that aim to be more realistic compared to purely synthetic datasets while still having ground truth on HTEs. We further include 2 widely studied real datasets: the Twins dataset that has known ground truth (Almond et al., 2005) (i.e., per twin, one has the treatment and the other does not, so that we observe both counterfactual outcomes), and the HIV clinical trial dataset without known ground truth (Hammer et al., 1996).
- **Comprehensive evaluation:** We compare representative estimators across all settings. Our results show that no single method dominates: performance depends on causal assumptions, censoring, and survival dynamics. Notably, S- and matching-learners among survival meta-learners demonstrate robustness under severe violations and high censoring.

While prior work has explored subsets of these design choices (e.g., Cui et al. (2023); Meir et al. (2025)), SURVHTE-BENCH is the first to systematically evaluate survival HTE methods under assumption violations, diverse survival models, and across synthetic, semi-synthetic, and real data. We focus on binary treatments and static covariates with right-censored outcomes, as even this basic setting lacks a standardized benchmark. More complex extensions (time-varying treatments, longitudinal covariates, and instrumental variables) are beyond our present scope.

## 2 BACKGROUND AND RELATED WORK

We briefly review the problem setup, identification assumptions, existing evaluation practices, and the three families of survival HTE estimators.

**Problem setup.** For each unit (data point) $i$, we observe covariates $X_i \in \mathcal{X}$, a binary treatment $W_i \in \{0, 1\}$, and an observed, possibly censored event time $\widetilde{T}_i = \min(T_i, C_i)$ with event indicator $\delta_i = \mathbb{1}\{T_i \leq C_i\}$, where $\delta_i$ is 1 if the event of interest happened (in which case $\widetilde{T}_i$ is the event time) or 0 if the outcome is censored (in which case $\widetilde{T}_i$ is the censoring time). Using the standard potential outcomes framework, $T_i(w)$ denotes the potential event time under treatment $w \in \{0, 1\}$ with $T_i = T_i(W_i)$. We assume that the tuple $(X_i, W_i, T_i(0), T_i(1), C_i)$ is i.i.d. across different $i$.

We aim to estimate the *conditional average treatment effect* (CATE) with respect to a transformation of the event time $y(\cdot)$:

$$\tau(x) := \mathbb{E}\big[y\big(T_i(1)\big) - y\big(T_i(0)\big)|X_i = x\big], \tag{1}$$

where $y(\cdot)$ encodes the survival estimand of interest, and the expectation is taken over the randomness of the two potential outcomes. For example, if we want the survival estimand to be the restricted mean survival time (RMST) up to a user-specified time horizon $h > 0$, then we would set $y(t) := \min\{t, h\}$. Other choices for estimands are also possible (e.g., median survival time, survival probability at a fixed time). In this paper, we focus on RMST, which is interpretable, robust under censoring, and widely adopted (Shen et al., 2018; Curth et al., 2021a; Cui et al., 2023), while

noting that our benchmark design allows extensions to other estimands, and we include example results for survival probabilities in Appendix G.4.

**Identification assumptions.** Identification of $\tau(x)$ relies on the following assumptions (Cui et al., 2023) (and in our benchmark, we examine settings where these assumptions do not hold):

- (A1) *Consistency*: $T_i = T_i(W_i)$ almost surely.
- (A2) *Ignorability*: $\{T_i(0), T_i(1)\} \perp W_i \mid X_i$.
- (A3) *Positivity*: $\eta_e \leq \mathbb{P}(W_i = 1 | X_i = x) \leq 1 - \eta_e$ for some $\eta_e > 0$.
- (A4) *Ignorable censoring*: $T_i \perp\!\!\!\perp C_i \mid X_i, W_i$.
- (A5) *Censoring positivity*: For horizon $h$, $\mathbb{P}(C_i < h | X_i, W_i) \leq 1 - \eta_C$ for some $0 < \eta_C \leq 1$.

Violations are common: unmeasured prognostic factors break ignorability, treatment guidelines undermine positivity, and drop-out linked to prognosis induces informative censoring. A central goal of SURVHTE-BENCH is to measure how estimators behave under such violations.

**Existing evaluation practice.** Because only one potential outcome is observed per unit, validation typically relies on author-specific simulations. Prior studies vary assumptions in narrow ways: e.g., censoring up to 30% (Bo et al., 2024) or heavy censoring but assuming ignorability (Meir et al., 2025). Consequently, results are not comparable across papers, and estimator robustness under simultaneous assumption violations remains unclear. To date, no public benchmark exists with known individual-level ground truth with varying levels of assumption violations and survival distributions.

**Overview of existing survival HTE estimators.** We group existing methods into three families:

- *Outcome imputation methods* (Xu et al., 2024; Meir et al., 2025): Replace censored times with imputed survival times (e.g., IPCW-based reweighting introduced in Qi et al. (2023)). Then apply standard CATE estimators such as Causal Forests (Athey et al., 2019), Double-ML (Chernozhukov et al., 2018), or meta-learners including S(ingle)-, T(wo)-, X(cross)-, D(oubly)R(obust)-learners (Athey & Imbens, 2015; Künzel et al., 2019; Kennedy, 2023).[1]
- *Direct-survival CATE methods:* Extend causal inference directly to time-to-event outcomes, e.g., targeted learning (Van der Laan & Rose, 2011), tree-based estimators (Zhang et al., 2017), Bayesian approaches (Henderson et al., 2020), SurvITE (Curth et al., 2021a), or Causal Survival Forests (Cui et al., 2023).
- *Survival meta-learners* (Xu et al., 2023; Bo et al., 2024; Noroozizadeh et al., 2025): Adapt S(ingle)-, T(wo)-, or matching-learners to survival outcomes by using survival models such as Random Survival Forests or deep survival models.

While these approaches appear in disparate lines of work, we use the taxonomy above as an organizing lens for our benchmark. In particular, we implement 53 representative methods spanning the three families in a unified, modular framework to enable consistent evaluation across datasets and estimands. We note that several emerging directions fall outside this taxonomy (e.g., generative causal margin modeling (Yang et al., 2025) and synthetic-control-based methods (Curth et al., 2024; Han & Shah, 2025)), whose systematic integration we leave for future work.

Additionally, while our benchmark focuses on static treatments under selection on observables, related work addresses HTEs in alternative settings. This includes instrumental variable approaches for survival (Tchetgen et al., 2015), dynamic treatment regimes (Rudolph et al., 2022; Bates et al., 2022; Rudolph et al., 2023; Cho et al., 2023), and Bayesian machine learning approaches (Chen et al., 2024). Additionally, Targeted Maximum Likelihood Estimation-based methods (Stitelman & van der Laan, 2010; Stitelman et al., 2011) offer robust estimation for survival parameters, though primarily for average or subgroup effects rather than continuous CATE functions.

## 3 SURVHTE-BENCH

SURVHTE-BENCH probes how survival CATE estimators behave when assumptions (A1)–(A5) hold and when they are either mildly or severely violated. As real data with ground-truth CATEs are scarce, the bulk of our benchmark relies on synthetic datasets. We also include semi-synthetic data (real covariates with simulated treatments and outcomes) and two real-world datasets. As already stated in Section 2, in this paper we focus on the case where the target estimand is RMST up to a user-specified time horizon $h$ (other estimands are possible, such as survival probability at predefined

---

[1]Standard CATE estimators do not handle censoring. By imputing censored times with survival times as a preprocessing step, we make it appear as if there is no censoring, so standard CATE estimators can be applied.

times, see Appendix G.4.2). Access to all of the datasets, except for those requiring credentialed approval, is provided at: `https://huggingface.co/datasets/snoroozi/SurvHTE-Bench`.

**Synthetic data.** We construct a modular suite of 40 synthetic datasets that systematically vary across two orthogonal axes: (1) causal configuration: treatment mechanism, positivity, confounding, censoring mechanism; (2) survival scenario: event-time distribution and censoring rate. Crossing 8 causal configurations with 5 survival scenarios yields $8 \times 5 = 40$ synthetic datasets, each with binary, time-fixed treatment, five independently sampled covariates, each distributed as $\text{Uniform}(0, 1)$, and up to 50,000 units. For each unit $i$, we generate both $T_i(0)$ and $T_i(1)$, ensuring that ground-truth CATEs are always known.

The 8 **causal configurations** (Table 1) include randomized controlled trials (RCT-50, RCT-5) and observational stud-

Table 1: Causal configurations of synthetic datasets. RCT = randomized controlled trial; OBS = observational study; 50(5) = 50%(5%) treatment rate; CPS= correctly specified propensity score (ignorability satisfied); UConf = unobserved confounding (ignorability violated); NoPos = lack of positivity; InfC = informative censoring (ignorable censoring violated). ✓= held, ✗= not held.

| Causal Configs. | RCT | Ignorability | Positivity | Ignorable Censoring |
|---|---|---|---|---|
| RCT-50 | ✓ | ✓ | ✓ | ✓ |
| RCT-5 | ✓ | ✓ | ✓ | ✓ |
| OBS-CPS | ✗ | ✓ | ✓ | ✓ |
| OBS-UConf | ✗ | ✗ | ✓ | ✓ |
| OBS-NoPos | ✗ | ✓ | ✗ | ✓ |
| OBS-CPS-InfC | ✗ | ✓ | ✓ | ✗ |
| OBS-UConf-InfC | ✗ | ✗ | ✓ | ✗ |
| OBS-NoPos-InfC | ✗ | ✓ | ✗ | ✗ |

ies with correctly specified propensity scores (i.e., these are known during training) with all confounders observed in estimation (OBS-CPS), unobserved confounding (OBS-UConf), or lack of positivity (OBS-NoPos). Each observational setting has variants with suffix "-InfC", where ignorable censoring is replaced by informative censoring, where censoring times depend stochastically on event times. These violations reflect common real-world challenges: unmeasured risk factors in treatment decisions (violating ignorability), treatment imbalance in observational studies (violating positivity), and dropout mechanisms correlated with health outcomes (violating ignorable censoring). We do not model interference (consistency violations) or censoring-positivity violations, which require specialized designs beyond our scope. Additional variations, such as informative censoring with the censoring time driven by unobserved factors, are included in the Appendix I to illustrate the extensibility of our modular setup.

The 5 **survival scenarios** (Table 2) include Cox proportional hazards (low censoring), accelerated failure time (AFT) models (low and high censoring), and Poisson hazards (medium and high censoring). These distributions cover proportional hazards (Cox) and non-proportional hazards (AFT[2], Poisson), with censoring levels ranging from under 30% to over 70%. This variety reflects practical challenges like high censoring common in EHR cohorts, accelerated processes in oncology, and discrete hazard approximations in epidemiology. Within each survival

Table 2: Survival scenarios of synthetic datasets. "Low" <30%, "Med" 30-70%, "High" >70% censoring. AFT = accelerated failure time.

| Survival Scenario | Survival Time Distribution | Censoring Rate |
|---|---|---|
| A | Cox | Low |
| B | AFT | Low |
| C | Poisson | Med |
| D | AFT | High |
| E | Poisson | High |

scenario, coefficients are tuned so that event times are comparable across different causal configurations. Full generation formulas and summary statistics (e.g., censoring rate, treatment rate, ATE) for each dataset are in Appendix A.

**Evaluation metrics.** Per dataset, averaged over 10 random splits, we report:

- CATE root mean square error (RMSE): $\sqrt{\frac{1}{n} \sum_{i=1}^{n} (\hat{\tau}(X_i) - \tau(X_i))^2}$.
- ATE bias: $\frac{1}{n} \sum_{i=1}^{n} \hat{\tau}(X_i) - \Delta$, where $\Delta$ is the true ATE from the population and can be approximated using the average CATE from a very large sample (i.e., from 50,000 simulated samples).
- Auxiliary imputation accuracy: mean absolute error (MAE) between imputed and true event times.
- Auxiliary regression/survival fit: MAE for regression-based learners, AUC for propensity score models, and the time-dependent C-index (Antolini et al., 2005) for survival models.

---

[2]The AFT noise distribution we use (that is additive in log survival time) is Gaussian so that the resulting model does *not* satisfy the proportional hazards assumption (which would require the noise to be Gumbel).

**Survival CATE methods implemented.** We evaluate the three broad families of survival CATE methods (53 variants total; see Appendix C for the full list, Appendix D for methodological details):

- *Outcome imputation methods*: meta-learners (S-, T-, X-, DR-Learners) paired with base regression learners (lasso, random forest, XGBoost), plus Double-ML and Causal Forest, each combined with the three imputations (Pseudo-obs, Margin, and IPCW-T (Qi et al., 2023), see Appendix B for details). In total, we implement 42 variants.
- *Direct-survival CATE methods*: We include SurvITE (Curth et al., 2021a) and Causal Survival Forests (Cui et al., 2023).
- *Survival meta-learners*: S-, T-, and matching-learners paired with survival learners (Random Survival Forests (Ishwaran et al., 2008), DeepSurv (Katzman et al., 2018), and DeepHit (Lee et al., 2018)), for a total of $3 \times 3 = 9$ variants.

Note that some implemented methods are straightforward extensions of existing ideas despite not having been previously published. For example, (Qi et al., 2023) suggested ways of replacing censoring times with imputed survival times for the purposes of model evaluation, but their imputation strategies naturally can be coupled with standard CATE learners to obtain survival CATE estimators. Similarly, pairing meta-learners with different base learners (e.g., lasso regression, XGBoost, or DeepSurv) yields natural yet previously unpublished variants.

**Semi-synthetic data.** We include 10 semi-synthetic datasets, pairing real covariates (ACTG HIV trial, MIMIC-IV ICU records) with simulated treatments and outcomes, covering moderate to extreme censoring regimes, covariate-dependent treatment assignment, and both linear and non-linear (interaction-based) event-time and censoring mechanisms. These datasets preserve realistic feature distributions while retaining ground-truth CATEs. Details are in Section 4.2.

**Real data.** Finally, we incorporate two real datasets, one with ground truth (for which we can use the same evaluation metrics as with synthetic data) and one without ground truth but with a low censoring rate (for which we compare how models perform on the original dataset vs on the dataset with artificially introduced censoring). These provide opportunities to evaluate how methods behave under real covariate and outcome structures. Details are in Section 4.3.

## 4 BENCHMARKING RESULTS

We now present benchmark results across synthetic, semi-synthetic, and real data, spanning controlled violations of causal assumptions to realistic covariate structures.

### 4.1 SYNTHETIC EXPERIMENT RESULTS AND ANALYSES

We begin with synthetic datasets, where we evaluate 53 estimator variants across the 40 synthetic datasets (Section 3), systematically spanning varying causal configurations and survival scenarios. This controlled setting enables us to probe estimator robustness under systematic violations of identification assumptions. Our analyses aim to address four questions: **(Q1)** Which estimators perform best overall in terms of CATE RMSE and ATE bias? **(Q2)** How do violations of causal assumptions (ignorability, positivity, ignorable censoring) affect performance? **(Q3)** How does the censoring rate influence estimation quality? **(Q4)** How do component choices (imputation algorithms and base learners) affect final CATE accuracy?

**Evaluation protocol.** For each synthetic dataset, we conduct experiments with a random selection of 5,000, 2,500, and 2,500 points for training, validation, and testing samples, repeated over 10 random splits. The validation set is used for selecting the best variant within each method family, while test sets are reserved strictly for evaluation. Additional convergence analyses with varying training set sizes are in Appendix F.7. Across all experiments, the horizon parameter $h$ is set to the maximum observed time in each dataset, which is a common practice that allows for consistent estimation of the RMST over the entire observed period. Further experimental details, including hyperparameters, are in Appendix E.

We present results using the following visualizations:

- **Borda count rankings.** To provide a clear summary across the diverse experimental settings, we adopt the Borda count method, which ranks methods by CATE RMSE in each dataset (lower is better) and then averages the ranks across datasets. This approach yields a single, interpretable score that reflects overall relative performance while accounting for variability across scenarios. Similar strategies have been used in other benchmarking studies (e.g., Han et al. 2022) to enable

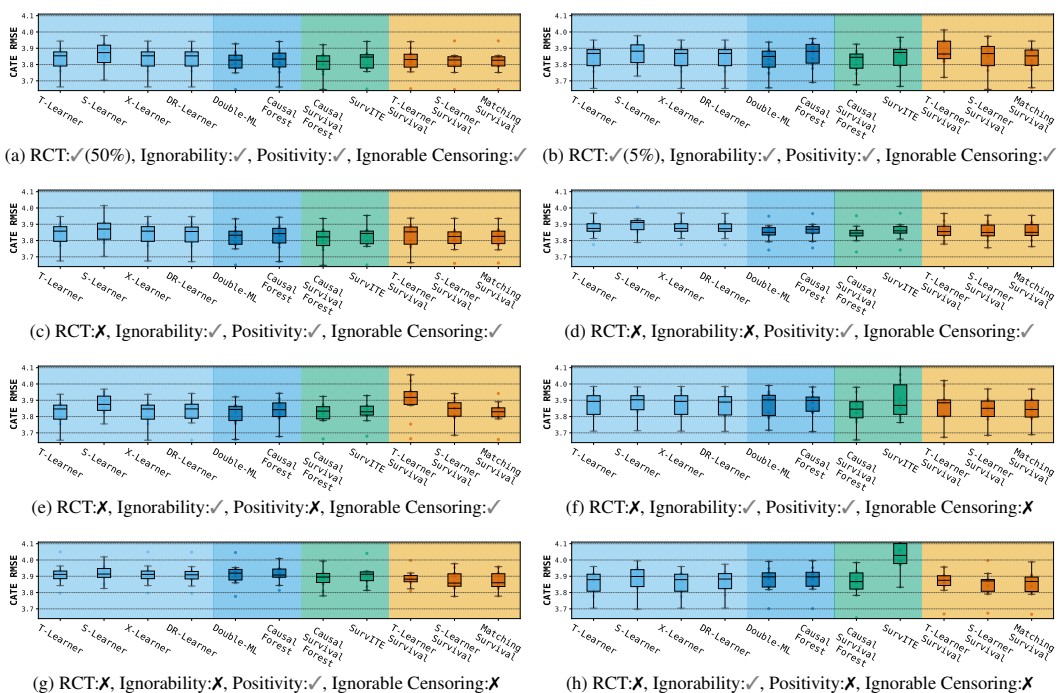

Figure 2: CATE RMSE in Scenario C across 10 experimental repeats.

transparent comparisons across heterogeneous tasks. We report rankings at two levels: (i) individual estimator variants (53 total; Figure 1, top), and (ii) aggregated method families, where the best variant per family is selected on validation data (11 total; Figure 1, bottom). The latter mimics a practical deployment setting where practitioners would tune and select the strongest model within a family. More granular rankings stratified by survival scenario (Figure 6) and by causal configuration (Figure 7) are provided in Appendix F.3.

- **CATE RMSE.** We report absolute CATE RMSE across 10 repeats, grouped by survival scenario, with one panel per set of eight causal configurations. In the main paper, we show Scenario C as an illustrative example (Figure 2); results for the other scenarios are deferred to Appendix F.4.
- **ATE bias.** We report ATE bias results, computed analogously to CATE RMSE, in Appendix F.5. While the focus of this benchmark is on CATE estimation, these serve as a complementary check.
- **Win-rate analyses.** Complementing the Borda rankings, we also report win-rates that quantify how often each method family attains Top-1, Top-3, and Top-5 performance according to CATE RMSE and ATE bias across all synthetic experiments in Appendix F. Overall win-rates aggregated over all survival scenarios and causal configurations are summarized in Table 15, while scenario-specific and configuration-specific win-rates are reported in Tables 16, 17, and 18. These summaries highlight not only which methods perform well on average, but also which ones most consistently appear among the top performers under varying censoring regimes and patterns of causal-assumption violations.

Additionally, in Appendix F.6, we report a series of auxiliary evaluations of key components, including imputation error (Appendix F.6.1) for imputation-based methods and regression model accuracy (Appendix F.6.2) or survival model performance (Appendix F.6.3) for meta-learners. These results establish how component-level performance relates to downstream CATE estimation.

**Key findings.** Overall, performance is strongly context-dependent. For example, in low-censoring randomized settings, outcome imputation methods such as X-Learner and Double-ML excel. As censoring intensifies or when assumptions are violated, survival meta-learners and Causal Survival Forests gain a clear advantage. Within method families, the choice of imputation algorithm or survival base model critically determines outcomes. Appendix F further supports this pattern quantitatively through Top-$k$ frequency summaries of CATE RMSE and ATE Bias (Tables 15- 18), highlighting methods that consistently rank at the top rather than merely performing well on average. We summarize detailed findings below.

**Overall performance (Q1).** Figure 1 (top) presents Borda count rankings of the top-10 performing methods out of the 53 total configurations evaluated (full ranking in Appendix F.1). The highest-performing estimators are survival meta-learners built on DeepSurv, with S-Learner-Survival (average rank 5.17 across 53 methods) and Matching-Survival (5.42) leading, followed by Double-ML with Margin imputation (6.65). Among outcome imputation approaches, Margin appears most frequently among the top performers, though Pseudo-obs and IPCW-T are also represented.

At the method family level (Figure 2 for Scenario C and Figure 1 (bottom)

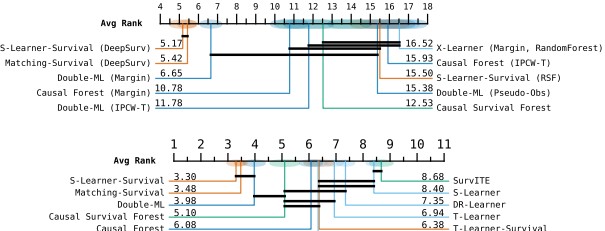

Figure 1: (top) Borda count rankings of the top 10 estimator variants (out of 53 total), based on CATE RMSE across 40 datasets and averaged over 10 repeats (lower is better). (bottom) Family-level rankings, where for each dataset the best method variant within each method family is chosen using validation performance and then ranked on the held-out test set. Black bands connect methods without statistically significant differences (Wilcoxon signed-rank test, FDR-corrected at $\alpha = 0.05$). Shaded regions indicate the standard error of the rank across datasets.

across all causal configurations and survival scenarios), we see how each approach performs when optimally configured. At this level, S-Learner-Survival (average rank 3.30 across 11 method families) and Matching-Survival (3.48) maintain their advantage, followed by Double-ML (3.98) and Causal Survival Forests (5.10).

**Violations of causal assumptions (Q2).** Performance shifts substantially depending on assumption violations (Figure 7). In randomized balanced trials (RCT-50), outcome imputation methods dominate, with Double-ML (3.60) and Causal Forest (5.60) showing performance comparable to high-ranking survival meta-learners. However, under imbalanced treatment (RCT-5), Double-ML remains strong (1.80), while T-Learner-Survival—which relies on fitting base models on treated units—drops to last place (9.00), alongside SurvITE, due to sparsity of treated samples.

Under ignorability violation (OBS-UConf, Figure 7d), Double-ML and X-Learner are the main competitive imputation methods, whereas survival meta-learners retain relatively stable performance. Examining ATE bias (Figures 13d-17d), we see that across all scenarios, survival meta-learners and Causal Survival Forests methods maintain relatively consistent bias levels despite ignorability violations, whereas meta-learners exhibit a slightly increased bias at times. This stability is also reflected in aggregated win-rate summaries for OBS-UConf, where survival-aware families most consistently appear among the Top-$k$ for both CATE RMSE and ATE Bias (Appendix F.3, Table 18).

Under positivity violation (OBS-NoPos, Figure 7e), we see the more sophisticated outcome imputation approaches like Double-ML and X-Learner maintain strong performance and outperform survival meta-learners. However, when positivity violations occur alongside other violations (Figure 7h), survival meta-learners regain their advantage, demonstrating their robustness to multiple simultaneous violations. This behavior is also reflected in the win-rate summaries, which show a similar shift in Top-$k$ coverage across configurations (Appendix F.3, Table 18). Additionally, Causal Survival Forests sees a large drop in its ranking when faced with only positivity violation, suggesting its limited robustness to regions of covariate space with deterministic treatment assignment.

Under informative censoring (InfC, Figure 2f- 2h), survival meta-learners and Causal Survival Forest continue to outperform outcome imputation approaches. However, all methods show degraded performance compared to their ignorable censoring counterparts, with substantially higher CATE RMSE variability, indicating the increased difficulty of estimation under dependent censoring.

**Impact of censoring rate (Q3).** For the impact of censoring rate and survival time distribution (Figure 6), in low-censoring Scenarios A and B, Double-ML leads the rankings, but as censoring increases through Scenarios C to E, survival meta-learners and Causal Survival Forests progressively move to the top. By Scenario D (high censoring), S-Learner-Survival (1.6) and Matching-Survival (2.4) dramatically outperform all other approaches. This pattern suggests that direct survival modeling provides increasing advantages as censoring rates rise, likely due to better handling of the uncertainty in heavily censored data compared to outcome imputation approaches. The same mono-

tonic shift toward survival-aware methods with increasing censoring is visible in their Top-$k$ win-rate coverage across scenarios (Appendix F.2).

Separately, in Appendix F.5, we show ATE bias across different datasets. We observe apparent divergence of the estimated ATE from the true ATE in Scenario D and slightly in Scenario E (Figure 16, 17), where the censoring rate is very high. Especially when the true underlying event time follows an AFT distribution (Scenario D), almost all estimators failed under all different causal configurations, suggesting the challenging task of treatment effect estimation under a high censoring rate.

**Component effects on CATE estimation (Q4).** Auxiliary evaluations in Appendix F.6 demonstrate that both imputation accuracy and base learner performance influence downstream CATE estimation. Among outcome imputation methods, Margin consistently achieves the lowest imputation error and degrades the least under heavy censoring (Appendix F.6.1), which translates into Margin-based variants appearing more frequently among the top-ranked estimators (Figure 1). For survival meta-learners, higher concordance indices of DeepSurv across survival scenarios (Appendix F.6.3) explain why DeepSurv-based configurations dominate overall rankings.

## 4.2 SEMI-SYNTHETIC DATA RESULTS

To bridge the gap between controlled synthetic experiments and real-world complexity, we evaluate methods on semi-synthetic datasets that pair real covariate distributions with simulated treatments and outcomes. This approach addresses a critical limitation of purely synthetic data—the potential lack of representativeness in covariate structures—while maintaining ground-truth CATEs for rigorous evaluation. These datasets preserve real-world covariate correlations, mixed data types, and high dimensionality while enabling controlled evaluation against known treatment effects.

**Dataset construction.** We construct 10 semi-synthetic datasets using covariates from two sources (Table 28 summarizes all datasets).

- **ACTG semi-synthetic**: Based on 23 covariates from the ACTG HIV clinical trial (Hammer et al., 1996), with treatment and event times simulated following Chapfuwa et al. (2021). This dataset exhibits moderate censoring (51%) with realistic treatment imbalance.
- **MIMIC semi-synthetic**: Derived from 36 covariates in the MIMIC-IV ICU database (Johnson et al., 2023). We construct nine MIMIC-based semi-synthetic variants organized into two complementary subsets: MIMIC-$i$–$v$ follow Meir et al. (2025) and vary censoring severity from 53% to 88% under covariate-independent treatment assignment, covering moderate to extreme censoring regimes common in longitudinal EHR studies. MIMIC-$vi$–$ix$ reuse the same covariates but introduce covariate-dependent treatment assignment and non-linear (interaction-based) event-time and censoring mechanisms. Full generative details are provided in Appendix G.

**Primary estimand and reporting.** In the main paper, we focus on CATE estimation for RMST evaluated at a large horizon (the maximum observed time in each dataset). Table 3 reports RMSE on ACTG and the censoring-severity sweep MIMIC-$i$–$v$, which provides a compact view of performance as censoring increases. Results on the remaining MIMIC variants (MIMIC-$vi$–$ix$) under the same RMST estimand, as well as results for additional estimands (horizon-specific survival-probability CATEs and RMST at a shorter horizon), are reported in Appendix G.

Table 3 presents CATE RMSE results across semi-synthetic datasets, showing how realistic covariate structure modulates the core performance patterns observed in synthetic experiments.

**Validation and extension of synthetic findings.** The semi-synthetic results broadly corroborate the synthetic benchmark: In ACTG, Double-ML achieves the lowest RMSE (10.65), consistent with the advantage of flexible, doubly robust approaches in moderate-dimensional settings with controlled confounding. Across MIMIC-$i$–$v$, methods remain competitive within a narrow band, with survival-oriented approaches (SurvITE and survival meta-learners) frequently among the top performers.

**Censoring sensitivity and stability.** The MIMIC-$i$–$v$ sweep (53%–88% censoring) reveals that differences in *stability* can be as important as differences in mean RMSE. For example, S-Learner-Survival remains stable across censoring levels (RMSE range 7.897–7.921), while T-Learner-Survival exhibits increased variability under extreme censoring (e.g., larger standard deviation at 82% censoring).

**Practical implications.** In these realistic covariate spaces, RMSE differences are often compressed, making method selection depend on secondary considerations such as stability under censoring,

Table 3: CATE RMSE (mean $\pm$ std over 10 repeats) on ACTG and MIMIC-$i$-$v$ semi-synthetic datasets. Best two methods per dataset are **bolded**.

| Method Family (censoring rate) | ACTG (51%) | MIMIC-$i$ (88%) | MIMIC-$ii$ (82%) | MIMIC-$iii$ (74%) | MIMIC-$iv$ (66%) | MIMIC-$v$ (53%) |
|---|---|---|---|---|---|---|
| *Outcome Imputation Methods* | | | | | | |
| T-Learner | $11.257 \pm 0.239$ | $7.964 \pm 0.046$ | $\mathbf{7.912 \pm 0.046}$ | $7.915 \pm 0.043$ | $7.912 \pm 0.043$ | $7.908 \pm 0.043$ |
| S-Learner | $11.300 \pm 0.221$ | $7.977 \pm 0.044$ | $7.968 \pm 0.047$ | $7.956 \pm 0.050$ | $7.959 \pm 0.046$ | $7.958 \pm 0.048$ |
| X-Learner | $\mathbf{11.072 \pm 0.196}$ | $7.964 \pm 0.046$ | $\mathbf{7.912 \pm 0.046}$ | $7.915 \pm 0.043$ | $7.912 \pm 0.043$ | $7.908 \pm 0.043$ |
| DR-Learner | $11.334 \pm 0.225$ | $7.964 \pm 0.046$ | $\mathbf{7.912 \pm 0.047}$ | $7.911 \pm 0.043$ | $7.911 \pm 0.043$ | $7.909 \pm 0.043$ |
| Double-ML | $\mathbf{10.651 \pm 0.239}$ | $7.954 \pm 0.047$ | $7.936 \pm 0.045$ | $7.919 \pm 0.044$ | $7.917 \pm 0.046$ | $\mathbf{7.891 \pm 0.050}$ |
| Causal Forest | $11.154 \pm 0.175$ | $7.967 \pm 0.045$ | $7.949 \pm 0.044$ | $7.934 \pm 0.043$ | $7.931 \pm 0.047$ | $7.909 \pm 0.044$ |
| *Direct-Survival CATE Methods* | | | | | | |
| Causal Survival Forests | $11.674 \pm 0.169$ | $7.963 \pm 0.057$ | $7.942 \pm 0.039$ | $7.929 \pm 0.037$ | $7.911 \pm 0.051$ | $\mathbf{7.893 \pm 0.042}$ |
| SurvITE | $12.714 \pm 0.559$ | $\mathbf{7.931 \pm 0.050}$ | $\mathbf{7.908 \pm 0.065}$ | $\mathbf{7.906 \pm 0.071}$ | $7.907 \pm 0.058$ | $7.906 \pm 0.066$ |
| *Survival Meta-Learners* | | | | | | |
| T-Learner-Survival | $11.428 \pm 0.160$ | $8.007 \pm 0.075$ | $7.980 \pm 0.233$ | $7.911 \pm 0.054$ | $\mathbf{7.902 \pm 0.042}$ | $7.902 \pm 0.046$ |
| S-Learner-Survival | $11.713 \pm 0.237$ | $\mathbf{7.921 \pm 0.044}$ | $\mathbf{7.912 \pm 0.052}$ | $\mathbf{7.900 \pm 0.045}$ | $\mathbf{7.901 \pm 0.046}$ | $7.897 \pm 0.042$ |
| Matching Survival | $12.523 \pm 0.289$ | $7.949 \pm 0.043$ | $7.935 \pm 0.053$ | $7.920 \pm 0.047$ | $7.921 \pm 0.046$ | $7.912 \pm 0.042$ |

interpretability, and computational cost. Overall, the semi-synthetic evaluation suggests: (i) flexible causal methods such as Double-ML remain strong in moderate-dimensional settings with balanced censoring, (ii) survival-oriented methods like SurvITE and survival meta-learners provide robust performance in highly censored EHR-like regimes (MIMIC), and (iii) the choice of method should explicitly consider dataset dimensionality and covariate complexity, not just censoring rates and sample sizes. In Appendix G, we provide details on the data generation process, experiment setup, and additional experiment results on the semi-synthetic.

## 4.3 BENCHMARKING ON REAL DATA

We also evaluate the three families of survival CATE estimators on two real-world datasets, one with known ground truth and one without.

**Twin data.** The Twins dataset (Almond et al., 2005; Curth et al., 2021a) includes twin births from 1989-1991, where being the heavier twin serves as treatment and time to mortality as the outcome. With known outcomes for both twins, this dataset provides ground truth for CATE evaluation. After replicating the same

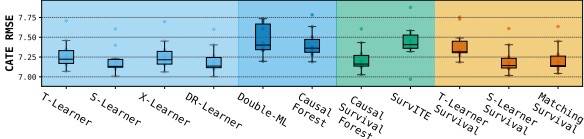

Figure 3: CATE RMSE for twin birth data with $h = 30$ days across 10 experimental runs.

random treatment assignment strategy and the censoring time assignment following Curth et al. (2021a), the treatment rate and censoring rate for the dataset are 68.1% and 84.8% respectively across 11,400 twin pairs. Since most of the mortality events occur within 30 days, we use $h = 30$ days during estimation. Figure 3 shows that S- and DR-Learners (with imputation) and S-Learner-Survival exhibit lower CATE RMSE (7.2 days). T-Learner-Survival and Causal Forest with imputation exhibit the worst performance, consistent with their overall lower ranking from the benchmarking on our synthetic datasets (Figure 1). Surprisingly, Double-ML with imputation exhibits the worst performance on the twin data, which is different from the overall ranking, suggesting potential unique patterns in this dataset. In Appendix H, we also show the result with $h = 180$ days; the conclusions are similar.

**HIV clinical trial.** The ACTG 175 dataset (Hammer et al., 1996) compared four antiretroviral treatments in 2,139 HIV-infected patients. Following Meir et al. (2025), we convert time to months with $h$=30 months (13.7% baseline censoring) and introduce artificial censoring to test robustness (increasing to >90% censoring). More details on data and processing can be found in Appendix H. Figure 4 compares CATE estimates between baseline and high-censoring conditions for the ZDV vs. ZDV+ddI comparison (results for other treatment comparisons are in Appendix H). Each point represents an individual patient, with the 45-degree dashed line indicating perfect consistency between conditions. We observe distinct behavioral patterns: Causal Survival Forest (green) produces estimates that cluster tightly around their original values; outcome imputation methods (blue) show

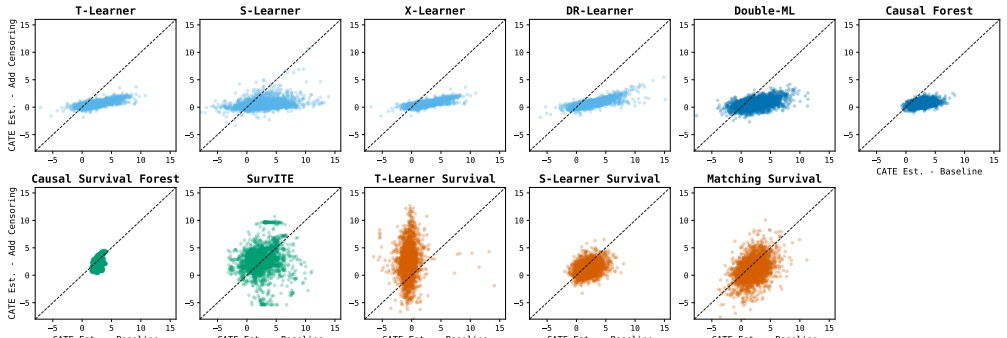

Figure 4: CATE estimation comparison between baseline and high-censoring conditions under ZDV vs. ZDV+ddI treatments. Each point represents an individual patient, with the dashed diagonal line indicating perfect consistency between baseline CATE estimation and that with the additional censoring injected.

higher variation in baseline estimates but concentrated predictions under high censoring; survival meta-learners (red) display substantial deviations from the 45-degree line, indicating sensitivity to censoring conditions. As ground truth is unknown, we cannot determine which approach is more accurate, but these patterns reveal fundamental differences in how estimators respond to increased censoring. For example, survival meta-learners (the red scatter plots), especially the T- and matching-learners, exhibit instability under increased censoring settings (large variance in y-axis values).

## 5 DISCUSSION

SURVHTE-BENCH provides the first comprehensive and extensible platform for systematically benchmarking heterogeneous treatment effect estimators under right-censored survival settings. By spanning synthetic, semi-synthetic, and real datasets, the benchmark enables both controlled stress-testing of estimators under systematic assumption violations and validation in realistic clinical-like settings. Our empirical evaluations reveal strengths and weaknesses across estimator families.

While we have attempted to make our benchmark representative of common causal survival setups, various limitations remain. First, the synthetic datasets include numerous scenarios representing common real-world violations; however, they do not encompass all possible complexities, such as RCT settings with informative censoring or varying degrees of severity in assumption violations. The binary nature of our violations (either present or absent) may not capture the nuanced continuum of partial violations. We recognize that in real-world applications, assumption violations often exist on a continuum of severity. Future extensions of our benchmark could incorporate graded sensitivity analyses, such as varying the magnitude of unmeasured confounding (e.g., via Rosenbaum's $\Gamma$) or the degree of overlap violation. This would allow for a more granular "dose-response" analysis to pinpoint the exact thresholds at which specific estimators break down. Second, the choice of the most relevant causal estimand is inherently task-specific. While our evaluations cover restricted mean survival time alongside survival probabilities at fixed horizons (Appendix G.4.2), different clinical or policy objectives often dictate the need for alternative measures. Extending the benchmark to natively support a wider array of task-specific estimands—such as conditional median survival times or time-varying hazard ratios—would further enrich its scope. Additionally, we limit our analysis to static, binary treatments with fixed baseline covariates, excluding scenarios involving time-varying treatments, instrumental variables, and dynamic covariate structures.

Future work could expand SURVHTE-BENCH in several directions. Incorporating a wider variety of direct causal estimation methods, such as g-computation approaches specifically designed for survival outcomes, would provide an even more comprehensive evaluation landscape, especially because Causal Survival Forests proved to be competitive but showed vulnerability to certain assumption violations like positivity. Exploring more complex data-generating mechanisms that better mimic the heterogeneity and longitudinal nature of real-world clinical data represents another promising direction. Finally, extending the benchmark to support multi-valued or continuous treatments would address important practical scenarios encountered in precision medicine and policy optimization.

ACKNOWLEDGMENTS

This research was supported by the Division of Intramural Research (DIR) of the National Library of Medicine (NLM), National Institutes of Health (NIH). G.H.C. was supported by NSF CAREER award #2047981. S.N. was supported by Carnegie Mellon University TCS Presidential Fellowship, and Natural Sciences and Engineering Research Council of Canada (NSERC) PGS-D award. S.N. was also supported in part by an appointment to the National Library of Medicine Research Participation Program administered by the Oak Ridge Institute for Science and Education (ORISE) through an interagency agreement between the U.S. Department of Energy (DOE) and the National Library of Medicine, National Institutes of Health. ORISE is managed by ORAU under DOE contract number DE-SC0014664. All opinions expressed in this paper are the authors' and do not necessarily reflect the policies and views of NIH, NLM, DOE, or ORAU/ORISE.

ETHICS STATEMENT

SURVHTE-BENCH has significant positive potential for improving personalized medicine and clinical decision-making by enabling systematic evaluation of survival analysis methods under realistic assumption violations. By providing standardized benchmarks and practical guidance on when different estimators excel or fail, our work could accelerate the development of more reliable causal inference methods for high-stakes healthcare applications, ultimately supporting better patient outcomes through more informed treatment selection.

At the same time, our benchmark carries potential risks if misapplied. Practitioners may misinterpret benchmark results or place undue confidence in algorithmic decision-making, which could reduce necessary human oversight in clinical contexts. Moreover, although our study is methodological and does not involve human subjects directly, differences in estimator performance across demographic groups could exacerbate existing healthcare disparities if ignored. We therefore stress that our benchmark should not be used as a substitute for rigorous domain-specific validation, fairness assessment, or clinical trial evidence.

All datasets used in this work are either publicly available synthetic or semi-synthetic datasets, or real-world datasets with proper access provisions (e.g., credentialed approval for MIMIC-IV). No personally identifiable information was used, and all data handling complies with the terms of use of the original sources. We encourage future applications of SURVHTE-BENCH to incorporate fairness audits, domain-specific validation, and appropriate safeguards to ensure responsible deployment.

REPRODUCIBILITY STATEMENT

We provide complete resources to reproduce our results across synthetic, semi-synthetic, and real-data settings. (1) *Synthetic data:* The benchmark design and evaluation protocol are described in the main text (Sections 3 and 4.1), including the 8 causal configurations and 5 survival scenarios (40 datasets total). Extended generation formulas and per-dataset summaries are in Appendix A; imputation procedures in Appendix B; the full list of implemented estimators in Appendix C; causal method overviews in Appendix D; training details and hyperparameter grids in Appendix E; and additional synthetic results/analyses in Appendix F. (2) *Semi-synthetic data:* Setup, statistics, and full results appear in Appendix G with summary discussion in Section 4.2. (3) *Real data:* Processing details and additional analyses are provided in Appendix H; see also Section 4.3. We further study additional censoring mechanisms in Appendix I.

**Code and instructions.** The full codebase used for all experiments is available at:`https://github.com/Shahriarnz14/SurvHTE-Bench`, with scripts and READMEs to reproduce all results in the paper.

**Datasets.** In the same repository as well as at `https://huggingface.co/datasets/snoroozi/SurvHTE-Bench`, we include: (i) the complete synthetic suite (40 datasets from the $8 \times 5$ design); (ii) the semi-synthetic datasets, comprising the *ACTG (semi-synthetic)* dataset; and (iii) real-data materials for *Twins* and *ACTG 175*. For the semi-synthetic MIMIC resources, because MIMIC-IV requires credentialed access, we provide code to generate these datasets rather than redistributing raw MIMIC data. The MIMIC-IV dataset itself is hosted on PhysioNet at https://physionet.org/content/mimiciv/3.1/ and is publicly available to researchers upon credentialed approval. All other datasets listed above are included in the supplementary package in preprocessed or generated form, together with scripts to reproduce all splits and metrics.

In addition to enabling replication of our reported results, we intend SURVHTE-BENCH to serve as community infrastructure for the evaluation of survival HTE methods. The benchmark is designed to be modular and extensible, allowing researchers to incorporate new estimators or datasets while preserving comparability. This ensures not only reproducibility of our experiments but also a lasting resource for the community, providing a standardized basis for measuring progress in survival causal inference, a resource that has been missing until now, as well as in related areas of machine learning.

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

## APPENDIX

In this appendix, we provide detailed descriptions of data generation processes, methodological explanations, experimental setups, and results supplementing the main text. We begin by describing the mathematical formulations used to create our synthetic datasets, followed by detailed explanations of imputation methods and causal inference techniques. We then provide comprehensive information about model training procedures and hyperparameter settings for reproducibility. The appendix concludes with additional experimental results on synthetic, semi-synthetic, and real-world datasets.

We would like to declare **the use of Large Language Models (LLMs)** in this work. LLMs were used as general-purpose assistive tools. Specifically, they supported parts of the writing process (editing, formatting, and polishing text) without contributing to the core methodology, scientific rigor, or originality of the research. In addition, LLMs were used to assist with improving visualization code for figures, documenting the code, and minor refactoring. No part of the conceptualization, design, or execution of the research relied on LLMs.

**Appendix A: Additional Details of the Synthetic Datasets.** This section provides the mathematical formulations for generating covariates, treatment assignments, event times, and censoring times across different scenarios. It describes how the synthetic datasets systematically vary across causal configurations and survival scenarios, including details on covariate generation, treatment assignment mechanisms, event time generation, censoring time generation, and observed data construction. This section also includes Kaplan-Meier curves for the synthetic event-time and censoring distributions, illustrating scenario-level variation used throughout the benchmark.

**Appendix B: Imputation Methods Details.** This section explains three surrogate imputation strategies for estimating true event time in right-censored survival data: Margin Imputation, IPCW-T Imputation, and Pseudo-observation Imputation. It provides mathematical formulations for each method and discusses their respective advantages and limitations.

**Appendix C: List of CATE Estimators in SURVHTE Benchmark.** This section details the 53 different conditional average treatment effect (CATE) estimator variants evaluated in the benchmark, including outcome imputation methods, direct-survival CATE models, and survival meta-learners, with a breakdown of how these variants are constructed.

**Appendix D: Detailed Overview of Causal Inference Methods.** This section provides comprehensive explanations of various causal inference methods, including meta-learners (T-Learner, S-Learner, X-Learner, DR-Learner), Double-ML, Causal Forest, Causal Survival Forests, SurvITE, and Survival Meta-Learners, discussing their implementation in survival contexts.

**Appendix E: Model Training Details and Hyperparameters.** This section covers the hyperparameter grids, model selection procedures, and computational costs associated with each method class evaluated in the benchmark, providing details on the experimental setup for reproducibility.

**Appendix F: Additional Experimental Results for Synthetic Dataset.** This section presents comprehensive experimental results on synthetic datasets, including full rankings of models, win-rate summaries (Top-1 / Top-3 / Top-5) across methods, performance across different survival scenarios and causal configurations, detailed CATE RMSE and ATE bias plots, evaluation of auxiliary components, and convergence behavior under varying training set sizes.

**Appendix G: Semi-Synthetic Datasets.** This section includes data setup and detailed analysis of semi-synthetic datasets derived from ACTG 175 and MIMIC-IV, including covariate statistics, censoring rate range, and comprehensive performance results across methods. It also presents results for the survival-probability-based CATE estimand across multiple horizons, and includes sensitivity analyses of the RMST-based CATE where the evaluation horizon is varied.

**Appendix H: Real-World Datasets.** This section provides detailed descriptions of data preprocessing and additional experimental results for the Twins dataset and the ACTG 175 HIV clinical trial dataset, including CATE RMSE results with different time horizons and comparisons of CATE estimates between baseline and high-censoring conditions.

**Appendix I: Informative Censoring via Unobserved Confounding.** An additional dataset violating the ignorable censoring assumption via latent confounders. This illustrates extensibility of SURVHTE-BENCH to incorporate alternative censoring mechanisms beyond the $8 \times 5$ design.

# A    ADDITIONAL DETAILS OF THE SYNTHETIC DATASETS

Our synthetic datasets systematically vary across two orthogonal dimensions: causal configurations (treatment assignment mechanisms and assumption violations) and survival scenarios (event-time distributions and censoring mechanisms). This section provides the mathematical formulations for generating covariates, treatment assignments, event times, and censoring times across all scenarios. For event time and censoring time distribution, we adapt the generation process from Meir et al. (2025) and make some adjustments, with, for example, different censoring mechanisms under informative censoring settings; for treatment assignment in observational study settings, we adapt the propensity score from Cui et al. (2023). For simplicity, we omit the unit index $i$ in this section.

## A.1    COVARIATE GENERATION

Following Cui et al. (2023); Meir et al. (2025), for all scenarios, we generate five baseline covariates independently from uniform distributions:

$$X_m \sim \text{Uniform}(0, 1), \quad m = 1, 2, 3, 4, 5$$

Additionally, we generate two latent confounders $U_1, U_2 \sim \text{Uniform}(0, 1)$ that are used when testing violations of the ignorability assumption.

## A.2    TREATMENT ASSIGNMENT MECHANISMS

The treatment assignment mechanism $W$ varies according to the causal configuration:

**Randomized controlled trials (`RCT-50, RCT-5`):** Treatment is assigned randomly with probability $p$:

$$W \sim \text{Bernoulli}(p)$$

where $p = 0.5$ for `RCT-50` and $p = 0.05$ for `RCT-5`.

**Observational studies (`OBS-`):** Treatment assignment depends on covariates through a propensity score mechanism:

$$e(X) = \frac{1 + \text{Beta}(X_1; 2, 4)}{4} \quad \text{(OBS-CPS)}$$

$$e(X, U) = \frac{1 + \text{Beta}(0.3X_1 + 0.7U_1; 2, 4)}{4} \quad \text{(OBS-UConf)}$$

$$e(X) = \begin{cases} 1 & \text{if } X_1 > 0.8 \\ 0 & \text{if } X_1 < 0.2 \quad \text{(OBS-NoPos)} \\ 0.5 & \text{otherwise} \end{cases}$$

where $\text{Beta}(x; a, b)$ denotes the Beta probability density function with parameters $a$ and $b$ evaluated at $x$. For all observational configurations, $W \sim \text{Bernoulli}(e(\cdot))$.

## A.3    EVENT TIME GENERATION

Event times $T(w)$ under treatment $w \in \{0, 1\}$ are generated according to five different survival scenarios:

**Scenario A (Cox model):** Event times follow a Cox proportional hazards model with Weibull baseline hazard:

$$\begin{aligned} \lambda_T(t|W, X) &= h_0(t) \cdot \exp(\beta^T Z) \\ &= 0.5t^{-0.5} \cdot \exp[X_1 + (-0.5 + X_2) \cdot W + \epsilon] \end{aligned}$$

where $h_0(t) = 0.5t^{-0.5}$ is the Weibull baseline hazard with shape parameter $k = 0.5$ and scale parameter $\lambda_0 = 1.0$, and $\epsilon = 0.5(U_1 - X_2)$ if unobserved confounding is present, and $\epsilon = 0$ otherwise. Event times are generated via inverse transform sampling from the corresponding survival function.

**Scenario B (AFT model):** Event times follow an Accelerated Failure Time (AFT) model:

$$\begin{aligned} \log T(w) = &-1.85 - 0.8 \cdot \mathbb{1}(X_1 < 0.5) + 0.7\sqrt{X_2} + 0.2X_3 \\ &+ [0.7 - 0.4 \cdot \mathbb{1}(X_1 < 0.5) - 0.4\sqrt{X_2}] \cdot W + \epsilon + \eta \end{aligned}$$

where $\eta \sim \mathcal{N}(0,1)$ and $\epsilon$ is defined as in Scenario A.

**Scenario C (Poisson model):** Event times follow a Poisson distribution:

$$\lambda(w) = X_2^2 + X_3 + 6 + 2(\sqrt{X_1} - 0.3) \cdot W + \epsilon$$
$$T(w) \sim \text{Poisson}(\lambda(w))$$

**Scenario D (AFT model):** Event times follow an AFT model with parameters adjusted for higher censoring:

$$\log T(w) = 0.3 - 0.5 \cdot \mathbb{1}(X_1 < 0.5) + 0.5\sqrt{X_2} + 0.2X_3$$
$$+ [1 - 0.8 \cdot \mathbb{1}(X_1 < 0.5) - 0.8\sqrt{X_2}] \cdot W + \epsilon + \eta$$

**Scenario E (Poisson model):** Event times follow a Poisson distribution with adjusted parameters:

$$\lambda(w) = X_2^2 + X_3 + 7 + 2(\sqrt{X_1} - 0.3) \cdot W + \epsilon$$
$$T(w) \sim \text{Poisson}(\lambda(w))$$

## A.4 CENSORING TIME GENERATION

Censoring times $C$ are generated differently across scenarios and depend on whether informative censoring is present:

**Ignorable censoring (non-InfC scenarios):**

Scenario A:  $C \sim \text{Uniform}(0,3)$

Scenario B:  $\lambda_C(t|W,X) = h_{0C}(t) \cdot \exp(\gamma^T Z)$
$= 2.0t^{1.0} \cdot \exp[\mu]$
where $\mu = -1.75 - 0.5\sqrt{X_2} + 0.2X_3 + [1.15 + 0.5 \cdot \mathbb{1}(X_1 < 0.5) - 0.3\sqrt{X_2}] \cdot W$

Scenario C:  $C = \begin{cases} \infty & \text{with probability } 0.6 \\ 1 + \mathbb{1}(X_4 < 0.5) & \text{with probability } 0.4 \end{cases}$

Scenario D:  $\lambda_C(t|W,X) = h_{0C}(t) \cdot \exp(\gamma^T Z)$
$= 2.0t^{1.0} \cdot \exp[\nu]$
where $\nu = -0.9 + 2\sqrt{X_2} + 2X_3 + [1.15 + 0.5 \cdot \mathbb{1}(X_1 < 0.5) - 0.3\sqrt{X_2}] \cdot W$

Scenario E:  $C \sim \text{Poisson}(3 + \log(1 + \exp(2X_2 + X_3)))$

For scenarios B and D, $h_{0C}(t) = 2.0t^{1.0}$ is the Weibull baseline hazard for censoring with shape parameter $k = 2.0$ and scale parameter $\lambda_0 = 1.0$. Censoring times are generated via inverse transform sampling from the corresponding survival function.

**Informative censoring (-InfC scenarios):** When testing violations of ignorable censoring assumptions, we replace the above mechanisms with:

$$C_i \sim \text{Exponential}(\text{rate} = \lambda_0 + \alpha \cdot T_i) \tag{2}$$

where $\lambda_0 = 1.0$ and $\alpha = 0.1$ are baseline parameters that create dependence between censoring and event times.

While in the main benchmark we induce informative censoring by making censoring times dependent on event times, this is not the only way to violate the ignorable censoring assumption. To demonstrate the extensibility of our modular design and for completeness, we additionally include in Appendix I a setting where informative censoring arises through unobserved confounding.

## A.5 OBSERVED DATA CONSTRUCTION

The observed survival data consists of:

$$\tilde{T} = \min(T, C) \quad \text{(observed time)}$$
$$\delta = \mathbb{1}(T \leq C) \quad \text{(event indicator)}$$

where $T = T(W)$ represents the factual event time under the observed treatment assignment.

The combination of these five survival scenarios with eight causal configurations yields our comprehensive benchmark of 40 synthetic datasets, each designed to test estimator performance under specific combinations of survival dynamics and causal assumption violations.

Table 4: Summary of event time and censoring time generation across survival scenarios

| Scenario | Event Time Distribution | Censoring Mechanism | Censoring Rate |
|----------|------------------------|---------------------|----------------|
| A | Cox (Weibull baseline, $k = 0.5$) | Uniform$(0, 3)$ | Low $(< 30\%)$ |
| B | AFT (Log-normal) | Cox (Weibull baseline, $k = 2.0$) | Low $(< 30\%)$ |
| C | Poisson | Piecewise uniform | Medium (30-70%) |
| D | AFT (Log-normal) | Cox (Weibull baseline, $k = 2.0$) | High $(> 70\%)$ |
| E | Poisson | Poisson | High $(> 70\%)$ |

Table 5: Censoring rate of synthetic datasets (50,000 samples). Notice that the censoring rates are different from Table 2 under informative censoring due to changes in the censoring distribution.

| Causal Configurations | Survival Scenarios | | | | |
|-----------------------|-------|-------|-------|-------|-------|
| | A | B | C | D | E |
| RCT-50 | 0.203 | 0.073 | 0.392 | 0.913 | 0.794 |
| RCT-5 | 0.200 | 0.036 | 0.390 | 0.881 | 0.770 |
| OBS-CPS | 0.201 | 0.066 | 0.393 | 0.914 | 0.789 |
| OBS-UConf | 0.201 | 0.073 | 0.392 | 0.918 | 0.795 |
| OBS-NoPos | 0.203 | 0.082 | 0.393 | 0.912 | 0.803 |
| OBS-CPS-InfC | 0.116 | 0.052 | 0.885 | 0.366 | 0.926 |
| OBS-UConf-InfC | 0.116 | 0.054 | 0.888 | 0.381 | 0.929 |
| OBS-NoPos-InfC | 0.116 | 0.058 | 0.891 | 0.403 | 0.932 |

Table 6: Treatment rate of synthetic datasets (50,000 samples).

| Causal Configurations | Survival Scenarios | | | | |
|-----------------------|-------|-------|-------|-------|-------|
| | A | B | C | D | E |
| RCT-50 | 0.502 | 0.502 | 0.502 | 0.502 | 0.502 |
| RCT-5 | 0.049 | 0.049 | 0.049 | 0.049 | 0.049 |
| OBS-CPS | 0.503 | 0.503 | 0.503 | 0.503 | 0.503 |
| OBS-UConf | 0.539 | 0.539 | 0.539 | 0.539 | 0.539 |
| OBS-NoPos | 0.500 | 0.500 | 0.500 | 0.500 | 0.500 |
| OBS-CPS-InfC | 0.503 | 0.503 | 0.503 | 0.503 | 0.503 |
| OBS-UConf-InfC | 0.539 | 0.539 | 0.539 | 0.539 | 0.539 |
| OBS-NoPos-InfC | 0.500 | 0.500 | 0.500 | 0.500 | 0.500 |

**Remark on parameter calibration.** The constants used in our synthetic generators are inherited from and aligned with prior causal-survival simulation setups (Cui et al., 2023; Meir et al., 2025), and are set to span distinct, interpretable regimes that the benchmark aims to cover. In particular, we calibrate (i) *censoring severity* by shifting the relative scales of event-time and censoring-time processes (e.g., the AFT intercept change between Scenarios B and D increases typical event times and, together with the corresponding censoring model, yields higher censoring in D); (ii) *treatment prevalence* and *confounding strength* by adjusting propensity-score weights so that most configurations

Table 7: Average treatment effect (ATE) of synthetic datasets (50,000 samples).

| Causal Configurations | Survival Scenarios | | | | |
|---|---|---|---|---|---|
| | A | B | C | D | E |
| RCT-50 | 0.163 | 0.125 | 0.750 | 0.724 | 0.754 |
| RCT-5 | 0.163 | 0.125 | 0.750 | 0.724 | 0.754 |
| OBS-CPS | 0.163 | 0.125 | 0.750 | 0.724 | 0.754 |
| OBS-UConf | 0.004 | 0.132 | 0.740 | 0.831 | 0.740 |
| OBS-NoPos | 0.163 | 0.125 | 0.750 | 0.724 | 0.754 |
| OBS-CPS-InfC | 0.163 | 0.125 | 0.750 | 0.724 | 0.754 |
| OBS-UConf-InfC | 0.004 | 0.132 | 0.740 | 0.831 | 0.740 |
| OBS-NoPos-InfC | 0.163 | 0.125 | 0.750 | 0.724 | 0.754 |

remain near balanced treatment except where imbalance is intentional (e.g., RCT-5), while allowing controlled dependence on observed or latent drivers; and (iii) *effect magnitude/heterogeneity* through the coefficients on $W$ and $W$–covariate interactions, which we keep in a moderate range for comparability across scenarios. These choices are not unique, and alternative parameterizations could yield valid benchmarks; our goal is to provide a principled and reproducible instantiation that cleanly separates survival dynamics from causal-assumption stress and produces a broad range of survival CATE evaluation settings.

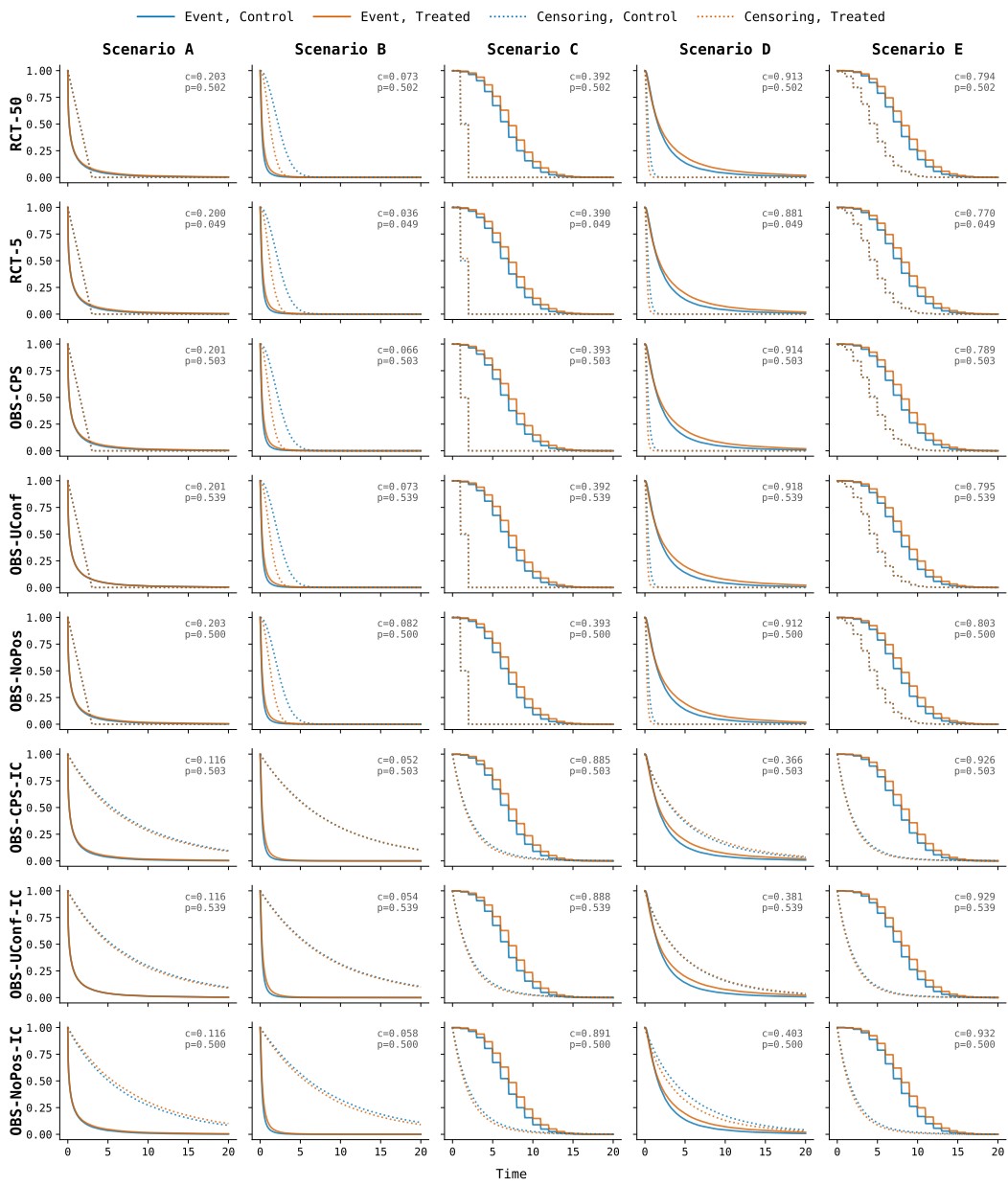

Figure 5: (Synthetic datasets) Kaplan-Meier curves across causal configurations (rows) and survival scenarios (columns). Solid lines show event-time survival under control (blue) and treatment (orange); dotted lines show censoring-time survival for each arm. Each panel reports the empirical censoring rate $c$ and treatment probability $p$.

## B    IMPUTATION METHODS DETAILS

We follow Qi et al. (2023) to implement three surrogate imputation strategies for estimating the true event time $T$ in right-censored survival data. Let $Y = \min(T, C)$ be the observed time, with censoring indicator $\delta = \mathbb{1}\{T \le C\}$. Let $S_{\text{KM}(\mathcal{D})}(t)$ denote the Kaplan-Meier estimate of the survival function using the dataset $\mathcal{D}$, and $N$ the number of subjects. The three methods below are used to impute a surrogate outcome $\tilde{T}_i$ for censored subject $i$ observed at time $t_i$.

**1. Margin imputation:**    This method assigns a "best guess" value to each censored subject using the nonparametric Kaplan-Meier estimator. This surrogate value, called the *margin time*, can be interpreted as the conditional expectation of the event time given that the event occurs after the censoring time. For a subject censored at time $t_i$, the margin-imputed event time is computed as:

$$\tilde{T}_i^{\text{margin}} = \mathbb{E}[T_i \mid T_i > t_i] = t_i + \frac{\int_{t_i}^{\infty} S_{\text{KM}(\mathcal{D})}(t)\, dt}{S_{\text{KM}(\mathcal{D})}(t_i)} \tag{3}$$

where $S_{\text{KM}(\mathcal{D})}(t)$ is the Kaplan-Meier survival estimate derived from the training dataset.

The reliability of this imputation depends on the censoring time. For example, if a subject is censored very early (e.g., at time 0), the margin time is highly uncertain due to the lack of observed data beyond that point. In contrast, if a subject is censored near the maximum observed follow-up, the margin time is more likely to be close to the true event time.

**2. IPCW-T imputation:**    This method imputes a surrogate event time for censored subjects based on the observed outcomes of subsequent uncensored individuals. Specifically, for a subject censored at time $t_i$, the imputed value is calculated as the average event time of all uncensored subjects with observed times after $t_i$:

$$\tilde{T}_i^{\text{IPCW}} = \frac{\sum_{j=1}^{N} \mathbb{1}\{t_i < t_j\} \cdot \mathbb{1}\{\delta_j = 1\} \cdot t_j}{\sum_{j=1}^{N} \mathbb{1}\{t_i < t_j\} \cdot \mathbb{1}\{\delta_j = 1\}} \tag{4}$$

This imputes the event time for subject $i$ by averaging the observed event times of those uncensored subjects who experienced the event after $t_i$. The method is motivated by the idea that these subsequent subjects provide empirical evidence about the possible timing of the unobserved event.

However, a limitation of this approach is that it fails to provide an imputation when there are no uncensored subjects observed after $t_i$. In such cases, the denominator of the expression becomes zero, and the method is unable to approximate the event time. In Qi et al. (2023), subjects for whom this occurs are excluded from evaluation, whereas in our setup we used the original observed time as the imputed time.

**3.    Pseudo-observation imputation:**    This method imputes the event time using pseudo-observations, which estimate the contribution of each subject to an overall unbiased estimator of the event time distribution. Let $\hat{\theta}$ be an estimator of the mean event time based on right-censored data, and let $\hat{\theta}^{-i}$ denote the same estimator computed with the $i$-th subject removed from the dataset. Then, the pseudo-observation for subject $i$ is defined as:

$$\tilde{T}_i^{\text{pseudo}} = e_{\text{Pseudo-Obs}}(t_i, \mathcal{D}) = N \cdot \hat{\theta} - (N - 1) \cdot \hat{\theta}^{-i} \tag{5}$$

This quantity can be interpreted as the individual contribution of subject $i$ to the overall estimate $\hat{\theta}$. In practice, both $\hat{\theta}$ and $\hat{\theta}^{-i}$ can be computed using the mean of the Kaplan-Meier survival curve:

$$\hat{\theta} = \mathbb{E}_t[S_{\text{KM}(\mathcal{D})}(t)], \quad \hat{\theta}^{-i} = \mathbb{E}_t[S_{\text{KM}(\mathcal{D} \setminus \{i\})}(t)]$$

Once the pseudo-observations $\tilde{T}_i^{\text{pseudo}}$ are computed for all censored subjects, they are substituted in place of the true event times for evaluation or modeling.

Although pseudo-observations are not exact conditional expectations, they can approximate $\mathbb{E}[T_i \mid X_i]$ under certain assumptions. In particular, when censoring is independent of covariates and the

sample size is large, pseudo-observations behave asymptotically like i.i.d. draws from the true conditional expectation:

$$\mathbb{E}[\tilde{T}_i^{\text{pseudo}} \mid X_i] \approx \mathbb{E}[T_i \mid X_i]$$

This makes the pseudo-observation method a principled, nonparametric approach for imputing censored survival times, particularly when estimating global quantities like the mean event time.

These imputation strategies enable us to transform the survival outcome into a fully observed target variable, allowing the application of standard regression-based methods in causal effect estimation. To ensure meaningful estimates, it is important that each imputed event time for a censored subject is guaranteed to be greater than or equal to the censoring time—reflecting the fact that the true event must occur after the last time it was observed. In our implementation, we manually enforce this constraint by setting the imputed value to the observed censoring time whenever the imputation procedure yields a value less than $t_i$.

## C    LIST OF CATE ESTIMATORS IN SURVHTE BENCHMARK

As mentioned in Section 3, in our benchmark, we evaluate three families of survival CATE methods, totaling 53 variants. We list the number of variants for each type of CATE estimator in Table 8.

For the outcome imputation methods, we first apply one of three imputation strategies (Pseudo-observation, Margin, or IPCW-T) (Qi et al., 2023) to handle the censored data, transforming the survival problem into a standard regression task. After imputation, we use these imputed outcomes with four different meta-learner frameworks (S-, T-, X-, and DR-Learners), each implemented with three different base regression models (Lasso Regression, Random Forest, and XGBoost), resulting in $3 \times 4 \times 3 = 36$ different variants. Additionally, we pair each imputation method with two specialized causal inference methods: Causal Forest (Athey et al., 2019) and Double-ML (Chernozhukov et al., 2018), which adds $3 \times 1 + 3 \times 1 = 6$ more variants, for a total of $42$ outcome imputation method variants.

For direct-survival CATE models, we include the Causal Survival Forests (CSF) (Cui et al., 2023) and SurvITE (Curth et al., 2021a), which are specifically designed to handle right-censored data without requiring separate imputation steps. SurvITE estimates individual treatment effects directly from right-censored survival data by learning balanced representations and optimizing a survival-specific loss.

For survival meta-learners, we implement three types of meta-learning frameworks that have been extended to handle censored data directly: S-Learner, T-Learner, and matching-learner (Norooz-izadeh et al., 2025). Each of these frameworks is combined with three different base survival models (Random Survival Forests (Ishwaran et al., 2008), DeepSurv (Katzman et al., 2018), and DeepHit (Lee et al., 2018)) that estimate the underlying survival functions, resulting in $3 \times 3 = 9$ survival meta-learner variants.

In total, our benchmark evaluates $42 + 2 + 9 = 53$ different method configurations across the 40 synthetic datasets and the two real-world datasets described in Section 3.

Table 8: Breakdown of benchmarked survival-CATE estimator variants used in our experiments. Each row corresponds to a specific combination of method class, imputation strategy (if applicable), base learner(s), and CATE learner(s). Cells with numbers in parentheses indicate how many variants are contributed by the method(s) listed in that cell. The final column reports the total number of method variants constructed using that combination.

| Method Class | Imputation (No. options) | Base Learner (No. options) | CATE Learner (No. options) | No. Variants |
|---|---|---|---|---|
| Outcome Imputation Method | Pseudo-obs, Margin, IPCW-T (3) | Lasso Regression, Random Forest, XGBoost (3) | Meta-Learners (S-, T-, X-, DR-) (4) | **36** |
| | | — | Causal Forest (1) | **3** |
| | | — | Double-ML (1) | **3** |
| Direct-Survival CATE Models | — | — | Causal Survival Forests (1) SurvITE (1) | **2** |
| Survival Meta-Learners | — | Random Survival Forests, DeepSurv, DeepHit (3) | Survival Meta-Learners (S-, T-, Matching-) (3) | **9** |
| **Total** | | | | **53** |

# D  DETAILED OVERVIEW OF CAUSAL INFERENCE METHODS

This section provides a comprehensive explanation of the causal inference methods evaluated in our benchmark. We begin with outcome imputation methods that transform censored survival data into standard regression problems, followed by direct-survival CATE models specifically designed for right-censored data, and finally survival meta-learners that adapt standard meta-learner frameworks to handle censoring. For each method, we present the theoretical foundation, algorithmic procedure, and specific implementation considerations in the survival analysis context. Our exposition focuses on highlighting the unique characteristics that make each approach suitable for different survival and causal inference scenarios, with particular attention to how these methods handle the challenges posed by censoring and treatment effect heterogeneity.

## D.1  OUTCOME IMPUTATION METHODS

Meta-learners represent a flexible framework for estimating conditional average treatment effects (CATEs) by decomposing the causal inference problem into standard supervised learning tasks. The key advantage of meta-learners is that they allow practitioners to leverage any out-of-the-box machine learning algorithm as a "base learner" while maintaining principled approaches to causal effect estimation. This modularity makes meta-learners particularly attractive in practice, as they can incorporate state-of-the-art ML methods (e.g., random forests, gradient boosting, neural networks) without requiring specialized causal inference implementations. For detailed explanations on meta-learners, one can refer to Künzel et al. (2019); Kennedy (2023). We provide a simplified overview below and largely refer to the documentation of the `econml` package.

**T-Learner (Künzel et al., 2019).** The T-Learner (Two-Learner) adopts the most straightforward approach by fitting separate outcome models for treated and control groups. Given binary treatment $W \in \{0, 1\}$, features $X$, and outcome $Y$, the T-Learner:

1. Splits the data by treatment assignment: $(X^0, Y^0)$ for controls and $(X^1, Y^1)$ for treated units
2. Trains separate outcome models (i.e. predicting the outcome $Y$ using features $X$):

$$\text{For control units: } \hat{\mu}_0 = M_0(Y^0 \sim X^0)$$

$$\text{For treated units: } \hat{\mu}_1 = M_1(Y^1 \sim X^1)$$

3. Estimates CATE as:

$$\hat{\tau}(x) = \hat{\mu}_1(x) - \hat{\mu}_0(x)$$

where $M_0$ and $M_1$ can be any regression algorithm. The T-Learner is conceptually simple but can suffer from high variance when treatment groups have different sizes or when the outcome models extrapolate poorly to regions with limited overlap.

**S-Learner (Künzel et al., 2019).** The S-Learner (Single-Learner) takes a unified modeling approach by including treatment assignment as an additional feature. The procedure involves:

1. Training a single model using all available data:

$$\hat{\mu} = M(Y \sim (X, W))$$

2. Estimating CATE as:

$$\hat{\tau}(x) = \hat{\mu}(x, 1) - \hat{\mu}(x, 0)$$

This approach leverages all available data for training and can be more sample-efficient than the T-Learner. However, it relies heavily on the base learner's ability to capture treatment-feature interactions, and may perform poorly when these interactions are complex or when the treatment effect is small relative to the baseline outcome.

**X-Learner (Künzel et al., 2019).** The X-Learner represents a more sophisticated approach that combines ideas from both T-Learner and inverse propensity weighting. The algorithm proceeds in multiple stages:

1. Fit initial outcome models:

$$\hat{\mu}_0 = M_1(Y^0 \sim X^0)$$
$$\hat{\mu}_1 = M_2(Y^1 \sim X^1)$$

2. Compute imputed treatment effects:

$$\text{For treated units: } \hat{D}^1 = Y^1 - \hat{\mu}_0(X^1)$$
$$\text{For control units: } \hat{D}^0 = \hat{\mu}_1(X^0) - Y^0$$

3. Model treatment effects:

$$\hat{\tau}_0 = M_3(\hat{D}^0 \sim X^0)$$
$$\hat{\tau}_1 = M_4(\hat{D}^1 \sim X^1)$$

4. Combine estimates using propensity scores:

$$\hat{\tau}(x) = g(x)\hat{\tau}_0(x) + (1 - g(x))\hat{\tau}_1(x)$$

where $g(x)$ is the estimation for propensity score $P(W = 1|X = x)$ and is typically fitted using logistic regression. The X-Learner is particularly effective when treatment groups have different sizes or when treatment effects are heterogeneous, as it explicitly models treatment effect variation and uses propensity weighting for optimal combination.

**DR-Learner (Kennedy, 2023).** The DR-Learner (Doubly Robust Learner) extends the doubly robust framework to meta-learning by combining outcome modeling with propensity score estimation. The approach constructs doubly robust scores that remain consistent if either the outcome model or propensity model is correctly specified. It includes the following steps:

1. Fit outcome modeling for each treatment

$$\hat{\mu}_0 = M_1(Y^0 \sim X^0)$$
$$\hat{\mu}_1 = M_2(Y^1 \sim X^1)$$

2. Construct propensity score modeling

$$\hat{g} = M_g(W \sim X)$$

3. Construct doubly robust outcomes:

$$\hat{Y}_0^{DR} = \hat{\mu}_0(X) + \frac{(Y - \hat{\mu}_0(X))}{\hat{g}(X)} \cdot \mathbb{1}\{W = 0\}$$
$$\hat{Y}_1^{DR} = \hat{\mu}_1(X) + \frac{(Y - \hat{\mu}_1(X))}{\hat{g}(X)} \cdot \mathbb{1}\{W = 1\}$$

4. Final CATE estimation: $\hat{\tau}(x) = \hat{Y}_1^{DR} - \hat{Y}_0^{DR}$

The DR-Learner provides theoretical robustness guarantees and often performs well in practice, particularly when either outcome or treatment assignment can be accurately modeled.

**Double-ML (Chernozhukov et al., 2018).** Double Machine Learning (Double-ML or DML) represents a principled framework for estimating heterogeneous treatment effects when confounders are high-dimensional or when their relationships with treatment and outcome cannot be adequately captured by parametric models. The key insight of DML is to decompose the causal inference problem into two predictive tasks that can be solved using arbitrary machine learning algorithms while maintaining favorable statistical properties. Specifically, DML assumes the following structural relationships:

- $Y = \theta(X) \cdot W + g(X, Z) + \epsilon \quad \text{with} \quad \mathbb{E}[\epsilon|X, Z] = 0$
- $W = f(X, Z) + \eta \quad \text{with} \quad \mathbb{E}[\eta|X, Z] = 0$
- $\mathbb{E}[\eta \cdot \epsilon|X, Z] = 0$

where $Y$ is the outcome, $W$ is the treatment, $X$ are the features of interest for heterogeneity, $Z$ are confounding variables, and $\theta(X)$ is the conditional average treatment effect we aim to estimate. The method proceeds by first estimating two nuisance functions:

- Outcome regression: $q(X, Z) = \mathbb{E}[Y|X, Z]$
- Treatment regression: $f(X, Z) = \mathbb{E}[W|X, Z]$

These nuisance functions can be estimated using any machine learning algorithm capable of regression (for continuous treatments) or classification (for binary treatments). Popular choices include random forests, gradient boosting, neural networks, or regularized linear models. After obtaining estimates $\hat{q}$ and $\hat{f}$, DML constructs residualized outcomes and treatments:

$$\tilde{Y} = Y - \hat{q}(X, Z)$$
$$\tilde{W} = W - \hat{f}(X, Z)$$

The final step estimates $\theta(X)$ by regressing $\tilde{Y}$ on $\tilde{W}$ and $X$

$$\hat{\theta} = \arg \min_{\theta} \mathbb{E}_n[(\tilde{Y} - \theta(X) \cdot \tilde{W})^2]$$

**Causal Forest (Athey et al., 2019).** Causal Forest extends the random forest methodology to directly estimate heterogeneous treatment effects in a non-parametric, data-adaptive manner. Unlike meta-learners that rely on global models, Causal Forest estimates treatment effects locally by learning similarity metrics in the feature space and weighting observations accordingly. Causal Forest builds upon the same structural assumptions as DML but estimates $\theta(x)$ locally for each target point $x$. The method constructs a forest where each tree is grown using a **causal splitting criterion** that maximizes treatment effect heterogeneity rather than prediction accuracy. For a target point $x$, the treatment effect is estimated by solving:

$$\hat{\theta}(x) = \arg \min_{\theta} \sum_{i=1}^{n} K_x(X_i) \cdot (\tilde{Y}_i - \theta \cdot \tilde{W}_i)^2$$

where $K_x(X_i)$ represents the similarity between points $x$ and $X_i$ as determined by how frequently they fall in the same leaf across the forest, and $\tilde{Y}$, $\tilde{W}$ are residuals from nuisance function estimates.

**Implementation in survival context** In our benchmark, meta-learners, Double-ML, and Causal Forest are applied to survival outcomes through outcome imputation methods. We first apply imputation techniques (Pseudo-obs, Margin, or IPCW-T, see Appendix B for details) to convert censored survival times into continuous outcomes, then apply the meta-learners described above with various base regression algorithms (Lasso Regression, Random Forest, XGBoost). This two-stage approach allows leveraging the rich ecosystem of causal inference methods developed for continuous outcomes while handling the complexities of censored data.

## D.2 DIRECT-SURVIVAL CATE METHODS

**Causal Survival Forests (CSF) (Cui et al., 2023)** extends the Causal Forest methodology directly to right-censored survival data by incorporating doubly robust estimating equations from survival analysis. Unlike meta-learners that require outcome imputation, CSF handles censored observations natively while maintaining the adaptive partitioning advantages of tree-based methods. CSF builds upon the Causal Forest framework of Athey et al. (2019) but adapts the splitting criterion and estimation procedure for survival outcomes. For a detailed explanation of the method, please refer to the original paper by Cui et al. (2023). We provide an overview of the estimation procedures as follows:

1. **Nuisance estimation**: Using cross-fitting, estimate nuisance components including:
   - Propensity scores: $\hat{e}(x) = P(W = 1 | X = x)$
   - Outcome regression: $\hat{m}(x) = E[y(T) | X = x]$
   - Censoring survival function: $\hat{S}_w^C(s|x) = P(C \geq s | W = w, X = x)$
   - Conditional expectations: $\hat{Q}_w(s|x) = E[y(T) | X = x, W = w, T \wedge h > s]$

   where $y(T)$ is a transformation applied on the event time $T$, the same as defined in Eq.1.
2. **Forest construction**: Build a forest where each tree uses a causal splitting criterion that maximizes treatment effect heterogeneity. The splitting rule targets variation in the doubly robust scores rather than prediction accuracy.
3. **Local estimation**: For a target point $x$, compute forest weights $\alpha(x)$ indicating similarity based on leaf co-occurrence across trees, then estimate the CATE by solving:

$$\sum \alpha(x) \psi_{\hat{\tau}(x)}(X, y(U), U \wedge h, W, \Delta^h; \hat{e}, \hat{m}, \hat{S}_w^C, \hat{Q}_w) = 0$$

where $\psi$ is the doubly robust score function that adjusts for both treatment assignment and censoring.

**SurvITE (Curth et al., 2021a)** adapts the representation learning paradigm for counterfactual inference to time-to-event data. Unlike methods that rely on local similarity in the covariate space, SurvITE addresses selection bias by learning a shared latent representation where the treated and control distributions are balanced, while simultaneously modeling the censoring mechanism. SurvITE builds upon the theoretical bounds of counterfactual regression but incorporates survival-specific loss functions to handle right-censored outcomes without requiring imputation. A brief outline of the method follows:

1. **Representation learning**: Map covariates $X$ to a latent representation $\Phi(X)$ via a deep neural network, subject to a discrepancy penalty. The objective is to minimize an Integral Probability Metric (IPM) (e.g., Wasserstein distance or MMD) between the treated and control populations in the latent space:

$$\mathrm{IPM}(P_\Phi(X|W=1), P_\Phi(X|W=0)) < \epsilon$$

2. **Factual loss minimization**: Simultaneously train treatment-specific hypothesis heads ($h_1$ and $h_0$) on top of $\Phi(X)$ using a survival loss function $\mathcal{L}_{surv}$ (discrete-time log-likelihood) that accounts for censoring:

$$\min_{\Phi, h_0, h_1} \sum_{i=1}^{N} w_i \mathcal{L}_{surv}(h_{W_i}(\Phi(x_i)), T_i, \Delta_i) + \alpha \cdot \mathrm{IPM}$$

3. **Effect estimation**: For a target point $x$, the CATE is estimated by passing $x$ through the learned representation and computing the difference between the outputs of the treatment and control heads:

$$\hat{\tau}(x) = E[y(T)|\Phi(x), W=1] - E[y(T)|\Phi(x), W=0]$$

where the expectation is derived from the predicted survival curves or time-to-event distributions output by $h_1$ and $h_0$.

### D.3 SURVIVAL META-LEARNERS

**T-Learner-Survival (Bo et al., 2024; Noroozizadeh et al., 2025).** The T-Learner can be adapted to right-censored survival data by fitting separate survival models for each treatment group. Let $W \in \{0, 1\}$ denote the treatment indicator, $X$ be the covariate vector, and $T$ the observed survival time with censoring indicator $\delta$, and $h$ the maximum follow-up time.

1. **Split data by treatment:** Partition the dataset into $(X^0, T^0, \delta^0)$ for $W = 0$ and $(X^1, T^1, \delta^1)$ for $W = 1$.
2. **Train separate survival models:** Fit a survival model (e.g., Random Survival Forests, Deep-Surv, DeepHit) to each group:

$$\widehat{S}_0(u|x) = \text{Survival model fitted on } (X^0, T^0, \delta^0)$$
$$\widehat{S}_1(u|x) = \text{Survival model fitted on } (X^1, T^1, \delta^1)$$

3. **Estimate restricted mean survival time (RMST):** Compute RMST for each treatment as:

$$\widehat{\mu}_0(x) = \int_0^h \widehat{S}_0(u|x)du, \quad \widehat{\mu}_1(x) = \int_0^h \widehat{S}_1(u|x)du$$

4. **Estimate CATE:** For any $x$, estimate treatment effect:

$$\widehat{\tau}_{\text{T-Learner}}(x) = \widehat{\mu}_1(x) - \widehat{\mu}_0(x)$$

**S-Learner-Survival (Bo et al., 2024; Noroozizadeh et al., 2025).** The S-Learner adapts by training a single survival model over all data with treatment as a covariate.

1. **Fit survival model:** Train a survival model over the full dataset using $(X, W)$ as inputs:

$$\widehat{S}(u|x, w) = \text{Survival model fitted on } ((X, W), T, \delta)$$

2. **Estimate restricted mean survival time (RMST):** Compute RMST under both treatment conditions:

$$\widehat{\mu}(x,0) = \int_0^h \widehat{S}(u|x,0)du, \quad \widehat{\mu}(x,1) = \int_0^h \widehat{S}(u|x,1)du$$

3. **Estimate CATE:**

$$\widehat{\tau}_{\text{S-Learner}}(x) = \widehat{\mu}(x,1) - \widehat{\mu}(x,0)$$

**Matching-Survival (Noroozizadeh et al., 2025).** The Matching-Learner estimates the CATE by imputing the counterfactual Restricted Mean Survival Time (RMST) using matched data points from the opposite treatment group.

1. **Estimate factual RMST:** Fit a survival model on the full dataset and compute:

$$\widehat{\mu}_{W_i}(X_i) = \int_0^h \widehat{S}(u|X_i, W_i)du$$

2. **Find matches:** For each individual $i$, identify $K$ nearest neighbors $J_K(i)$ from the opposite treatment group $(1 - W_i)$.

3. **Estimate counterfactual RMST:** Average factual RMSTs of matched neighbors:

$$\widehat{\mu}_{1-W_i}(X_i) = \frac{1}{K} \sum_{j \in J_K(i)} \widehat{\mu}_{W_j}(X_j)$$

4. **Estimate CATE:** Compute CATE for each unit:

$$\widehat{\tau}_{\text{matching}}(X_i) = \left(\widehat{\mu}_{W_i}(X_i) - \widehat{\mu}_{1-W_i}(X_i)\right) \cdot \left(2W_i - 1\right)$$

This approach makes minimal modeling assumptions beyond nearest-neighbor similarity and is particularly helpful in settings with low overlap or where global models may be misspecified.

# E    MODEL TRAINING DETAILS AND HYPERPARAMETERS ON BENCHMARKING WITH SYNTHETIC DATA

To rigorously evaluate and compare the performance of causal inference models under controlled conditions, we conducted extensive benchmarking on synthetic datasets. Each synthetic dataset consisted of 50,000 samples generated under known data-generating processes explained in Appendix A. For each experimental repeat, we selected a subset of 5,000 samples for training, 2,500 for validation, and 2,500 for testing, using 10 distinct random seeds (experimental repeats) to ensure robustness. Hyperparameters for each model were tuned on the validation set to minimize the Conditional Average Treatment Effect Root Mean Squared Error (CATE-RMSE). Throughout this paper, final results are always reported on the held-out test set using the best-performing configuration. Appendix F.7 provides complementary experiments that analyze the convergence behavior of each method under varying training set sizes.

This appendix details the hyperparameter grids used for model selection, the specific survival and outcome models applied within each causal inference framework, and the average computational cost associated with each method class.

## E.1    HYPERPARAMETERS FOR OUTCOME IMPUTATION METHODS

For methods based on outcome imputation, we employed standard regressors to estimate the conditional mean of the survival outcome given covariates and treatment assignment. We considered Lasso regression, Random Forest, and XGBoost as base models. Each was optimized using cross-validated grid search on the training set. The corresponding hyperparameter grids are listed in Table 9.

Table 9: Set of hyperparameters for outcome imputation methods

| Regressor | Hyperparameter Grid |
|---|---|
| Lasso | Alpha: {0.001, 0.01, 0.1, 1, 10} |
| Random Forest | Number of trees: {50, 100}
Maximum depth: {3, 5, None} |
| XGBoost | Number of trees: {50, 100}
Learning rate: {0.01, 0.1}
Maximum depth: {3, 5} |

### E.2 HYPERPARAMETERS FOR DIRECT-SURVIVAL CATE METHODS

For direct modeling of survival outcomes, we employed the Causal Survival Forests (CSF), which adapts the Causal Forest framework to handle right-censored data. We used the default hyperparameters from the original implementation. These are summarized in Table 10.

Table 10: Set of hyperparameters for Causal Survival Forests

| Parameter | Default Value |
| --- | --- |
| Number of trees grown | 2000 |
| Fraction of data per tree | 0.5 |
| Variables tried per split | $\min(\lceil \sqrt{p} + 20 \rceil, p)$ |
| Minimum samples in a leaf | 5 |
| Maximum imbalance of splits | 0.05 |
| Penalty for imbalance at split | 0 |
| Account for treatment and censoring in split stability | TRUE |
| Trees per subsample for confidence intervals | 2 |

For SurvITE, we implemented a PyTorch version based on the original architecture and repository. The main configuration is summarized in Table 11.

Table 11: Set of hyperparameters SurvITE.

| Parameter | Value |
| --- | --- |
| Model width $z_{\text{dim}}, h_{\text{dim1}}, h_{\text{dim2}}$ | $\{8, 16, 32\}$ (synthetic, ACTG) or $\{16, 32, 64\}$ (MIMIC, Twin) |
| Number of shared layers | 2 |
| Number of head layers | 2 |
| Activation function | ReLU |
| Dropout rate | 0.3 |
| IPM type | Wasserstein |
| IPM regularization weight $\beta$ | $10^{-3}$ |
| Smoothing parameter $\gamma$ | 0 |
| Learning rate | $10^{-3}$ |
| Batch size | 256 |
| Maximum epochs | 5000 |
| Early stopping | Checked every 100 epochs; stop after 10 non-improving checks |

### E.3 HYPERPARAMETERS FOR SURVIVAL META-LEARNERS

For survival meta-learners–specifically T-Learner-Survival, S-Learner-Survival, and Matching-Learner-Survival–we used three different base survival models: Random Survival Forests (RSF), DeepSurv, and DeepHit. Each of these models was tuned using a predefined hyperparameter grid, listed in Table 12.

Table 12: Set of hyperparameters for Survival Meta-Learners

| Model | Hyperparameter | Values |
|---|---|---|
| RSF | Number of estimators | {100, 250, 500} |
| | Minimum samples per split | {5, 10, 20} |
| | Minimum samples per leaf | {2, 5, 10} |
| DeepHit | Number of nodes per layer | {32, 64, 128, 256} |
| | Batch normalization | {True, False} |
| | Dropout rate | {0.0, 0.1, 0.2, 0.3} |
| | Learning rate | {0.001, 0.01, 0.05} |
| | Batch size | {128, 256, 512} |
| | Epochs | {200, 512, 1000} |
| | Alpha | {0.1, 0.2, 0.3, 0.5} |
| | Sigma | {0.05, 0.1, 0.2, 0.3} |
| DeepSurv | Number of nodes per layer | {32, 64, 128, 256} |
| | Batch normalization | {True, False} |
| | Dropout rate | {0.0, 0.1, 0.2, 0.3} |
| | Learning rate | {0.001, 0.01, 0.05} |
| | Batch size | {128, 256, 512} |
| | Epochs | {200, 512, 1000} |

Hyperparameters were selected through empirical tuning informed by prior literature. For neural network-based models (DeepSurv, DeepHit), we used early stopping to mitigate overfitting. All experiments were made reproducible by setting random seeds. The best-performing hyperparameter configuration was selected using CATE-RMSE on the validation set, and all final results were obtained on the test set using these optimal configurations.

### E.4 COMPUTATION TIME OF SURVIVAL CATE METHODS

We also measured the computational cost of each CATE estimation method in terms of average runtime per dataset and experimental repeat. Each runtime was recorded using Python's `time.time()` and averaged across 40 synthetic datasets and 10 random seeds. Table 13 presents the mean runtime (in seconds) and standard deviation (excluding the time required for imputation). As expected, neural network-based survival models incur substantially higher computational costs than classical or tree-based methods. All experiments were conducted on a machine equipped with an AMD Ryzen 9 5900X CPU, 128GB RAM, and an NVIDIA GeForce RTX 4090 GPU (CUDA version 12.2).

Table 13: Average computation time per dataset per experimental repeat for each causal method. Runtime is reported in seconds with standard deviation across runs.

| Method Class | Method | Runtime (s) |
|---|---|---|
| Outcome Imputation Methods: Meta-learners | T-Learner | $2.14 \pm 1.38$ |
| | S-Learner | $1.84 \pm 1.22$ |
| | X-Learner | $2.92 \pm 2.42$ |
| | DR-Learner | $3.34 \pm 1.88$ |
| Outcome Imputation Methods: Forest / ML-based learners | Double-ML | $5.27 \pm 0.40$ |
| | Causal Forest | $5.75 \pm 0.40$ |
| Direct-Survival CATE Methods | Causal Survival Forests | $0.78 \pm 0.06$ |
| | SurvITE | $43.15 \pm 6.85$ |
| Survival Meta-Learners | T-Learner-Survival | $31.31 \pm 16.88$ |
| | S-Learner-Survival | $22.99 \pm 14.23$ |
| | Matching-Survival | $49.40 \pm 23.25$ |

## F    Additional Experimental Results for Synthetic Dataset

This section provides comprehensive experimental results on our synthetic datasets, expanding on the key findings presented in the main text. We begin in Appendix F.1 with a full Borda ranking of all 53 model combinations, summarizing global performance across every causal configuration and survival scenario. In Appendix F.2, we explore how performance varies across different survival scenarios–illustrating the impact of censoring patterns and time-to-event distributions on method rankings. Appendix F.3 then delves into how violations of causal assumptions (treatment randomization, ignorability, positivity, and censoring mechanisms) reshape the ranking of models for effectiveness of each causal method.

Subsequent sections (F.4 and F.5) present detailed performance metrics—CATE RMSE and ATE bias, respectively—across all 8 causal configurations and 5 survival scenarios, with box plots capturing variability over 10 experiment repetitions. We also evaluate auxiliary components in F.6, including imputation methods and base learners (regression, survival, and propensity models), and in Appendix F.7 we examine convergence behavior under varying training set sizes. Together, these detailed results support the robustness, sensitivity, and practical trade-offs of each model family in a wide spectrum of data-generating and causal settings.

In addition to average-rank summaries, in Appendix F.1– F.3, we also report a set of win-rate analyses that track how often each method family attains Top-1, Top-3, and Top-5 performance on both CATE RMSE and ATE Bias. Overall win-rates across all survival scenarios and causal configurations are summarized in Table 15, while Tables 16, 17, and 18 provide scenario-specific and causal-configuration-specific win-rates. These complementary views highlight not only which methods achieve strong average performance, but also which ones most consistently appear among the top performers across varying censoring regimes, survival experimental conditions, and patterns of causal assumption violations.

### F.1    Full ranking of models

To compare the overall performance of the methods across all synthetic datasets, we computed a Borda ranking based on the average rank of each method's test set CATE RMSE (Table 14). The ranking procedure aggregates method performance across all combinations of causal configurations and survival scenarios. For each method, we first computed its RMSE on the test subset of the CATE predictions for each (causal configuration, survival scenario) pair. We then ranked all 53 methods (described in Appendix C) within each pair and calculated the average rank across these conditions. This average rank represents the method's Borda score and serves as a unified summary of its performance robustness in our synthetic data experiments.

In addition to the Borda rankings, we also summarize how often each method family achieves leading performance across all experimental settings by reporting the percentage of times a method appears in the Top-1, Top-3, and Top-5 for both CATE RMSE and ATE Bias. This provides a complementary view that focuses on frequency of strong performance rather than average rank, and helps separate methods that occasionally perform well from those that do so consistently across our full set of survival scenarios and causal configurations.

Overall, Table 15 shows that the strongest performance comes from method families that explicitly incorporate survival structure, particularly the survival meta-learners. Matching-Survival has the most consistent high-rank presence on CATE RMSE (Top-5: 92.5%), while S-Learner-Survival achieves the highest Top-3 and Top-5 rates on CATE RMSE (67.5% and 85.0%, respectively) and also performs strongly on ATE Bias (Top-3: 57.5%, Top-5: 77.5%). Across direct-survival CATE methods, Causal Survival Forests remains competitive and relatively stable, with the highest Top-1 rate on CATE RMSE among direct-survival CATE methods (25.0%) and strong ATE Bias coverage (Top-5: 75.0%). Double-ML performs well primarily on CATE RMSE (Top-3: 52.5%, Top-5: 72.5%) but is less competitive on ATE Bias. In contrast, the classical outcome-imputation meta-learners (T-, S-, X-, and DR-Learners) attain Top-1 positions only rarely and generally have lower Top-3/Top-5 rates than the survival-aware families, highlighting the advantage of accounting for time-to-event structure when estimating heterogeneous treatment effects in our wide range of experimental scenarios.

Table 14: Borda ranking of all methods

| Rank | Method | Score | Rank | Method | Score |
|---|---|---|---|---|---|
| 1 | (S-Learner-Survival, DeepSurv) | 5.18 | 28 | (Causal Forest, Pseudo-Obs) | 24.70 |
| 2 | (Matching-Survival, DeepSurv) | 5.43 | 29 | (T-Learner, Margin, RandomForest) | 26.23 |
| 3 | (Double-ML, Margin) | 6.65 | 30 | (S-Learner, IPCW-T, RandomForest) | 26.50 |
| 4 | (Causal Forest, Margin) | 10.78 | 31 | (SurvITE) | 27.28 |
| 5 | (Double-ML, IPCW-T) | 11.78 | 32 | (S-Learner, Margin, Lasso) | 27.98 |
| 6 | (Causal Survival Forests) | 12.53 | 33 | (S-Learner, IPCW-T, Lasso) | 27.98 |
| 7 | (Double-ML, Pseudo-Obs) | 15.38 | 34 | (S-Learner, Pseudo-Obs, Lasso) | 28.18 |
| 8 | (S-Learner-Survival, RSF) | 15.50 | 35 | (T-Learner, IPCW-T, RandomForest) | 29.85 |
| 9 | (Causal Forest, IPCW-T) | 15.93 | 36 | (T-Learner-Survival, DeepHit) | 30.30 |
| 10 | (X-Learner, Margin, RandomForest) | 16.53 | 37 | (T-Learner-Survival, RSF) | 30.83 |
| 11 | (S-Learner, Margin, XGB) | 18.33 | 38 | (S-Learner, Pseudo-Obs, XGB) | 32.80 |
| 12 | (Matching-Survival, DeepHit) | 18.50 | 39 | (S-Learner, Pseudo-Obs, RandomForest) | 32.85 |
| 13 | (DR-Learner, Margin, Lasso) | 18.83 | 40 | (X-Learner, Margin, XGB) | 34.00 |
| 14 | (T-Learner-Survival, DeepSurv) | 19.60 | 41 | (X-Learner, Pseudo-Obs, RandomForest) | 35.33 |
| 15 | (S-Learner-Survival, DeepHit) | 19.78 | 42 | (X-Learner, IPCW-T, XGB) | 35.75 |
| 16 | (X-Learner, Margin, Lasso) | 20.53 | 43 | (DR-Learner, Margin, RandomForest) | 37.45 |
| 17 | (T-Learner, Margin, Lasso) | 20.53 | 44 | (DR-Learner, IPCW-T, RandomForest) | 38.70 |
| 18 | (DR-Learner, Pseudo-Obs, Lasso) | 20.93 | 45 | (T-Learner, Margin, XGB) | 41.18 |
| 19 | (S-Learner, IPCW-T, XGB) | 21.85 | 46 | (T-Learner, IPCW-T, XGB) | 42.08 |
| 20 | (X-Learner, IPCW-T, RandomForest) | 22.05 | 47 | (T-Learner, Pseudo-Obs, RandomForest) | 42.50 |
| 21 | (Matching-Survival, RSF) | 22.35 | 48 | (DR-Learner, IPCW-T, XGB) | 46.63 |
| 22 | (DR-Learner, IPCW-T, Lasso) | 22.65 | 49 | (DR-Learner, Margin, XGB) | 46.70 |
| 23 | (X-Learner, Pseudo-Obs, Lasso) | 22.85 | 50 | (X-Learner, Pseudo-Obs, XGB) | 47.35 |
| 24 | (T-Learner, Pseudo-Obs, Lasso) | 22.90 | 51 | (DR-Learner, Pseudo-Obs, RandomForest) | 49.60 |
| 25 | (S-Learner, Margin, RandomForest) | 23.05 | 52 | (T-Learner, Pseudo-Obs, XGB) | 50.55 |
| 26 | (T-Learner, IPCW-T, Lasso) | 24.40 | 53 | (DR-Learner, Pseudo-Obs, XGB) | 52.80 |
| 27 | (X-Learner, IPCW-T, Lasso) | 24.45 | | | |

Table 15: Win-rate (%) of method families across all experimental configurations. Values denote the percentage of times a method appears in the Top-1, Top-3, and Top-5 according to CATE RMSE and ATE Bias.

| Method Family | CATE RMSE | | | ATE Bias | | |
|---|---|---|---|---|---|---|
| | Top-1 | Top-3 | Top-5 | Top-1 | Top-3 | Top-5 |
| *Outcome Imputation Methods* | | | | | | |
| T-Learner | 2.5 | 12.5 | 30.0 | 2.5 | 15.0 | 32.5 |
| S-Learner | 0 | 2.5 | 2.5 | 5.0 | 15.0 | 25.0 |
| X-Learner | 2.5 | 12.5 | 35.0 | 5.0 | 17.5 | 40.0 |
| DR-Learner | 0 | 7.5 | 22.5 | 5.0 | 12.5 | 30.0 |
| Double-ML | 20.0 | 52.5 | 72.5 | 2.5 | 5.0 | 17.5 |
| Causal Forest | 7.5 | 17.5 | 42.5 | 2.5 | 7.5 | 27.5 |
| *Direct-Survival CATE Methods* | | | | | | |
| Causal Survival Forests | 25.0 | 45.0 | 52.5 | 17.5 | 52.5 | 75.0 |
| SurvITE | 2.5 | 5.0 | 22.5 | 7.5 | 22.5 | 40.0 |
| *Survival Meta-Learners* | | | | | | |
| T-Learner-Survival | 12.5 | 32.5 | 42.5 | 25.0 | 37.5 | 62.5 |
| S-Learner-Survival | 17.5 | 67.5 | 85.0 | 12.5 | 57.5 | 77.5 |
| Matching-Survival | 12.5 | 50.0 | 92.5 | 17.5 | 57.5 | 72.5 |

## F.2  RANKING OF CAUSAL METHODS FOR DIFFERENT SURVIVAL SCENARIOS

In Figure 6, we present the Borda ranking of all causal model families across five different survival scenarios (A–E). For each scenario, the average rank of each method is computed over 8 distinct causal configurations, allowing us to assess robustness across varying underlying data-generating processes. The horizontal layout of each plot ranks methods from best (left, top to bottom) to worst (right, bottom to top), with rank values annotated next to each method for clarity. The colors of the lines connecting the methods to the horizontal axis represent the specific family of the method (e.g., outcome imputation, direct-survival CATE, survival meta-learners). Additionally, thick black horizontal bands connect methods whose difference in ranking is not statistically significant.

These plots illustrate how model performance shifts as censoring rates and survival distributions vary. In Scenario A, which involves minimal censoring, Double-ML achieves the best overall ranking (1.5), though survival meta-learner approaches like T-Learner-Survival and S-Learner-Survival also perform highly, sharing statistical overlap with the top spot. However, as we move toward Scenarios C, D, and E—which are characterized by higher censoring—direct survival modeling approaches consistently rise to the top. Specifically, S-Learner-Survival, Matching-Survival, and Causal Survival Forests dominate the highest ranks across these later scenarios, although SurvITE does not show a really strong average performance even with more censoring. Conversely, the relative performance of Double-ML steadily declines, eventually dropping to the bottom half of average ranking (rank 6.9) in Scenario E. This pattern reinforces that survival-specific modeling is better equipped to handle the uncertainty introduced by heavy censoring, outperforming standard outcome imputation strategies in such settings.

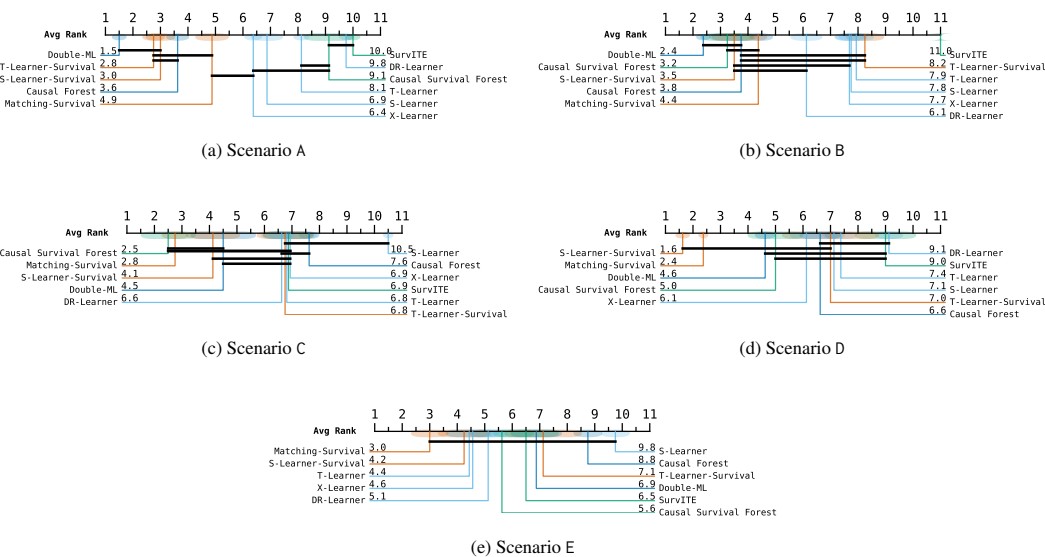

Figure 6: Average ranking of each model for each Survival Scenario. Shaded regions indicate the standard error of the rank across datasets.

In addition to the global rankings in Figure 6, we report scenario-specific win-rates in Table 16. For each survival scenario (A–E), we compute how often each method family appears in the Top-1, Top-3, and Top-5 positions for CATE RMSE and ATE Bias across the eight causal configurations. This provides a complementary view of robustness, highlighting which families consistently occupy the top ranks as we vary the survival time model (Cox, AFT, Poisson) and the censoring rate (low, medium, high; Table 2).

Under low censoring (Scenarios A and B), outcome-imputation approaches can still be competitive on CATE RMSE, but the winners are scenario-dependent and survival-aware methods remain highly prevalent in the Top-$k$ sets. In Scenario A (Cox, low censoring), Double-ML dominates CATE RMSE (62.5% Top-1 and 100% Top-3/Top-5), while survival meta-learners also appear frequently in Top-3/Top-5 (e.g., T-Learner-Survival and S-Learner-Survival are both 100% Top-5). ATE Bias is more dispersed: S-Learner and DR-Learner each attain 25.0% Top-1, and survival meta-learners

(S-Learner-Survival and Matching-Survival) contribute strongly to Top-3/Top-5, whereas the direct-survival CATE methods have limited presence. In Scenario B (AFT, low censoring), Causal Survival Forests becomes a leading family for both metrics (37.5% Top-1 CATE RMSE; 25.0% Top-1 ATE Bias; 75.0% Top-3/Top-5 ATE Bias). At the same time, Causal Forest remains competitive on CATE RMSE (37.5% Top-1), and survival meta-learners retain substantial Top-5 coverage, particularly Matching-Survival (87.5% Top-5 CATE RMSE) and S-Learner-Survival (87.5% Top-5 CATE RMSE). Notably, SurvITE does not appear among the Top-$k$ in this scenario.

As censoring increases, survival-specific modeling becomes increasingly important and accounts for a larger fraction of the best-performing sets. In Scenario C (Poisson, medium censoring), Causal Survival Forests clearly leads on both metrics (62.5% Top-1 CATE RMSE; 37.5% Top-1 ATE Bias; 100% Top-5 ATE Bias), while Matching-Survival and S-Learner-Survival frequently appear among the best methods (e.g., Matching-Survival is 25.0% Top-1 CATE RMSE and 100% Top-5 CATE RMSE; both Matching-Survival and S-Learner-Survival reach 75.0% Top-3 CATE RMSE). Under high censoring, the dominance shifts even more strongly toward survival-aware families, but with different emphases across scenarios. In Scenario D (AFT, high censoring), S-Learner-Survival is the primary winner on CATE RMSE (50.0% Top-1 and 100% Top-3/Top-5), while Matching-Survival achieves perfect Top-3/Top-5 coverage (100% in both cases). For ATE Bias, the top positions are shared across multiple survival-aware families (Causal Survival Forests, T-Learner-Survival, and S-Learner-Survival each at 25.0% Top-1). In Scenario E (Poisson, high censoring), the strongest ATE Bias performance is achieved by T-Learner-Survival (50.0% Top-1 and 87.5% Top-5), with Matching-Survival and SurvITE following, whereas CATE RMSE Top-1 is more distributed (Matching-Survival: 37.5% Top-1; several other methods across both outcome-imputation and direct-survival CATE methods at 12.5%). Overall, across all scenarios, classical outcome-imputation methods are rarely dominant or ranked Top-1 under moderate or high censoring, reinforcing the benefit of explicitly modeling time-to-event structure when the censoring rate is high.

Table 16: Win-rate (%) of method families by Survival Scenario. Values denote the percentage of times a method appears in the Top-1, Top-3, and Top-5 according to CATE RMSE and ATE Bias across the eight causal configurations for each scenario.

### Scenario A: Cox, Low Censoring

| Method Family | CATE RMSE | | | ATE Bias | | |
|---|---|---|---|---|---|---|
| | Top-1 | Top-3 | Top-5 | Top-1 | Top-3 | Top-5 |
| *Outcome Imputation Methods* | | | | | | |
| T-Learner | 0 | 0 | 0 | 0 | 25.0 | 25.0 |
| S-Learner | 0 | 12.5 | 12.5 | 25.0 | 50.0 | 50.0 |
| X-Learner | 0 | 0 | 12.5 | 0 | 25.0 | 62.5 |
| DR-Learner | 0 | 0 | 0 | 25.0 | 25.0 | 62.5 |
| Double-ML | 62.5 | 100.0 | 100.0 | 0 | 12.5 | 37.5 |
| Causal Forest | 0 | 37.5 | 100.0 | 12.5 | 12.5 | 37.5 |
| *Direct-Survival CATE Methods* | | | | | | |
| Causal Survival Forests | 0 | 0 | 0 | 0 | 12.5 | 25.0 |
| SurvITE | 0 | 0 | 0 | 0 | 0 | 25.0 |
| *Survival Meta-Learners* | | | | | | |
| T-Learner-Survival | 12.5 | 87.5 | 100.0 | 12.5 | 25.0 | 50.0 |
| S-Learner-Survival | 25.0 | 50.0 | 100.0 | 12.5 | 62.5 | 75.0 |
| Matching-Survival | 0 | 12.5 | 75.0 | 12.5 | 50.0 | 50.0 |

### Scenario B: AFT, Low Censoring

| Method Family | CATE RMSE | | | ATE Bias | | |
|---|---|---|---|---|---|---|
| | Top-1 | Top-3 | Top-5 | Top-1 | Top-3 | Top-5 |
| *Outcome Imputation Methods* | | | | | | |
| T-Learner | 0 | 0 | 12.5 | 0 | 12.5 | 25.0 |
| S-Learner | 0 | 0 | 0 | 0 | 12.5 | 25.0 |
| X-Learner | 0 | 0 | 12.5 | 12.5 | 12.5 | 37.5 |
| DR-Learner | 0 | 12.5 | 25.0 | 0 | 25.0 | 37.5 |
| Double-ML | 12.5 | 100.0 | 100.0 | 12.5 | 12.5 | 37.5 |
| Causal Forest | 37.5 | 50.0 | 75.0 | 0 | 25.0 | 62.5 |
| *Direct-Survival CATE Methods* | | | | | | |
| Causal Survival Forests | 37.5 | 62.5 | 87.5 | 25.0 | 75.0 | 75.0 |
| SurvITE | 0 | 0 | 0 | 0 | 0 | 0 |
| *Survival Meta-Learners* | | | | | | |
| T-Learner-Survival | 0 | 0 | 12.5 | 25.0 | 37.5 | 62.5 |
| S-Learner-Survival | 12.5 | 62.5 | 87.5 | 12.5 | 37.5 | 62.5 |
| Matching-Survival | 0 | 12.5 | 87.5 | 12.5 | 50.0 | 75.0 |

### Scenario C: Poisson, Medium Censoring

| Method Family | CATE RMSE | | | ATE Bias | | |
|---|---|---|---|---|---|---|
| | Top-1 | Top-3 | Top-5 | Top-1 | Top-3 | Top-5 |
| *Outcome Imputation Methods* | | | | | | |
| T-Learner | 0 | 25.0 | 37.5 | 0 | 12.5 | 25.0 |
| S-Learner | 0 | 0 | 0 | 0 | 0 | 0 |
| X-Learner | 0 | 25.0 | 37.5 | 0 | 0 | 25.0 |
| DR-Learner | 0 | 0 | 25.0 | 0 | 0 | 0 |
| Double-ML | 12.5 | 50.0 | 62.5 | 0 | 0 | 0 |
| Causal Forest | 0 | 0 | 0 | 0 | 0 | 0 |
| *Direct-Survival CATE Methods* | | | | | | |
| Causal Survival Forests | 62.5 | 75.0 | 87.5 | 37.5 | 75.0 | 100.0 |
| SurvITE | 0 | 0 | 50.0 | 25.0 | 50.0 | 87.5 |
| *Survival Meta-Learners* | | | | | | |
| T-Learner-Survival | 0 | 12.5 | 25.0 | 12.5 | 25.0 | 75.0 |
| S-Learner-Survival | 0 | 62.5 | 75.0 | 0 | 62.5 | 100.0 |
| Matching-Survival | 25.0 | 75.0 | 100.0 | 25.0 | 75.0 | 87.5 |

### Scenario D: AFT, High Censoring

| Method Family | CATE RMSE | | | ATE Bias | | |
|---|---|---|---|---|---|---|
| | Top-1 | Top-3 | Top-5 | Top-1 | Top-3 | Top-5 |
| *Outcome Imputation Methods* | | | | | | |
| T-Learner | 0 | 0 | 25.0 | 12.5 | 12.5 | 50.0 |
| S-Learner | 0 | 0 | 0 | 0 | 12.5 | 50.0 |
| X-Learner | 0 | 0 | 37.5 | 12.5 | 12.5 | 25.0 |
| DR-Learner | 0 | 0 | 0 | 0 | 12.5 | 37.5 |
| Double-ML | 0 | 0 | 87.5 | 0 | 0 | 12.5 |
| Causal Forest | 0 | 0 | 37.5 | 0 | 0 | 37.5 |
| *Direct-Survival CATE Methods* | | | | | | |
| Causal Survival Forests | 12.5 | 50.0 | 50.0 | 25.0 | 62.5 | 87.5 |
| SurvITE | 0 | 12.5 | 25.0 | 0 | 12.5 | 12.5 |
| *Survival Meta-Learners* | | | | | | |
| T-Learner-Survival | 37.5 | 37.5 | 37.5 | 25.0 | 25.0 | 37.5 |
| S-Learner-Survival | 50.0 | 100.0 | 100.0 | 25.0 | 75.0 | 75.0 |
| Matching-Survival | 0 | 100.0 | 100.0 | 12.5 | 75.0 | 75.0 |

### Scenario E: Poisson, High Censoring

| Method Family | CATE RMSE | | | ATE Bias | | |
|---|---|---|---|---|---|---|
| | Top-1 | Top-3 | Top-5 | Top-1 | Top-3 | Top-5 |
| *Outcome Imputation Methods* | | | | | | |
| T-Learner | 12.5 | 37.5 | 75.0 | 0 | 12.5 | 37.5 |
| S-Learner | 0 | 0 | 0 | 0 | 0 | 0 |
| X-Learner | 12.5 | 37.5 | 75.0 | 0 | 37.5 | 50.0 |
| DR-Learner | 0 | 25.0 | 62.5 | 0 | 0 | 12.5 |
| Double-ML | 12.5 | 12.5 | 12.5 | 0 | 0 | 0 |
| Causal Forest | 0 | 0 | 0 | 0 | 0 | 0 |
| *Direct-Survival CATE Methods* | | | | | | |
| Causal Survival Forests | 12.5 | 37.5 | 37.5 | 0 | 37.5 | 87.5 |
| SurvITE | 12.5 | 12.5 | 37.5 | 12.5 | 50.0 | 75.0 |
| *Survival Meta-Learners* | | | | | | |
| T-Learner-Survival | 12.5 | 25.0 | 37.5 | 50.0 | 75.0 | 87.5 |
| S-Learner-Survival | 0 | 62.5 | 62.5 | 12.5 | 50.0 | 75.0 |
| Matching-Survival | 37.5 | 50.0 | 100.0 | 25.0 | 37.5 | 75.0 |

F.3    RANKING OF CAUSAL METHODS FOR DIFFERENT CAUSAL CONFIGURATIONS

In Figure 7, we present the Borda ranking of causal model families across eight distinct causal configurations, each representing different combinations of assumptions related to treatment assignment (RCT vs. observational), ignorability, positivity, and censoring mechanisms. Within each configuration, the average rank of each method is computed over all survival scenarios, allowing us to isolate how assumption violations affect model performance independently of survival data characteristics. The colors of the lines connecting the methods to the horizontal axis represent the specific family of the method (e.g., outcome imputation based, direct-survival CATE, survival meta-learners).

Notably, outcome imputation approaches perform best in randomized settings with unbalanced treatment (e.g., RCT-5%, Figure 7b), where Double-ML achieves the top rank of 1.80. However, their performance deteriorates as we move to settings with unmeasured confounding (Figure 7d), or more visibly with informative censoring (Figures 7f, 7g, 7h), where Double-ML drops to the bottom of the top half and X-Learner falls entirely into the lower-performing half of the rankings. In contrast, survival-specific methods of survival meta-learners such as S-Learner-Survival, Matching-Survival, and Causal Survival Forests (belonging to the direct-survival CATE family) consistently rise in the rankings under these conditions, particularly when multiple violations occur simultaneously (e.g., Figures 7g, 7h), where S-Learner-Survival and Matching-Survival take the top two spots. This trend suggests that survival meta-learners and Causal Survival Forests, which directly model the survival process, offer increased robustness to violations of standard causal assumptions, especially in the presence of unmeasured confounding and informative censoring. Another finding is that Causal Survival Forests maintains strong performance across many configurations, consistently ranking in the top half—particularly in settings involving informative censoring (e.g., Figures 7f, 7g). However, when the positivity assumption is violated while censoring remains ignorable (Figure 7e), its performance declines substantially, dropping to a rank of 6.60 in the bottom half of the ranking. This suggests limitations in modeling highly sparse regions of the covariate space with deterministic treatment assignment under certain censoring conditions.

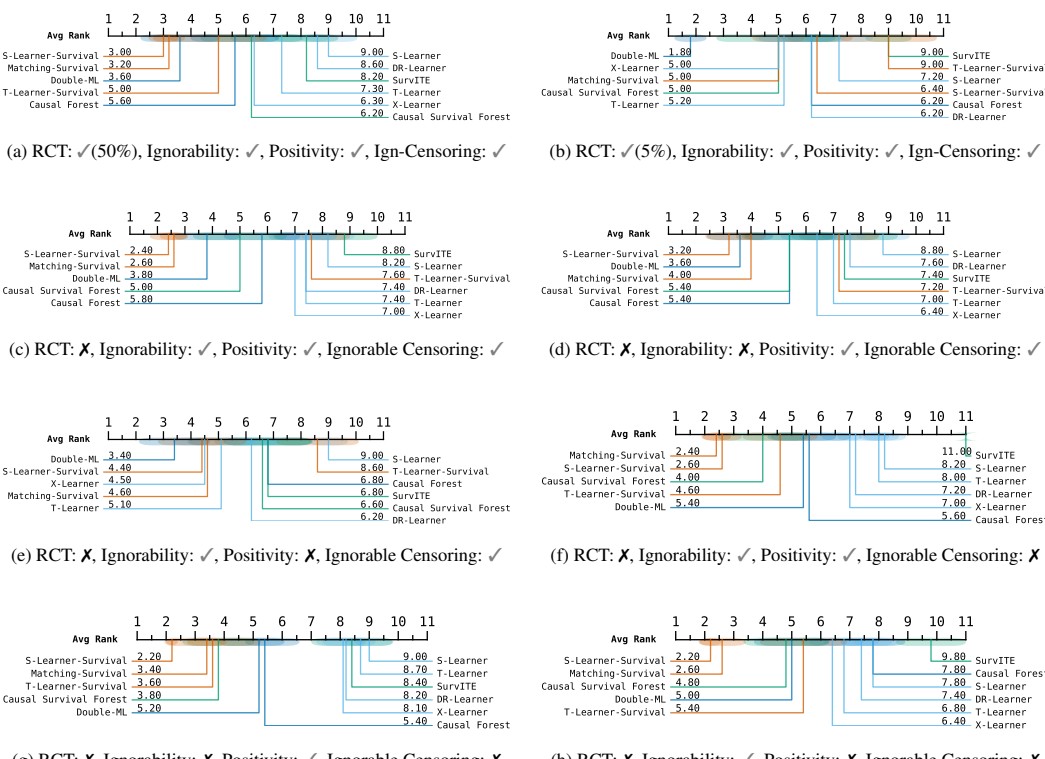

Figure 7: Average ranking of each model for each causal configuration. Shaded regions indicate the standard error of the rank across datasets.

Table 17: Win-rate (%) of method families by Causal Configuration (randomized controlled trial settings). Values denote the percentage of times a method appears in the Top-1, Top-3, and Top-5 according to CATE RMSE and ATE Bias across the five survival scenarios for each configuration.

### RCT-50: 50% treatment rate

| Method Family | CATE RMSE | | | ATE Bias | | |
|---|---|---|---|---|---|---|
| | Top-1 | Top-3 | Top-5 | Top-1 | Top-3 | Top-5 |
| *Outcome Imputation Methods* | | | | | | |
| T-Learner | 0 | 0 | 20.0 | 0 | 20.0 | 60.0 |
| S-Learner | 0 | 0 | 0 | 0 | 0 | 20.0 |
| X-Learner | 0 | 0 | 40.0 | 20.0 | 40.0 | 40.0 |
| DR-Learner | 0 | 0 | 0 | 0 | 20.0 | 40.0 |
| Double-ML | 20.0 | 60.0 | 80.0 | 0 | 0 | 0 |
| Causal Forest | 20.0 | 40.0 | 40.0 | 0 | 0 | 20.0 |
| *Direct-Survival CATE Methods* | | | | | | |
| Causal Survival Forests | 20.0 | 40.0 | 40.0 | 20.0 | 60.0 | 60.0 |
| SurvITE | 0 | 0 | 20.0 | 0 | 20.0 | 20.0 |
| *Survival Meta-Learners* | | | | | | |
| T-Learner-Survival | 20.0 | 40.0 | 60.0 | 20.0 | 40.0 | 80.0 |
| S-Learner-Survival | 20.0 | 60.0 | 100.0 | 40.0 | 60.0 | 80.0 |
| Matching-Survival | 0 | 60.0 | 100.0 | 0 | 40.0 | 80.0 |

### RCT-5: 5% treatment rate

| Method Family | CATE RMSE | | | ATE Bias | | |
|---|---|---|---|---|---|---|
| | Top-1 | Top-3 | Top-5 | Top-1 | Top-3 | Top-5 |
| *Outcome Imputation Methods* | | | | | | |
| T-Learner | 0 | 20.0 | 60.0 | 0 | 20.0 | 60.0 |
| S-Learner | 0 | 20.0 | 20.0 | 20.0 | 20.0 | 20.0 |
| X-Learner | 0 | 20.0 | 60.0 | 0 | 20.0 | 60.0 |
| DR-Learner | 0 | 20.0 | 40.0 | 0 | 20.0 | 60.0 |
| Double-ML | 60.0 | 80.0 | 100.0 | 20.0 | 20.0 | 60.0 |
| Causal Forest | 0 | 0 | 40.0 | 0 | 20.0 | 40.0 |
| *Direct-Survival CATE Methods* | | | | | | |
| Causal Survival Forests | 40.0 | 60.0 | 60.0 | 20.0 | 40.0 | 60.0 |
| SurvITE | 0 | 0 | 0 | 0 | 0 | 0 |
| *Survival Meta-Learners* | | | | | | |
| T-Learner-Survival | 0 | 20.0 | 20.0 | 0 | 0 | 0 |
| S-Learner-Survival | 0 | 20.0 | 40.0 | 0 | 60.0 | 60.0 |
| Matching-Survival | 0 | 40.0 | 60.0 | 40.0 | 80.0 | 80.0 |

In addition to the configuration-agnostic rankings in Figure 7, we report win-rates by causal configuration in Tables 17 and 18. For each configuration, we compute how often each method family appears in the Top-1, Top-3, and Top-5 positions for CATE RMSE and ATE Bias, aggregating over the five survival scenarios. This lets us separate the effect of causal assumptions (randomization, ignorability, positivity, and censoring) from the influence of the survival time model. The randomized settings (`RCT-50`, `RCT-5`) serve as our classical baselines, while the observational settings introduce unmeasured confounding, positivity violations, and informative censoring in a controlled way (Table 1).

In the randomized configurations, CATE RMSE performance is split between outcome-imputation baselines and survival meta-learners, while ATE Bias results tend to favor survival-aware approaches. Under `RCT-50`, Double-ML, Causal Forest, Causal Survival Forests, T-Learner-Survival, and S-Learner-Survival all attain 20.0% Top-1 on CATE RMSE, but the strongest Top-$k$ coverage comes from the survival meta-learners: S-Learner-Survival and Matching-Survival reach 100.0% Top-5, and Matching-Survival achieves 60.0% Top-3. For ATE Bias in `RCT-50`, S-Learner-Survival is the most frequent Top-1 method (40.0%), with Causal Survival Forests and the survival meta-learners (including Matching-Survival and T-Learner-Survival) dominating Top-3/Top-5 (e.g., 60.0% Top-3 for Causal Survival Forests and S-Learner-Survival; 80.0% Top-5 for T-Learner-Survival, S-Learner-Survival, and Matching-Survival). When treatment assignment becomes sparse in `RCT-5`, Double-ML clearly leads on CATE RMSE (60.0% Top-1 and 100.0% Top-5), with Causal Survival Forests following (40.0% Top-1; 60.0% Top-3/Top-5). However, the ATE Bias rankings are largely driven by survival meta-learning: Matching-Survival achieves 40.0% Top-1 and 80.0% Top-3/Top-5, while S-Learner-Survival reaches 60.0% Top-3/Top-5. Notably, SurvITE does not appear among the Top-$k$ in `RCT-5` and has only limited presence in `RCT-50`.

The observational configurations further highlight how assumption violations shift performance toward survival-aware approaches, while revealing clear differences between *direct-survival CATE* and *survival meta-learner* families (Table 18). In `OBS-CPS` (no violations), CATE RMSE Top-1 is split across direct-survival CATE (Causal Survival Forests at 20.0%) and Survival Meta-Learners (S-Learner-Survival and Matching-Survival at 20.0% each), but the strongest Top-$k$ coverage comes from the Survival Meta-Learners: S-Learner-Survival and Matching-Survival reach 80.0% Top-3 and 100.0% Top-5. For ATE Bias in `OBS-CPS`, the lead is primarily driven by Survival Meta-Learners (S-Learner-Survival at 40.0% Top-1; Matching-Survival at 80.0% Top-3 and 100.0% Top-5; T-Learner-Survival at 80.0% Top-5), while the direct-survival CATE method Causal Survival Forests remains highly competitive in the upper ranks (60.0% Top-3 and 80.0% Top-5). Once unmeasured confounding is introduced (`OBS-UConf`), outcome imputation methods remain competitive for CATE RMSE (e.g., Double-ML at 20.0% Top-1 and 80.0% Top-5), but the top positions are shared with direct-survival CATE and Survival Meta-Learners (Causal Survival Forests, SurvITE, and S-Learner-Survival each at 20.0% Top-1). For ATE Bias under `OBS-UConf`, direct-survival CATE takes the clearest lead at the very top (Causal Survival Forests at 40.0% Top-1), while Survival Meta-Learners provide the strongest Top-3 presence (Matching-Survival at 60.0% Top-3) and consis-

tent Top-5 coverage (e.g., 60.0% Top-5 for T-Learner-Survival, S-Learner-Survival, and Matching-Survival). Under positivity violations (`OBS-NoPos`), Outcome imputation dominates CATE RMSE Top-1 (Double-ML at 40.0% and also T-Learner and X-Learner at 20.0% each), whereas the Survival Meta-Learners remain highly competitive (S-Learner-Survival at 40.0% Top-1 and 60.0% Top-5; Matching-Survival at 80.0% Top-5). In contrast, ATE Bias in `OBS-NoPos` is driven mainly by direct-survival CATE and Survival Meta-Learners: Causal Survival Forests reaches 80.0% Top-3/Top-5, while Matching-Survival attains 40.0% Top-1 coverage.

Informative censoring amplifies these differences and further separates the three method families. In `OBS-CPS-InfC`, CATE RMSE Top-1 is led by the direct-survival CATE family via Causal Survival Forests (40.0%), while Survival Meta-Learners dominate Top-3/Top-5 coverage (S-Learner-Survival and Matching-Survival at 80.0% Top-3 and 100.0% Top-5). For ATE Bias in `OBS-CPS-InfC`, Survival Meta-Learners are the clear winners in the upper ranks (S-Learner-Survival at 100.0% Top-3/Top-5; T-Learner-Survival at 40.0% Top-1 and 80.0% Top-5), with direct-survival CATE (Causal Survival Forests) still frequently among the top methods (80.0% Top-5). Under `OBS-UConf-InfC`, CATE RMSE Top-1 is shared by direct-survival CATE (Causal Survival Forests at 40.0%) and Survival Meta-Learners (T-Learner-Survival at 40.0%), while Survival Meta-Learners dominate Top-3/Top-5 (S-Learner-Survival at 100.0% Top-3/Top-5; Matching-Survival at 100.0% Top-5). For ATE Bias in `OBS-UConf-InfC`, the strongest Top-1 signal comes from Survival Meta-Learners (T-Learner-Survival at 40.0%), while direct-survival CATE shows the most dominant Top-5 presence (Causal Survival Forests at 100.0% Top-5). Finally, in `OBS-NoPos-InfC`, Survival Meta-Learners clearly lead CATE RMSE (Matching-Survival at 40.0% Top-1 and 100.0% Top-5; S-Learner-Survival at 100.0% Top-3/Top-5), whereas ATE Bias is again primarily driven by Survival Meta-Learners (T-Learner-Survival at 60.0% Top-1 and 100.0% Top-5), with direct-survival CATE remaining highly competitive in the upper ranks (Causal Survival Forests at 60.0% Top-3 and 80.0% Top-5). Overall, these patterns reinforce that as assumptions are progressively violated, Outcome imputation families can remain competitive for CATE RMSE in some regimes (notably `OBS-NoPos`), but direct-survival CATE and Survival Meta-Learners dominate the upper ranks for ATE Bias, especially under informative censoring and in settings with multiple simultaneous violations.

Table 18: Win-rate (%) of method families by Causal Configuration (observational study settings). Values denote the percentage of times a method appears in the Top-1, Top-3, and Top-5 according to CATE RMSE and ATE Bias across the five survival scenarios for each configuration.

**OBS-CPS**

| Method Family | CATE RMSE | | | ATE Bias | | |
|---|---|---|---|---|---|---|
| | Top-1 | Top-3 | Top-5 | Top-1 | Top-3 | Top-5 |
| *Outcome Imputation Methods* | | | | | | |
| T-Learner | 0 | 0 | 40.0 | 0 | 0 | 20.0 |
| S-Learner | 0 | 0 | 0 | 0 | 0 | 20.0 |
| X-Learner | 0 | 0 | 20.0 | 0 | 20.0 | 20.0 |
| DR-Learner | 0 | 20.0 | 20.0 | 0 | 0 | 0 |
| Double-ML | 20.0 | 40.0 | 80.0 | 0 | 0 | 0 |
| Causal Forest | 20.0 | 20.0 | 40.0 | 0 | 20.0 | 20.0 |
| *Direct-Survival CATE Methods* | | | | | | |
| Causal Survival Forests | 20.0 | 40.0 | 60.0 | 0 | 60.0 | 80.0 |
| SurvITE | 0 | 0 | 20.0 | 20.0 | 20.0 | 60.0 |
| *Survival Meta-Learners* | | | | | | |
| T-Learner-Survival | 0 | 20.0 | 20.0 | 20.0 | 40.0 | 80.0 |
| S-Learner-Survival | 20.0 | 80.0 | 100.0 | 40.0 | 60.0 | 100.0 |
| Matching-Survival | 20.0 | 80.0 | 100.0 | 20.0 | 80.0 | 100.0 |

**OBS-UConf**

| Method Family | CATE RMSE | | | ATE Bias | | |
|---|---|---|---|---|---|---|
| | Top-1 | Top-3 | Top-5 | Top-1 | Top-3 | Top-5 |
| *Outcome Imputation Methods* | | | | | | |
| T-Learner | 0 | 20.0 | 20.0 | 0 | 20.0 | 40.0 |
| S-Learner | 0 | 0 | 0 | 0 | 40.0 | 40.0 |
| X-Learner | 0 | 20.0 | 40.0 | 0 | 0 | 60.0 |
| DR-Learner | 0 | 0 | 20.0 | 20.0 | 40.0 | 40.0 |
| Double-ML | 20.0 | 60.0 | 80.0 | 0 | 0 | 20.0 |
| Causal Forest | 20.0 | 40.0 | 40.0 | 0 | 0 | 20.0 |
| *Direct-Survival CATE Methods* | | | | | | |
| Causal Survival Forests | 20.0 | 40.0 | 60.0 | 40.0 | 60.0 | 60.0 |
| SurvITE | 20.0 | 20.0 | 40.0 | 20.0 | 40.0 | 40.0 |
| *Survival Meta-Learners* | | | | | | |
| T-Learner-Survival | 0 | 20.0 | 20.0 | 0 | 0 | 60.0 |
| S-Learner-Survival | 20.0 | 60.0 | 80.0 | 0 | 40.0 | 60.0 |
| Matching-Survival | 0 | 20.0 | 100.0 | 20.0 | 60.0 | 60.0 |

**OBS-NoPos**

| Method Family | CATE RMSE | | | ATE Bias | | |
|---|---|---|---|---|---|---|
| | Top-1 | Top-3 | Top-5 | Top-1 | Top-3 | Top-5 |
| *Outcome Imputation Methods* | | | | | | |
| T-Learner | 20.0 | 40.0 | 60.0 | 0 | 20.0 | 40.0 |
| S-Learner | 0 | 0 | 0 | 0 | 40.0 | 60.0 |
| X-Learner | 20.0 | 40.0 | 80.0 | 0 | 20.0 | 60.0 |
| DR-Learner | 0 | 20.0 | 20.0 | 0 | 0 | 40.0 |
| Double-ML | 40.0 | 60.0 | 60.0 | 0 | 20.0 | 20.0 |
| Causal Forest | 0 | 20.0 | 40.0 | 20.0 | 20.0 | 20.0 |
| *Direct-Survival CATE Methods* | | | | | | |
| Causal Survival Forests | 0 | 40.0 | 40.0 | 20.0 | 80.0 | 80.0 |
| SurvITE | 0 | 20.0 | 40.0 | 0 | 20.0 | 40.0 |
| *Survival Meta-Learners* | | | | | | |
| T-Learner-Survival | 0 | 20.0 | 20.0 | 20.0 | 20.0 | 20.0 |
| S-Learner-Survival | 40.0 | 40.0 | 60.0 | 0 | 20.0 | 80.0 |
| Matching-Survival | 0 | 20.0 | 80.0 | 40.0 | 40.0 | 40.0 |

**OBS-CPS-InfC**

| Method Family | CATE RMSE | | | ATE Bias | | |
|---|---|---|---|---|---|---|
| | Top-1 | Top-3 | Top-5 | Top-1 | Top-3 | Top-5 |
| *Outcome Imputation Methods* | | | | | | |
| T-Learner | 0 | 0 | 0 | 0 | 0 | 0 |
| S-Learner | 0 | 0 | 0 | 0 | 0 | 0 |
| X-Learner | 0 | 0 | 0 | 0 | 0 | 40.0 |
| DR-Learner | 0 | 0 | 40.0 | 20.0 | 20.0 | 20.0 |
| Double-ML | 0 | 40.0 | 60.0 | 0 | 0 | 0 |
| Causal Forest | 0 | 20.0 | 60.0 | 0 | 0 | 40.0 |
| *Direct-Survival CATE Methods* | | | | | | |
| Causal Survival Forests | 40.0 | 60.0 | 60.0 | 20.0 | 40.0 | 80.0 |
| SurvITE | 0 | 0 | 0 | 20.0 | 20.0 | 60.0 |
| *Survival Meta-Learners* | | | | | | |
| T-Learner-Survival | 20.0 | 20.0 | 80.0 | 40.0 | 60.0 | 80.0 |
| S-Learner-Survival | 20.0 | 80.0 | 100.0 | 0 | 100.0 | 100.0 |
| Matching-Survival | 20.0 | 80.0 | 100.0 | 0 | 60.0 | 80.0 |

**OBS-UConf-InfC**

| Method Family | CATE RMSE | | | ATE Bias | | |
|---|---|---|---|---|---|---|
| | Top-1 | Top-3 | Top-5 | Top-1 | Top-3 | Top-5 |
| *Outcome Imputation Methods* | | | | | | |
| T-Learner | 0 | 0 | 0 | 0 | 20.0 | 20.0 |
| S-Learner | 0 | 0 | 0 | 20.0 | 20.0 | 20.0 |
| X-Learner | 0 | 0 | 0 | 0 | 20.0 | 20.0 |
| DR-Learner | 0 | 0 | 0 | 0 | 0 | 20.0 |
| Double-ML | 0 | 40.0 | 60.0 | 0 | 0 | 20.0 |
| Causal Forest | 0 | 0 | 60.0 | 0 | 0 | 20.0 |
| *Direct-Survival CATE Methods* | | | | | | |
| Causal Survival Forests | 40.0 | 40.0 | 60.0 | 0 | 20.0 | 100.0 |
| SurvITE | 0 | 0 | 40.0 | 0 | 20.0 | 40.0 |
| *Survival Meta-Learners* | | | | | | |
| T-Learner-Survival | 40.0 | 80.0 | 80.0 | 40.0 | 60.0 | 80.0 |
| S-Learner-Survival | 0 | 100.0 | 100.0 | 20.0 | 80.0 | 80.0 |
| Matching-Survival | 20.0 | 40.0 | 100.0 | 20.0 | 60.0 | 80.0 |

**OBS-NoPos-InfC**

| Method Family | CATE RMSE | | | ATE Bias | | |
|---|---|---|---|---|---|---|
| | Top-1 | Top-3 | Top-5 | Top-1 | Top-3 | Top-5 |
| *Outcome Imputation Methods* | | | | | | |
| T-Learner | 0 | 20.0 | 40.0 | 20.0 | 20.0 | 20.0 |
| S-Learner | 0 | 0 | 0 | 0 | 0 | 20.0 |
| X-Learner | 0 | 20.0 | 40.0 | 20.0 | 20.0 | 20.0 |
| DR-Learner | 0 | 0 | 40.0 | 0 | 0 | 20.0 |
| Double-ML | 0 | 40.0 | 60.0 | 0 | 0 | 20.0 |
| Causal Forest | 0 | 0 | 20.0 | 0 | 0 | 40.0 |
| *Direct-Survival CATE Methods* | | | | | | |
| Causal Survival Forests | 20.0 | 40.0 | 40.0 | 20.0 | 60.0 | 80.0 |
| SurvITE | 0 | 0 | 20.0 | 0 | 40.0 | 60.0 |
| *Survival Meta-Learners* | | | | | | |
| T-Learner-Survival | 20.0 | 40.0 | 40.0 | 60.0 | 80.0 | 100.0 |
| S-Learner-Survival | 20.0 | 100.0 | 100.0 | 0 | 40.0 | 60.0 |
| Matching-Survival | 40.0 | 60.0 | 100.0 | 0 | 40.0 | 60.0 |

## F.4 Figure results - CATE RMSE

This section presents the complete CATE RMSE results for each family of causal inference methods across various survival analysis scenarios. For each scenario, we display performance under 8 distinct causal configurations, each varying in terms of treatment assignment (RCT vs. observational), ignorability, positivity, and censoring assumptions. These results highlight the robustness and sensitivity of different methods under varying degrees of assumption violations.

For each survival scenario and causal configuration, we selected the best hyperparameter setting and base model configuration for each causal method family based on validation set performance. The RMSE values shown in the figures reflect the performance of these selected models on the test set. The box plots are from the 10 independent experimental repeats to account for random seed variability.

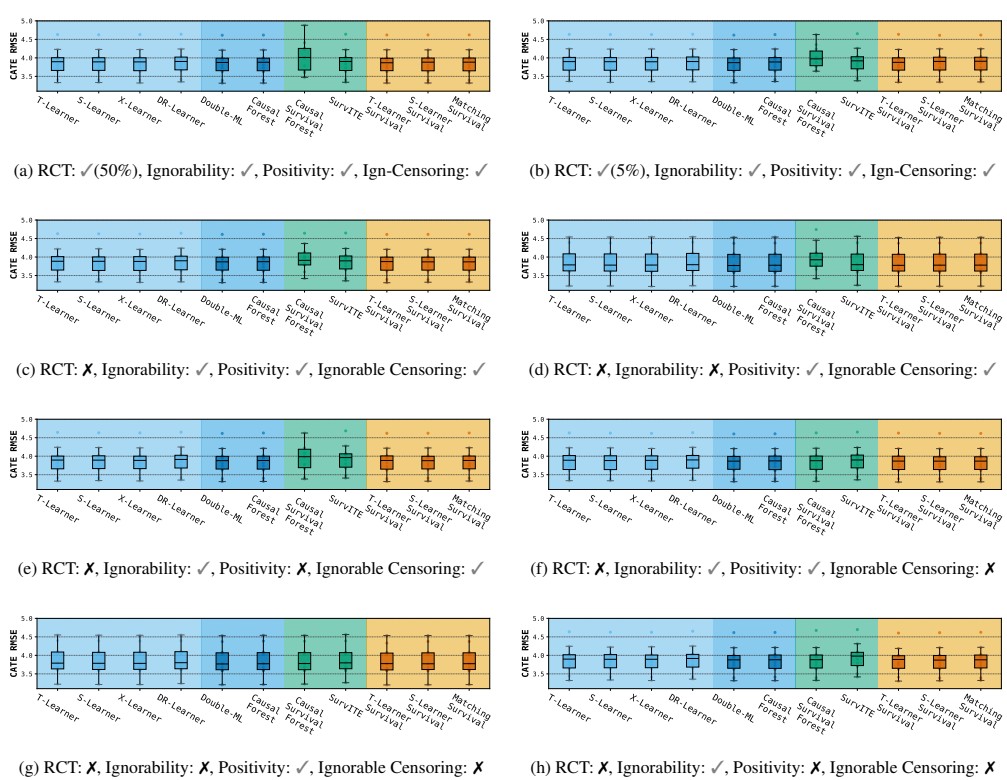

Figure 8: CATE RMSE across different experiments in Scenario A.

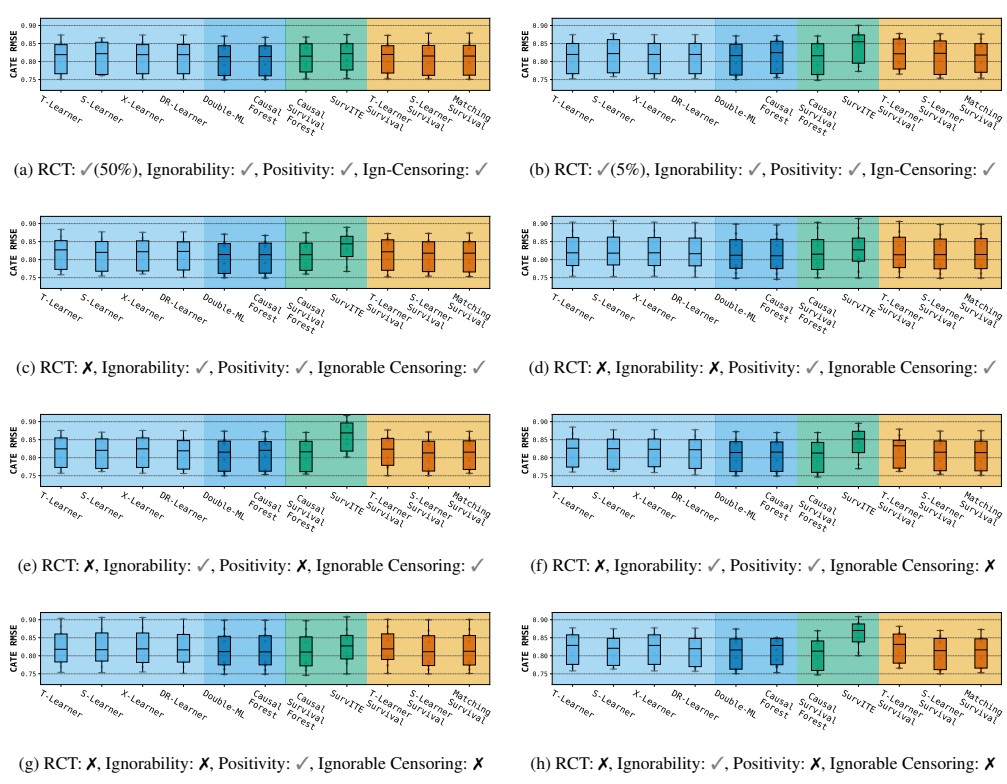

(a) RCT: ✓(50%), Ignorability: ✓, Positivity: ✓, Ign-Censoring: ✓     (b) RCT: ✓(5%), Ignorability: ✓, Positivity: ✓, Ign-Censoring: ✓

(c) RCT: ✗, Ignorability: ✓, Positivity: ✓, Ignorable Censoring: ✓     (d) RCT: ✗, Ignorability: ✗, Positivity: ✓, Ignorable Censoring: ✓

(e) RCT: ✗, Ignorability: ✓, Positivity: ✗, Ignorable Censoring: ✓     (f) RCT: ✗, Ignorability: ✓, Positivity: ✓, Ignorable Censoring: ✗

(g) RCT: ✗, Ignorability: ✗, Positivity: ✓, Ignorable Censoring: ✗     (h) RCT: ✗, Ignorability: ✓, Positivity: ✗, Ignorable Censoring: ✗

Figure 9: CATE RMSE across different experiments in Scenario B.

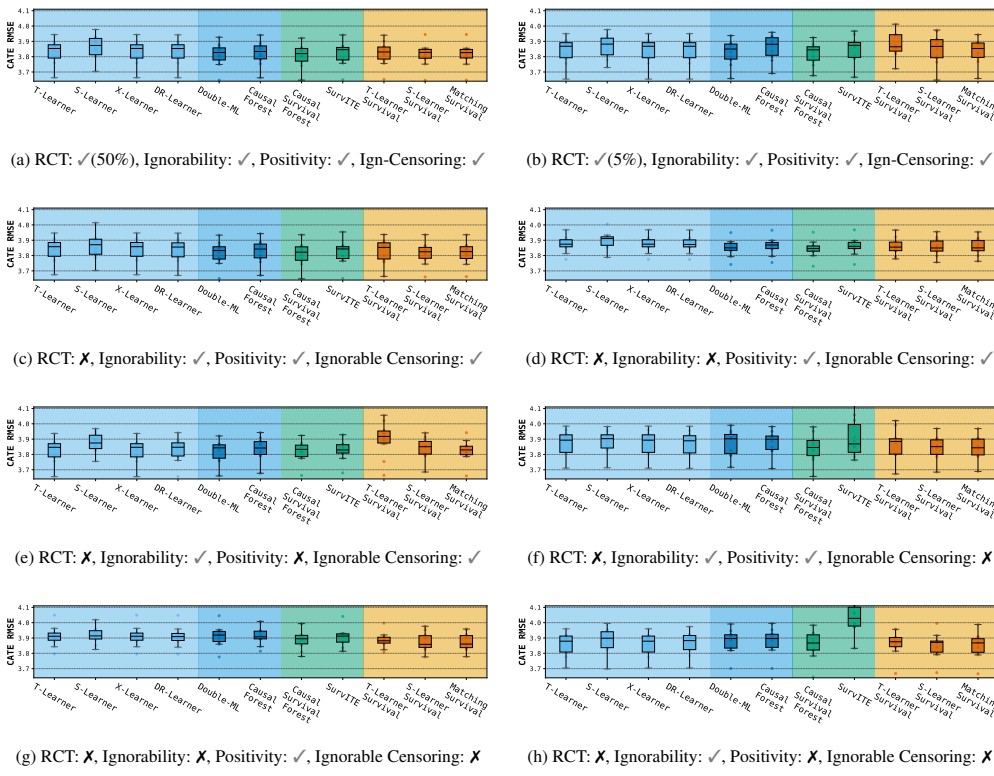

(a) RCT: ✓(50%), Ignorability: ✓, Positivity: ✓, Ign-Censoring: ✓     (b) RCT: ✓(5%), Ignorability: ✓, Positivity: ✓, Ign-Censoring: ✓

(c) RCT: ✗, Ignorability: ✓, Positivity: ✓, Ignorable Censoring: ✓     (d) RCT: ✗, Ignorability: ✗, Positivity: ✓, Ignorable Censoring: ✓

(e) RCT: ✗, Ignorability: ✓, Positivity: ✗, Ignorable Censoring: ✓     (f) RCT: ✗, Ignorability: ✓, Positivity: ✓, Ignorable Censoring: ✗

(g) RCT: ✗, Ignorability: ✗, Positivity: ✓, Ignorable Censoring: ✗     (h) RCT: ✗, Ignorability: ✓, Positivity: ✗, Ignorable Censoring: ✗

Figure 10: CATE RMSE across different experiments in Scenario C.

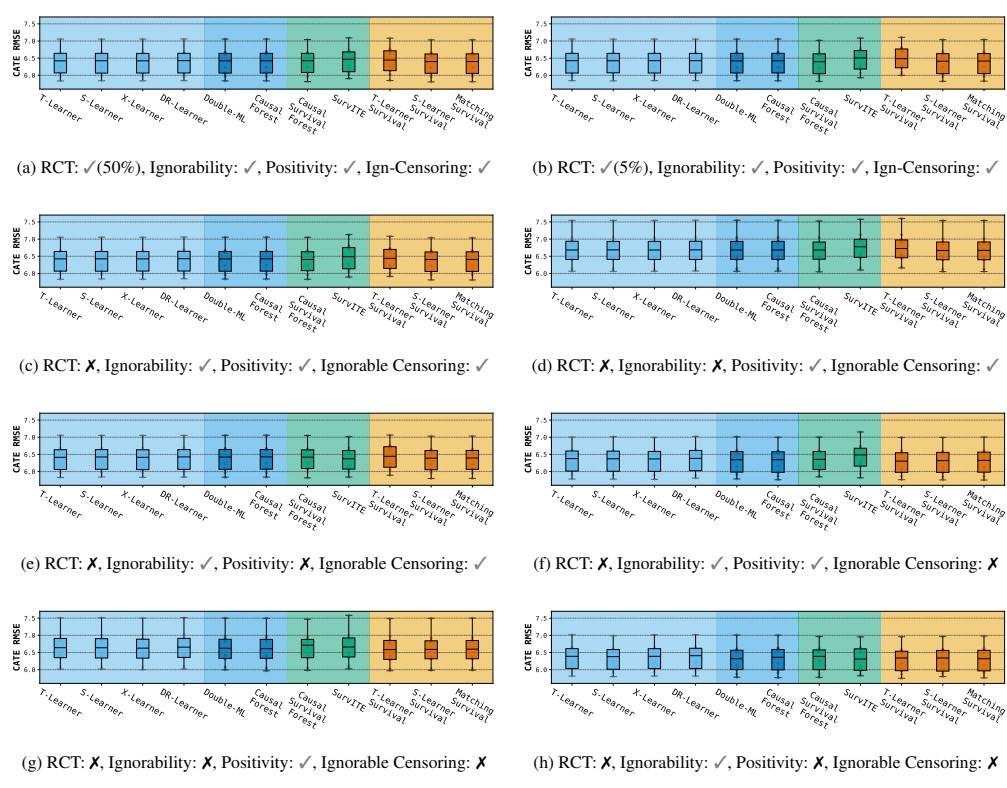

Figure 11: CATE RMSE across different experiments in Scenario D.

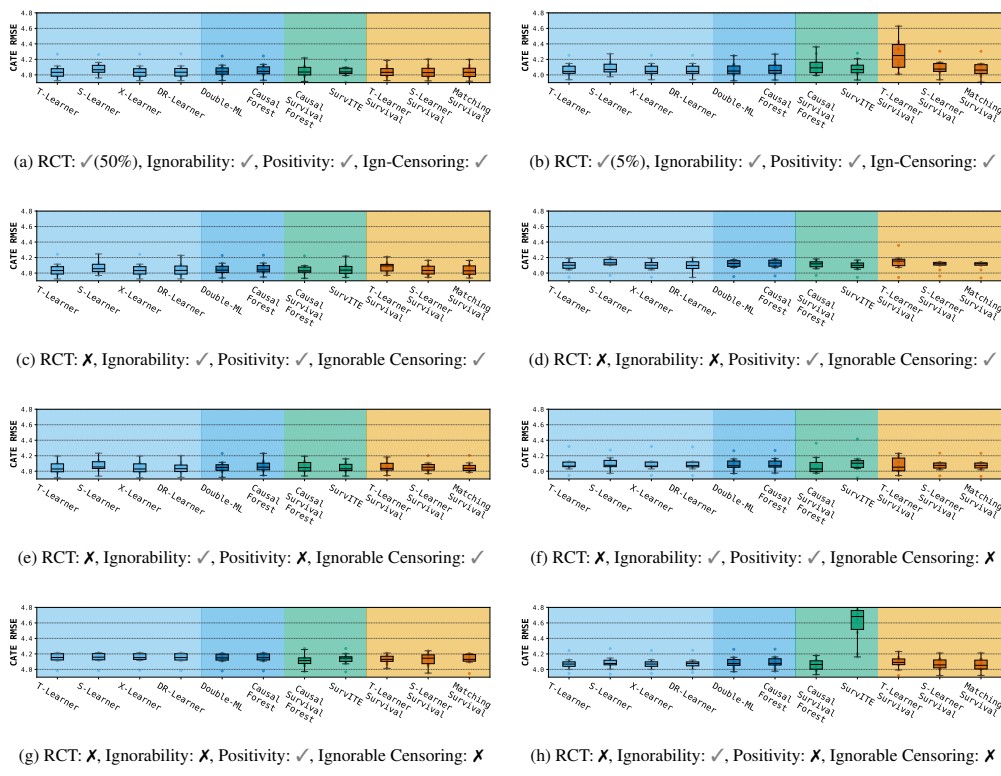

Figure 12: CATE RMSE across different experiments in Scenario E.

### F.5 FIGURE RESULTS - ATE BIAS

This section presents the ATE bias results for each family of causal inference methods across various survival scenarios. As with the CATE RMSE results in Appendix F.4, we display performance under 8 distinct causal configurations per scenario, each varying in treatment assignment (RCT vs. observational), ignorability, positivity, and censoring assumptions.

For each survival scenario and causal configuration, the model shown corresponds to the best hyperparameter setting and base model configuration selected based on CATE RMSE performance on the validation set — ATE bias was not used for model selection for consistent results with other sections. The reported ATE bias values are computed on the test set and defined as the difference between the *predicted ATE* from the test population and the *true ATE* in the full population.

Each box plot represents results from 10 independent experimental repeats to account for random seed variability. For meta-learners and double machine learning models, which by design can provide 95% confidence intervals for ATE estimates, we also include these intervals in the plots— adjusted accordingly to center around the ATE bias. These confidence intervals are obtained via 100 bootstrap samples and are notably wider than the variability observed across the 10 experimental repeats. The zero bias line is shown as a dashed reference line.

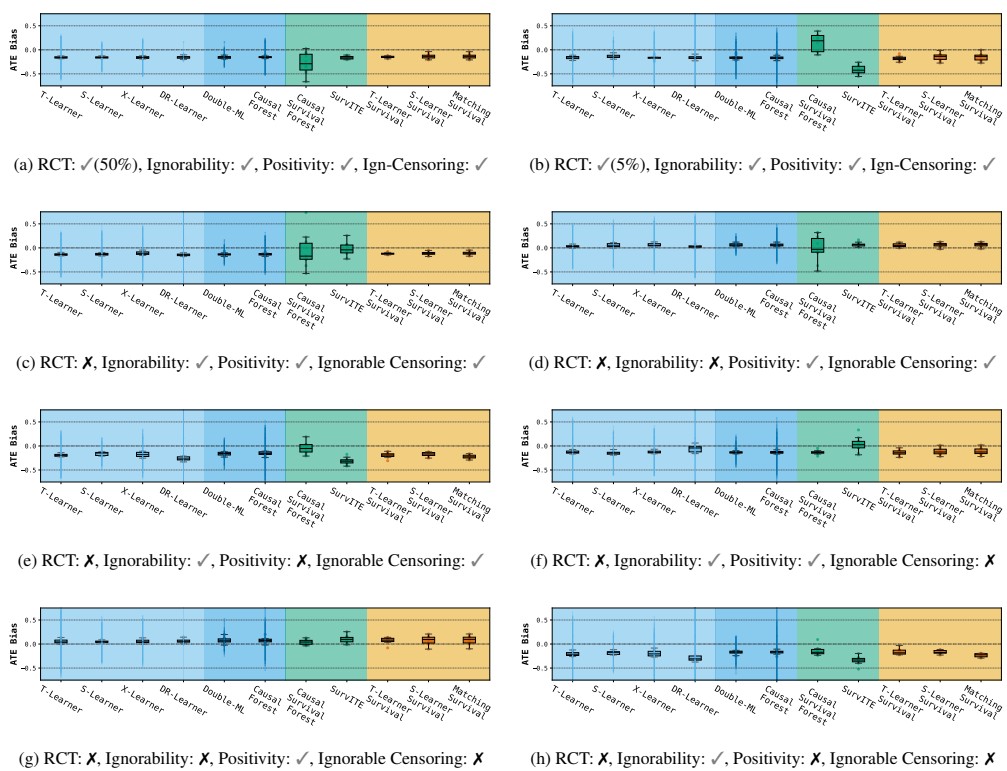

(a) RCT: ✓(50%), Ignorability: ✓, Positivity: ✓, Ign-Censoring: ✓

(b) RCT: ✓(5%), Ignorability: ✓, Positivity: ✓, Ign-Censoring: ✓

(c) RCT: ✗, Ignorability: ✓, Positivity: ✓, Ignorable Censoring: ✓

(d) RCT: ✗, Ignorability: ✗, Positivity: ✓, Ignorable Censoring: ✓

(e) RCT: ✗, Ignorability: ✓, Positivity: ✗, Ignorable Censoring: ✓

(f) RCT: ✗, Ignorability: ✓, Positivity: ✓, Ignorable Censoring: ✗

(g) RCT: ✗, Ignorability: ✗, Positivity: ✓, Ignorable Censoring: ✗

(h) RCT: ✗, Ignorability: ✓, Positivity: ✗, Ignorable Censoring: ✗

Figure 13: ATE Bias across different experiments in Scenario A.

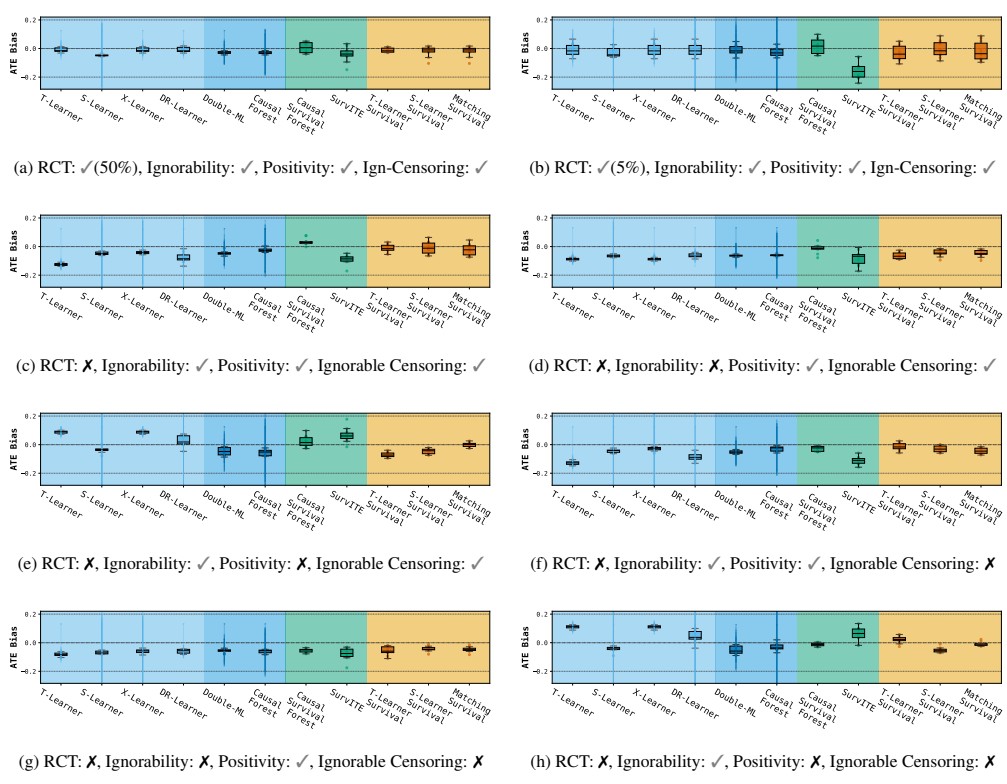

(a) RCT: ✓(50%), Ignorability: ✓, Positivity: ✓, Ign-Censoring: ✓

(b) RCT: ✓(5%), Ignorability: ✓, Positivity: ✓, Ign-Censoring: ✓

(c) RCT: ✗, Ignorability: ✓, Positivity: ✓, Ignorable Censoring: ✓

(d) RCT: ✗, Ignorability: ✗, Positivity: ✓, Ignorable Censoring: ✓

(e) RCT: ✗, Ignorability: ✓, Positivity: ✗, Ignorable Censoring: ✓

(f) RCT: ✗, Ignorability: ✓, Positivity: ✓, Ignorable Censoring: ✗

(g) RCT: ✗, Ignorability: ✗, Positivity: ✓, Ignorable Censoring: ✗

(h) RCT: ✗, Ignorability: ✓, Positivity: ✗, Ignorable Censoring: ✗

Figure 14: ATE Bias across different experiments in Scenario B.

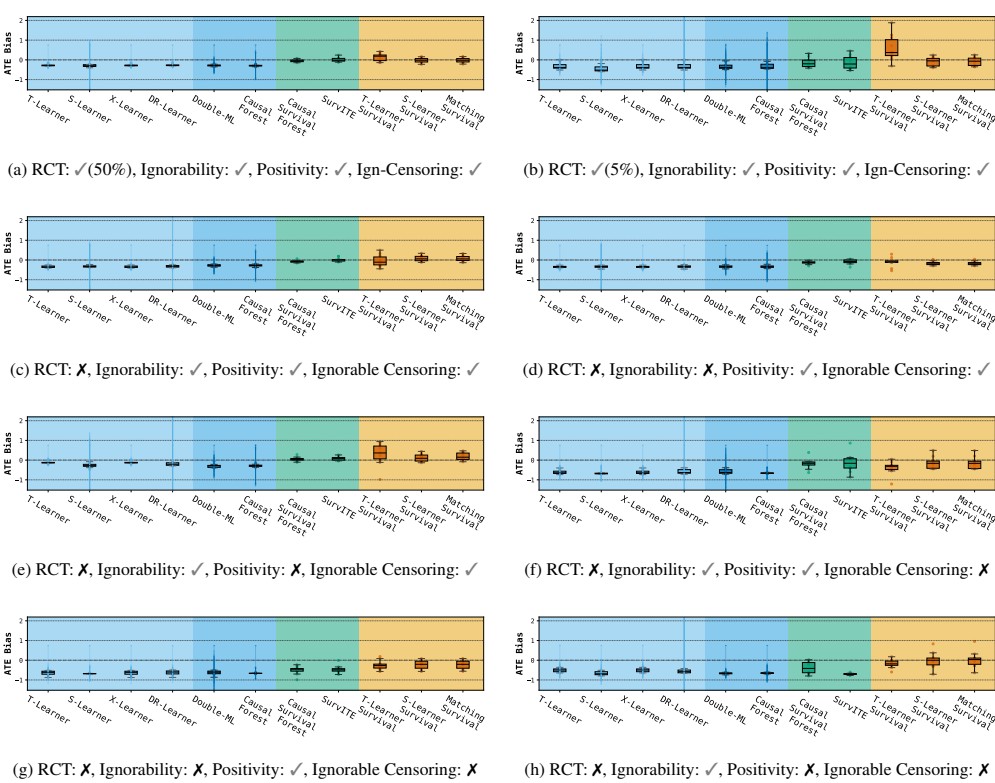

(a) RCT: ✓(50%), Ignorability: ✓, Positivity: ✓, Ign-Censoring: ✓

(b) RCT: ✓(5%), Ignorability: ✓, Positivity: ✓, Ign-Censoring: ✓

(c) RCT: ✗, Ignorability: ✓, Positivity: ✓, Ignorable Censoring: ✓

(d) RCT: ✗, Ignorability: ✗, Positivity: ✓, Ignorable Censoring: ✓

(e) RCT: ✗, Ignorability: ✓, Positivity: ✗, Ignorable Censoring: ✓

(f) RCT: ✗, Ignorability: ✓, Positivity: ✓, Ignorable Censoring: ✗

(g) RCT: ✗, Ignorability: ✗, Positivity: ✓, Ignorable Censoring: ✗

(h) RCT: ✗, Ignorability: ✓, Positivity: ✗, Ignorable Censoring: ✗

Figure 15: ATE Bias across different experiments in Scenario C.

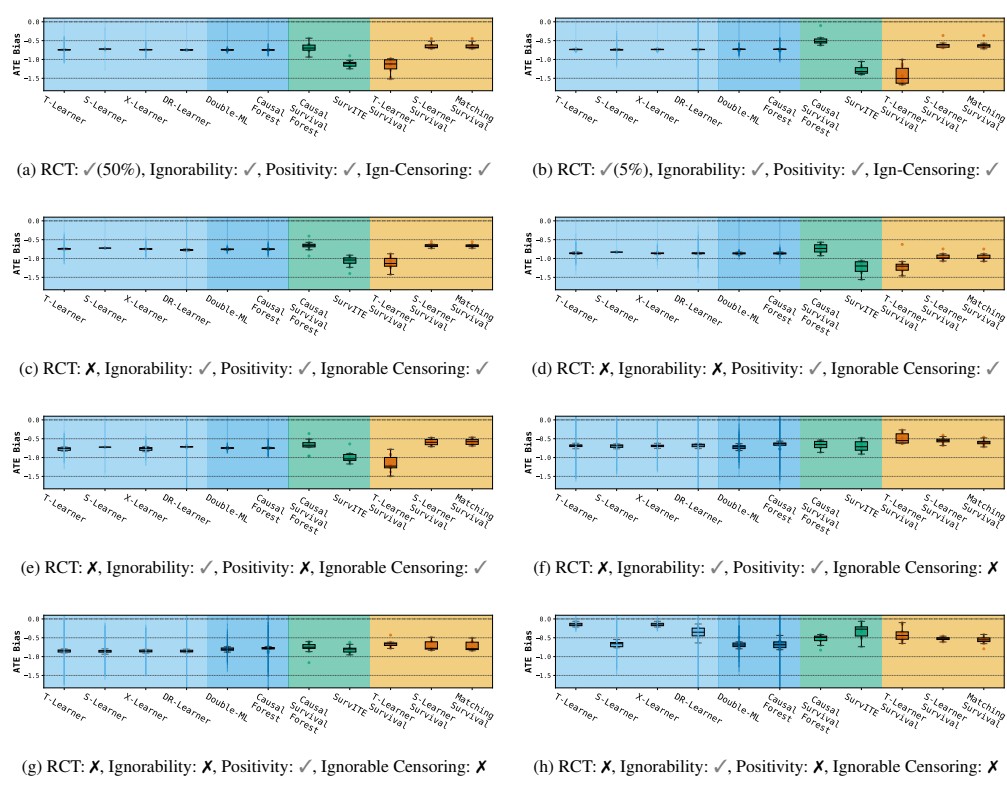

(a) RCT: ✓(50%), Ignorability: ✓, Positivity: ✓, Ign-Censoring: ✓     (b) RCT: ✓(5%), Ignorability: ✓, Positivity: ✓, Ign-Censoring: ✓

(c) RCT: ✗, Ignorability: ✓, Positivity: ✓, Ignorable Censoring: ✓     (d) RCT: ✗, Ignorability: ✗, Positivity: ✓, Ignorable Censoring: ✓

(e) RCT: ✗, Ignorability: ✓, Positivity: ✗, Ignorable Censoring: ✓     (f) RCT: ✗, Ignorability: ✓, Positivity: ✓, Ignorable Censoring: ✗

(g) RCT: ✗, Ignorability: ✗, Positivity: ✓, Ignorable Censoring: ✗     (h) RCT: ✗, Ignorability: ✓, Positivity: ✗, Ignorable Censoring: ✗

Figure 16: ATE Bias across different experiments in Scenario D.

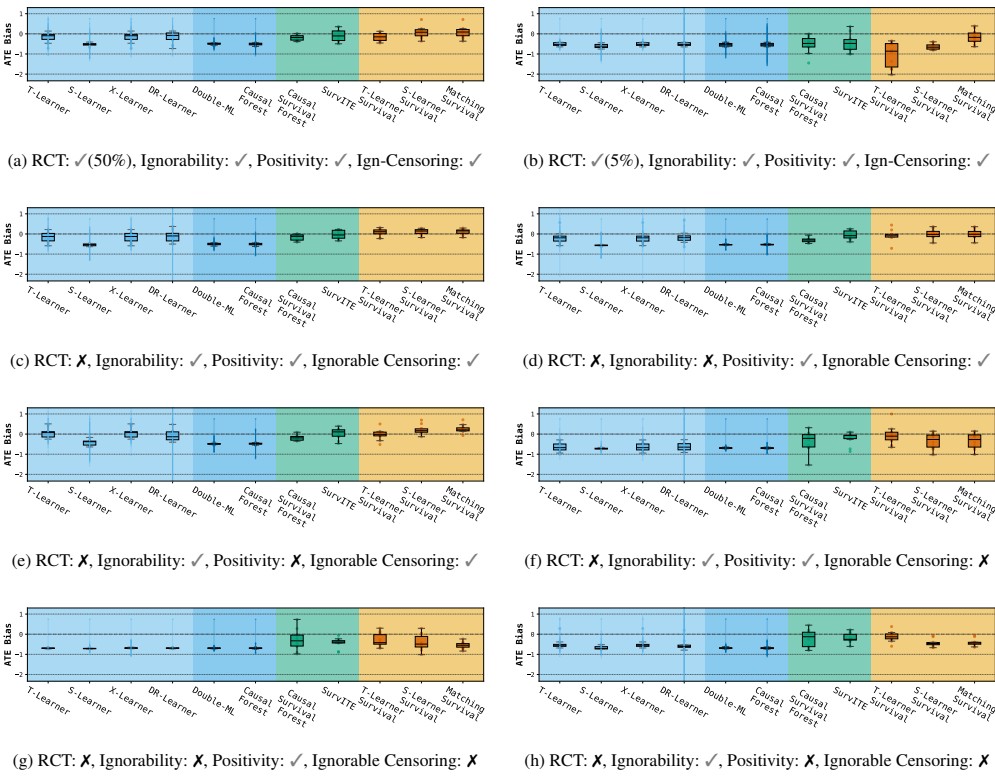

(a) RCT: ✓(50%), Ignorability: ✓, Positivity: ✓, Ign-Censoring: ✓     (b) RCT: ✓(5%), Ignorability: ✓, Positivity: ✓, Ign-Censoring: ✓

(c) RCT: ✗, Ignorability: ✓, Positivity: ✓, Ignorable Censoring: ✓     (d) RCT: ✗, Ignorability: ✗, Positivity: ✓, Ignorable Censoring: ✓

(e) RCT: ✗, Ignorability: ✓, Positivity: ✗, Ignorable Censoring: ✓     (f) RCT: ✗, Ignorability: ✓, Positivity: ✓, Ignorable Censoring: ✗

(g) RCT: ✗, Ignorability: ✗, Positivity: ✓, Ignorable Censoring: ✗     (h) RCT: ✗, Ignorability: ✓, Positivity: ✗, Ignorable Censoring: ✗

Figure 17: ATE Bias across different experiments in Scenario E.

## F.6 EVALUATION ON AUXILIARY IMPUTATION AND BASE LEARNERS

In this section, we report the performance of auxiliary imputation and base regression or survival learners on the test sets.

### F.6.1 IMPUTATION EVALUATION

Table 19 reports the MAE of the three imputation methods (Pseudo-obs, Margin, IPCW-T) across eight causal configurations and five censoring scenarios on the test sets. Recall that Scenarios A and B have low censoring (<30%), Scenario C medium (30–70%), and Scenarios D and E high (>70%), except it is switched in -InfC causal configurations, as mentioned in Appendix A. We can tell that the imputation method Pseudo-obs is only competitive under minimal censoring and suffers from high variability. Margin imputation provides the best balance of accuracy and robustness, especially as censoring intensifies. IPCW-T imputation improves over Pseudo-obs in most cases, but generally underperforms relative to Margin in medium- and high-censor contexts.

Table 19: Evaluation on imputation methods across different survival scenarios and causal configurations. MAE between the imputed and true event times on testing set is reported as mean $\pm$ std. over 10 experimental repeats. "Total Win" row counts the number of survival configurations $\times$ random split combinations ($8 \times 10 = 80$) in which each method achieved the lowest MAE, and is calculated within each scenario. The same rule applies to all the tables below in Appendix F.6.

| Survival Scenario | Causal Configuration | Imputation Method | | |
|---|---|---|---|---|
| | | Pseudo-obs | Margin | IPCW-T |
| A | RCT-50 | **0.437**±**0.021** | 0.446±0.025 | 0.470±0.027 |
| | RCT-5 | **0.378**±**0.027** | 0.387±0.029 | 0.405±0.032 |
| | OBS-CPS | **0.448**±**0.014** | 0.459±0.014 | 0.481±0.015 |
| | OBS-UConf | **0.423**±**0.026** | 0.520±0.028 | 0.455±0.03 |
| | OBS-NoPos | **0.411**±**0.023** | 0.420±0.023 | 0.442±0.025 |
| | OBS-CPS-InfC | 0.390±0.020 | **0.374**±**0.014** | 0.388±0.014 |
| | OBS-UConf-InfC | 0.369±0.029 | 0.482±0.027 | **0.362**±**0.028** |
| | OBS-NoPos-InfC | 0.347±0.023 | **0.336**±**0.024** | 0.349±0.026 |
| | Total Win | 51 | 21 | 8 |
| B | RCT-50 | 0.061±0.005 | 0.05±0.003 | **0.048**±**0.004** |
| | RCT-5 | 0.027±0.003 | 0.022±0.002 | **0.021**±**0.003** |
| | OBS-CPS | 0.052±0.005 | 0.042±0.004 | **0.040**±**0.003** |
| | OBS-UConf | 0.058±0.004 | 0.152±0.007 | **0.046**±**0.004** |
| | OBS-NoPos | 0.068±0.008 | 0.057±0.005 | **0.056**±**0.005** |
| | OBS-CPS-InfC | 0.039±0.005 | 0.037±0.005 | **0.036**±**0.005** |
| | OBS-UConf-InfC | 0.040±0.004 | 0.140±0.005 | **0.038**±**0.004** |
| | OBS-NoPos-InfC | 0.048±0.007 | 0.046±0.008 | **0.045**±**0.008** |
| | Total Win | 0 | 3 | 77 |
| C | RCT-50 | **0.837**±**0.008** | 0.838±0.008 | 0.841±0.007 |
| | RCT-5 | 0.803±0.013 | 0.804±0.013 | **0.793**±**0.009** |
| | OBS-CPS | 0.829±0.014 | 0.830±0.014 | **0.828**±**0.014** |
| | OBS-UConf | **0.835**±**0.026** | 2.701±0.033 | 0.837±0.027 |
| | OBS-NoPos | **0.845**±**0.014** | **0.845**±**0.015** | 0.855±0.012 |
| | OBS-CPS-InfC | 2.786±0.079 | **2.090**±**0.046** | 2.858±0.055 |
| | OBS-UConf-InfC | **2.753**±**0.074** | 2.443±0.023 | 2.852±0.058 |
| | OBS-NoPos-InfC | 2.904±0.061 | **2.197**±**0.045** | 3.006±0.034 |
| | Total Win | 23 | 35 | 22 |
| D | RCT-50 | 3.303±0.333 | **2.241**±**0.065** | 2.624±0.054 |
| | RCT-5 | 2.897±0.257 | **1.845**±**0.059** | 2.192±0.059 |
| | OBS-CPS | 3.191±0.449 | **2.109**±**0.062** | 2.421±0.068 |
| | OBS-UConf | 3.463±0.706 | **2.361**±**0.198** | 2.610±0.073 |
| | OBS-NoPos | 3.536±0.435 | **2.404**±**0.074** | 2.853±0.072 |
| | OBS-CPS-InfC | 1.395±0.067 | **1.289**±**0.064** | 1.366±0.068 |
| | OBS-UConf-InfC | 1.524±0.069 | 1.737±0.054 | **1.511**±**0.063** |
| | OBS-NoPos-InfC | 1.689±0.074 | **1.595**±**0.069** | 1.698±0.073 |
| | Total Win | 3 | 68 | 9 |
| E | RCT-50 | 2.672±0.348 | **1.595**±**0.019** | 2.033±0.022 |
| | RCT-5 | 2.238±0.218 | **1.468**±**0.023** | 1.823±0.023 |
| | OBS-CPS | 2.446±0.262 | **1.577**±**0.022** | 1.992±0.032 |
| | OBS-UConf | 2.531±0.191 | 2.651±0.054 | **2.051**±**0.031** |
| | OBS-NoPos | 2.669±0.288 | **1.639**±**0.021** | 2.102±0.035 |
| | OBS-CPS-InfC | 3.324±0.136 | **2.483**±**0.05** | 3.491±0.054 |
| | OBS-UConf-InfC | 3.346±0.147 | **2.686**±**0.071** | 3.526±0.036 |
| | OBS-NoPos-InfC | 3.373±0.101 | **2.541**±**0.038** | 3.648±0.072 |
| | Total Win | 0 | 70 | 10 |

F.6.2 BASE REGRESSION LEARNER EVALUATION

See Table 20, 21, 22, 23 for MAE of prediction by the base regression learners for S-, T-, X-, DR-Learners. The MAE is calculated by comparing a base learner's predicted event times and imputed event times by the imputation method (the latter is used as the "ground truth" for the base regression learners). Since there are three imputation methods used, we first take the average of MAE across three different imputation methods within each random split, then report the mean and standard deviation of the average MAE across 10 experimental repeats with different random splits.

See Table 24 for the AUC on the evaluation of the predicted propensity score of DR-Learners.

Table 20: S-Learner MAE

| Survival Scenario | Causal Configuration | Base Regression Model | | |
|---|---|---|---|---|
| | | Lasso Reg. | Random Forest | XGBoost |
| A | RCT-50 | 0.661±0.012 | **0.655±0.014** | 0.671±0.013 |
| | RCT-5 | **0.645±0.011** | 0.649±0.012 | 0.667±0.014 |
| | OBS-CPS | 0.653±0.011 | **0.646±0.010** | 0.662±0.012 |
| | OBS-UConf | **0.604±0.009** | 0.608±0.009 | 0.620±0.007 |
| | OBS-NoPos | 0.657±0.010 | **0.654±0.011** | 0.671±0.013 |
| | OBS-CPS-InfC | **0.727±0.018** | 0.730±0.021 | 0.752±0.023 |
| | OBS-UConf-InfC | **0.675±0.019** | 0.693±0.023 | 0.713±0.023 |
| | OBS-NoPos-InfC | **0.724±0.014** | 0.732±0.018 | 0.755±0.02 |
| | Total Win | 44 | 36 | 0 |
| B | RCT-50 | 0.33±0.008 | **0.315±0.011** | 0.334±0.015 |
| | RCT-5 | 0.278±0.006 | **0.277±0.006** | 0.292±0.008 |
| | OBS-CPS | 0.315±0.011 | **0.307±0.012** | 0.324±0.019 |
| | OBS-UConf | 0.354±0.007 | **0.341±0.008** | 0.359±0.011 |
| | OBS-NoPos | 0.345±0.009 | **0.323±0.011** | 0.341±0.011 |
| | OBS-CPS-InfC | 0.301±0.007 | **0.294±0.006** | 0.309±0.008 |
| | OBS-UConf-InfC | 0.34±0.004 | **0.328±0.005** | 0.347±0.006 |
| | OBS-NoPos-InfC | 0.337±0.005 | **0.319±0.006** | 0.337±0.007 |
| | Total Win | 3 | 77 | 0 |
| C | RCT-50 | **1.430±0.022** | 1.593±0.025 | 1.738±0.024 |
| | RCT-5 | **1.403±0.019** | 1.532±0.018 | 1.687±0.027 |
| | OBS-CPS | **1.409±0.021** | 1.555±0.025 | 1.706±0.028 |
| | OBS-UConf | **1.419±0.023** | 1.563±0.026 | 1.721±0.020 |
| | OBS-NoPos | **1.453±0.019** | 1.612±0.024 | 1.755±0.022 |
| | OBS-CPS-InfC | **0.931±0.027** | 1.011±0.03 | 1.148±0.045 |
| | OBS-UConf-InfC | **0.895±0.033** | 0.966±0.04 | 1.096±0.059 |
| | OBS-NoPos-InfC | **0.916±0.057** | 1.003±0.067 | 1.123±0.086 |
| | Total Win | 80 | 0 | 0 |
| D | RCT-50 | **0.941±0.18** | 1.002±0.202 | 1.082±0.206 |
| | RCT-5 | **1.016±0.121** | 1.095±0.123 | 1.210±0.248 |
| | OBS-CPS | **1.031±0.258** | 1.082±0.264 | 1.188±0.345 |
| | OBS-UConf | **0.985±0.34** | 1.015±0.300 | 1.073±0.383 |
| | OBS-NoPos | **0.967±0.238** | 1.03±0.249 | 1.107±0.295 |
| | OBS-CPS-InfC | **1.146±0.030** | 1.148±0.034 | 1.209±0.044 |
| | OBS-UConf-InfC | 1.179±0.024 | **1.172±0.031** | 1.234±0.032 |
| | OBS-NoPos-InfC | **1.169±0.026** | **1.169±0.026** | 1.230±0.028 |
| | Total Win | 48 | 29 | 3 |
| E | RCT-50 | **1.75±0.186** | 1.906±0.207 | 2.124±0.247 |
| | RCT-5 | **1.604±0.125** | 1.731±0.133 | 1.901±0.17 |
| | OBS-CPS | **1.651±0.161** | 1.799±0.201 | 1.990±0.243 |
| | OBS-UConf | **1.630±0.127** | 1.779±0.159 | 1.974±0.212 |
| | OBS-NoPos | **1.698±0.139** | 1.856±0.162 | 2.059±0.202 |
| | OBS-CPS-InfC | **0.917±0.089** | 1.012±0.113 | 1.140±0.141 |
| | OBS-UConf-InfC | **0.968±0.130** | 1.074±0.164 | 1.222±0.237 |
| | OBS-NoPos-InfC | **0.928±0.060** | 1.012±0.063 | 1.130±0.071 |
| | Total Win | 80 | 0 | 0 |

Table 21: T-Learner MAE

| Survival Scenario | Causal Configuration | Base Regression Model (Treated) | | | Base Regression Model (Control) | | |
|---|---|---|---|---|---|---|---|
| | | Lasso Reg. | Random Forest | XGBoost | Lasso Reg. | Random Forest | XGBoost |
| A | RCT-50 | 0.669±0.012 | **0.652±0.013** | 0.680±0.015 | **0.652±0.019** | 0.657±0.019 | 0.685±0.019 |
| | RCT-5 | 0.656±0.044 | **0.641±0.049** | 0.677±0.050 | **0.644±0.011** | 0.650±0.012 | 0.668±0.014 |
| | OBS-CPS | 0.711±0.012 | **0.697±0.012** | 0.727±0.012 | **0.588±0.014** | 0.595±0.014 | 0.621±0.018 |
| | OBS-UConf | **0.640±0.009** | 0.641±0.007 | 0.668±0.005 | **0.558±0.012** | 0.566±0.013 | 0.593±0.015 |
| | OBS-NoPos | 0.568±0.014 | **0.561±0.013** | 0.587±0.016 | **0.733±0.015** | 0.747±0.016 | 0.776±0.016 |
| | OBS-CPS-InfC | 0.799±0.020 | **0.797±0.022** | 0.838±0.023 | **0.646±0.023** | 0.664±0.025 | 0.699±0.029 |
| | OBS-UConf-InfC | **0.723±0.023** | 0.740±0.031 | 0.778±0.030 | **0.614±0.019** | 0.633±0.023 | 0.668±0.025 |
| | OBS-NoPos-InfC | **0.608±0.018** | 0.609±0.016 | 0.642±0.020 | **0.824±0.021** | 0.855±0.021 | 0.895±0.029 |
| | Total Win | 28 | 52 | 0 | 77 | 3 | 0 |
| B | RCT-50 | 0.375±0.012 | **0.350±0.014** | 0.374±0.016 | **0.279±0.007** | 0.281±0.008 | 0.303±0.006 |
| | RCT-5 | 0.393±0.051 | **0.383±0.047** | 0.404±0.043 | **0.271±0.005** | 0.273±0.005 | 0.287±0.006 |
| | OBS-CPS | 0.326±0.014 | **0.302±0.014** | 0.322±0.016 | **0.305±0.011** | 0.313±0.012 | 0.338±0.015 |
| | OBS-UConf | 0.375±0.008 | **0.348±0.009** | 0.371±0.011 | **0.327±0.008** | 0.334±0.007 | 0.363±0.012 |
| | OBS-NoPos | 0.425±0.014 | **0.411±0.015** | 0.442±0.017 | **0.231±0.007** | 0.235±0.007 | 0.253±0.008 |
| | OBS-CPS-InfC | 0.311±0.007 | **0.292±0.006** | 0.311±0.008 | **0.291±0.009** | 0.297±0.010 | 0.322±0.011 |
| | OBS-UConf-InfC | 0.365±0.004 | **0.344±0.007** | 0.370±0.009 | **0.310±0.007** | 0.313±0.007 | 0.337±0.008 |
| | OBS-NoPos-InfC | 0.415±0.009 | **0.406±0.009** | 0.438±0.011 | **0.228±0.005** | 0.233±0.006 | 0.249±0.006 |
| | Total Win | 4 | 76 | 0 | 68 | 12 | 0 |
| C | RCT-50 | **1.534±0.037** | 1.653±0.041 | 1.839±0.046 | **1.387±0.029** | 1.542±0.020 | 1.747±0.027 |
| | RCT-5 | **1.643±0.091** | 1.751±0.085 | 1.935±0.074 | **1.393±0.020** | 1.527±0.02 | 1.688±0.021 |
| | OBS-CPS | **1.484±0.026** | 1.599±0.031 | 1.797±0.036 | **1.379±0.030** | 1.529±0.029 | 1.726±0.033 |
| | OBS-UConf | **1.493±0.037** | 1.617±0.034 | 1.809±0.028 | **1.363±0.019** | 1.528±0.021 | 1.740±0.027 |
| | OBS-NoPos | **1.568±0.040** | 1.665±0.038 | 1.860±0.037 | **1.420±0.037** | 1.565±0.027 | 1.763±0.035 |
| | OBS-CPS-InfC | **0.909±0.047** | 1.002±0.052 | 1.144±0.050 | **0.953±0.042** | 1.040±0.048 | 1.198±0.060 |
| | OBS-UConf-InfC | **0.876±0.039** | 0.951±0.052 | 1.088±0.070 | **0.916±0.044** | 1.004±0.056 | 1.166±0.079 |
| | OBS-NoPos-InfC | **0.902±0.057** | 1.009±0.074 | 1.149±0.088 | **0.928±0.076** | 1.016±0.087 | 1.152±0.088 |
| | Total Win | 79 | 1 | 0 | 80 | 0 | 0 |
| D | RCT-50 | **0.302±0.083** | 0.316±0.093 | 0.339±0.11 | **1.546±0.304** | 1.691±0.484 | 1.767±0.429 |
| | RCT-5 | **0.284±0.044** | 0.296±0.049 | 0.312±0.052 | **1.051±0.127** | 1.118±0.129 | 1.285±0.304 |
| | OBS-CPS | **0.347±0.084** | 0.366±0.087 | 0.384±0.094 | **1.651±0.416** | 1.758±0.443 | 1.924±0.594 |
| | OBS-UConf | **0.328±0.134** | 0.343±0.130 | 0.362±0.138 | **1.691±0.564** | 1.797±0.529 | 1.809±0.705 |
| | OBS-NoPos | **0.334±0.105** | 0.342±0.117 | 0.366±0.129 | **1.571±0.382** | 1.659±0.392 | 1.939±0.612 |
| | OBS-CPS-InfC | 1.138±0.029 | **1.097±0.027** | 1.171±0.028 | **1.147±0.041** | 1.201±0.051 | 1.300±0.056 |
| | OBS-UConf-InfC | 1.206±0.029 | **1.156±0.043** | 1.240±0.044 | **1.146±0.023** | 1.197±0.028 | 1.298±0.029 |
| | OBS-NoPos-InfC | 1.216±0.030 | **1.198±0.035** | 1.297±0.046 | **1.100±0.030** | 1.143±0.037 | 1.222±0.035 |
| | Total Win | 45 | 35 | 0 | 66 | 10 | 4 |
| E | RCT-50 | **1.784±0.230** | 1.955±0.255 | 2.22±0.310 | **1.720±0.154** | 1.899±0.184 | 2.108±0.198 |
| | RCT-5 | **1.607±0.244** | 1.859±0.436 | 1.969±0.335 | **1.605±0.121** | 1.733±0.131 | 1.894±0.147 |
| | OBS-CPS | **1.670±0.194** | 1.852±0.261 | 2.092±0.312 | **1.637±0.141** | 1.785±0.158 | 1.993±0.200 |
| | OBS-UConf | **1.650±0.145** | 1.809±0.167 | 2.053±0.239 | **1.616±0.130** | 1.763±0.144 | 1.982±0.201 |
| | OBS-NoPos | **1.740±0.161** | 1.907±0.200 | 2.198±0.338 | **1.663±0.140** | 1.804±0.148 | 2.027±0.167 |
| | OBS-CPS-InfC | **0.911±0.111** | 1.007±0.123 | 1.146±0.140 | **0.925±0.074** | 1.015±0.096 | 1.158±0.104 |
| | OBS-UConf-InfC | **0.953±0.107** | 1.049±0.118 | 1.221±0.146 | **0.987±0.159** | 1.103±0.222 | 1.271±0.228 |
| | OBS-NoPos-InfC | **0.949±0.085** | 1.043±0.095 | 1.221±0.112 | **0.908±0.046** | 1.000±0.044 | 1.136±0.064 |
| | Total Win | 80 | 0 | 0 | 80 | 0 | 0 |

Table 22: X-Learner MAE

| Survival Scenario | Causal Configuration | Base Regression Model (Treated) | | | Base Regression Model (Control) | | |
|---|---|---|---|---|---|---|---|
| | | Lasso Reg. | Random Forest | XGBoost | Lasso Reg. | Random Forest | XGBoost |
| A | RCT-50 | **0.620±0.011** | 0.622±0.013 | 0.631±0.013 | **0.624±0.019** | **0.624±0.019** | 0.631±0.018 |
| | RCT-5 | **0.613±0.048** | 0.624±0.056 | 0.634±0.050 | 0.618±0.012 | **0.617±0.012** | 0.623±0.010 |
| | OBS-CPS | **0.664±0.011** | 0.667±0.010 | 0.675±0.012 | **0.561±0.013** | 0.562±0.014 | 0.568±0.013 |
| | OBS-UConf | **0.608±0.008** | 0.610±0.008 | 0.616±0.008 | **0.530±0.012** | 0.531±0.012 | 0.538±0.012 |
| | OBS-NoPos | **0.532±0.013** | 0.533±0.013 | 0.539±0.015 | **0.711±0.014** | 0.712±0.014 | 0.718±0.015 |
| | OBS-CPS-InfC | **0.747±0.018** | 0.751±0.018 | 0.763±0.019 | **0.616±0.021** | **0.616±0.022** | 0.623±0.022 |
| | OBS-UConf-InfC | **0.689±0.024** | 0.691±0.023 | 0.698±0.024 | **0.583±0.019** | 0.585±0.020 | 0.592±0.019 |
| | OBS-NoPos-InfC | **0.570±0.016** | **0.570±0.017** | 0.578±0.018 | **0.800±0.021** | 0.802±0.021 | 0.809±0.020 |
| | Total Win | 64 | 16 | 0 | 47 | 33 | 0 |
| B | RCT-50 | 0.339±0.012 | **0.328±0.012** | 0.333±0.012 | 0.265±0.007 | **0.261±0.007** | 0.264±0.007 |
| | RCT-5 | 0.365±0.045 | **0.364±0.051** | 0.368±0.046 | 0.256±0.005 | **0.253±0.004** | 0.255±0.004 |
| | OBS-CPS | 0.296±0.014 | **0.283±0.013** | 0.287±0.013 | **0.290±0.010** | **0.290±0.012** | 0.291±0.011 |
| | OBS-UConf | 0.339±0.009 | **0.327±0.008** | 0.331±0.008 | 0.312±0.008 | **0.309±0.006** | 0.311±0.008 |
| | OBS-NoPos | 0.396±0.013 | **0.387±0.014** | 0.393±0.013 | 0.221±0.006 | **0.217±0.007** | 0.219±0.007 |
| | OBS-CPS-InfC | 0.283±0.006 | **0.272±0.007** | 0.275±0.006 | 0.277±0.008 | **0.275±0.008** | 0.278±0.009 |
| | OBS-UConf-InfC | 0.332±0.005 | **0.321±0.006** | 0.326±0.006 | 0.295±0.008 | **0.291±0.007** | 0.293±0.007 |
| | OBS-NoPos-InfC | 0.388±0.008 | **0.379±0.008** | 0.386±0.008 | 0.219±0.005 | **0.215±0.005** | 0.216±0.005 |
| | Total Win | 3 | 73 | 4 | 5 | 70 | 5 |
| C | RCT-50 | 1.542±0.041 | 1.547±0.032 | **1.533±0.033** | 1.417±0.025 | 1.422±0.023 | **1.403±0.026** |
| | RCT-5 | 1.651±0.092 | 1.662±0.091 | **1.640±0.087** | 1.411±0.018 | 1.414±0.019 | **1.400±0.019** |
| | OBS-CPS | 1.492±0.031 | 1.492±0.031 | **1.481±0.026** | 1.409±0.032 | 1.413±0.030 | **1.394±0.031** |
| | OBS-UConf | 1.509±0.038 | 1.510±0.039 | **1.497±0.042** | 1.396±0.016 | 1.402±0.014 | **1.385±0.016** |
| | OBS-NoPos | 1.562±0.042 | 1.563±0.038 | **1.561±0.038** | 1.445±0.033 | 1.449±0.035 | **1.433±0.037** |
| | OBS-CPS-InfC | **0.909±0.047** | 0.915±0.048 | **0.909±0.046** | **0.952±0.043** | 0.959±0.041 | 0.954±0.043 |
| | OBS-UConf-InfC | 0.876±0.039 | 0.879±0.042 | **0.875±0.039** | **0.916±0.044** | 0.921±0.045 | 0.919±0.047 |
| | OBS-NoPos-InfC | **0.902±0.058** | 0.909±0.061 | 0.904±0.060 | **0.928±0.076** | 0.935±0.081 | **0.928±0.074** |
| | Total Win | 27 | 11 | 42 | 17 | 7 | 56 |
| D | RCT-50 | 0.306±0.088 | 0.300±0.089 | **0.292±0.083** | 1.633±0.342 | 1.613±0.503 | **1.506±0.302** |
| | RCT-5 | 0.283±0.046 | 0.280±0.042 | **0.277±0.047** | 1.124±0.125 | 1.055±0.128 | **1.033±0.150** |
| | OBS-CPS | 0.355±0.085 | 0.343±0.080 | **0.334±0.081** | 1.819±0.545 | 1.681±0.456 | **1.613±0.436** |
| | OBS-UConf | 0.336±0.137 | 0.324±0.126 | **0.322±0.123** | 1.772±0.571 | 1.737±0.498 | **1.613±0.557** |
| | OBS-NoPos | 0.336±0.110 | 0.322±0.106 | **0.319±0.104** | 1.688±0.418 | 1.607±0.383 | **1.540±0.388** |
| | OBS-CPS-InfC | 1.067±0.026 | **1.043±0.025** | 1.049±0.026 | **1.133±0.041** | 1.137±0.043 | 1.136±0.043 |
| | OBS-UConf-InfC | 1.116±0.032 | **1.098±0.033** | 1.103±0.027 | **1.135±0.021** | 1.138±0.021 | 1.136±0.021 |
| | OBS-NoPos-InfC | 1.160±0.029 | **1.137±0.029** | 1.140±0.029 | **1.085±0.032** | 1.088±0.032 | 1.087±0.031 |
| | Total Win | 4 | 36 | 40 | 17 | 21 | 42 |
| E | RCT-50 | **1.771±0.226** | 1.798±0.237 | 1.772±0.230 | 1.734±0.154 | 1.748±0.156 | **1.721±0.150** |
| | RCT-5 | **1.607±0.240** | 1.731±0.366 | 1.608±0.245 | 1.602±0.120 | 1.606±0.121 | **1.597±0.118** |
| | OBS-CPS | **1.668±0.199** | 1.722±0.244 | 1.668±0.197 | 1.638±0.138 | 1.652±0.139 | **1.633±0.141** |
| | OBS-UConf | 1.661±0.157 | 1.659±0.151 | **1.646±0.152** | **1.613±0.130** | 1.632±0.132 | 1.635±0.187 |
| | OBS-NoPos | **1.726±0.162** | 1.753±0.176 | 1.773±0.226 | 1.670±0.137 | 1.671±0.135 | **1.656±0.136** |
| | OBS-CPS-InfC | **0.911±0.111** | 0.919±0.108 | 0.913±0.107 | **0.925±0.074** | 0.930±0.081 | 0.924±0.075 |
| | OBS-UConf-InfC | **0.953±0.107** | 0.960±0.114 | **0.953±0.106** | 0.987±0.159 | 1.008±0.209 | **0.986±0.151** |
| | OBS-NoPos-InfC | 0.949±0.085 | **0.947±0.077** | 0.951±0.083 | **0.908±0.046** | 0.923±0.043 | 0.910±0.047 |
| | Total Win | 37 | 14 | 29 | 26 | 11 | 43 |

Table 23: DR-Learner MAE

| Survival Scenario | Causal Configuration | Base Regression Model | | |
|---|---|---|---|---|
| | | Lasso Reg. | Random Forest | XGBoost |
| A | RCT-50 | 0.661±0.012 | **0.658±0.013** | 0.685±0.013 |
| | RCT-5 | **0.645±0.011** | 0.652±0.013 | 0.680±0.014 |
| | OBS-CPS | 0.653±0.011 | **0.649±0.012** | 0.676±0.012 |
| | OBS-UConf | **0.604±0.009** | 0.611±0.009 | 0.635±0.010 |
| | OBS-NoPos | 0.657±0.010 | **0.656±0.011** | 0.684±0.014 |
| | OBS-CPS-InfC | **0.727±0.018** | 0.735±0.023 | 0.770±0.025 |
| | OBS-UConf-InfC | **0.675±0.019** | 0.696±0.022 | 0.730±0.023 |
| | OBS-NoPos-InfC | **0.724±0.014** | 0.735±0.015 | 0.770±0.020 |
| | Total Win | 54 | 26 | 0 |
| B | RCT-50 | 0.330±0.008 | **0.318±0.011** | 0.341±0.015 |
| | RCT-5 | **0.278±0.006** | **0.278±0.006** | 0.300±0.007 |
| | OBS-CPS | 0.316±0.011 | **0.308±0.013** | 0.332±0.017 |
| | OBS-UConf | 0.354±0.007 | **0.344±0.008** | 0.370±0.010 |
| | OBS-NoPos | 0.345±0.009 | **0.326±0.011** | 0.349±0.012 |
| | OBS-CPS-InfC | 0.301±0.007 | **0.295±0.006** | 0.316±0.008 |
| | OBS-UConf-InfC | 0.340±0.004 | **0.331±0.004** | 0.355±0.006 |
| | OBS-NoPos-InfC | 0.337±0.005 | **0.321±0.005** | 0.345±0.007 |
| | Total Win | 6 | 74 | 0 |
| C | RCT-50 | **1.430±0.022** | 1.591±0.021 | 1.791±0.024 |
| | RCT-5 | **1.404±0.019** | 1.540±0.018 | 1.742±0.021 |
| | OBS-CPS | **1.410±0.021** | 1.564±0.026 | 1.763±0.021 |
| | OBS-UConf | **1.420±0.023** | 1.572±0.024 | 1.777±0.019 |
| | OBS-NoPos | **1.454±0.019** | 1.614±0.020 | 1.805±0.015 |
| | OBS-CPS-InfC | **0.931±0.027** | 1.021±0.030 | 1.173±0.038 |
| | OBS-UConf-InfC | **0.895±0.033** | 0.977±0.044 | 1.133±0.054 |
| | OBS-NoPos-InfC | **0.916±0.057** | 1.009±0.065 | 1.172±0.083 |
| | Total Win | 80 | 0 | 0 |
| D | RCT-50 | **0.943±0.180** | 0.986±0.187 | 1.096±0.253 |
| | RCT-5 | **1.020±0.120** | 1.095±0.118 | 1.177±0.176 |
| | OBS-CPS | **1.031±0.256** | 1.103±0.259 | 1.158±0.339 |
| | OBS-UConf | **0.986±0.340** | 1.024±0.345 | 1.119±0.357 |
| | OBS-NoPos | **0.969±0.238** | 1.038±0.292 | 1.124±0.243 |
| | OBS-CPS-InfC | **1.146±0.030** | 1.154±0.037 | 1.238±0.048 |
| | OBS-UConf-InfC | 1.179±0.024 | **1.176±0.031** | 1.262±0.034 |
| | OBS-NoPos-InfC | **1.169±0.026** | 1.173±0.026 | 1.260±0.028 |
| | Total Win | 65 | 15 | 0 |
| E | RCT-50 | **1.753±0.185** | 1.917±0.206 | 2.161±0.243 |
| | RCT-5 | **1.605±0.127** | 1.742±0.144 | 1.949±0.161 |
| | OBS-CPS | **1.652±0.163** | 1.811±0.199 | 2.033±0.241 |
| | OBS-UConf | **1.632±0.127** | 1.785±0.166 | 2.041±0.255 |
| | OBS-NoPos | **1.701±0.139** | 1.859±0.162 | 2.112±0.202 |
| | OBS-CPS-InfC | **0.917±0.089** | 1.016±0.104 | 1.164±0.114 |
| | OBS-UConf-InfC | **0.969±0.130** | 1.080±0.158 | 1.276±0.240 |
| | OBS-NoPos-InfC | **0.928±0.060** | 1.022±0.068 | 1.182±0.096 |
| | Total Win | 80 | 0 | 0 |

Table 24: DR-Learner propensity score AUC. Note that we use the `econml` package in Python, which by default uses logistic regression for predicting the treatment assignment. Thus, we report the AUC of the treatment prediction by the logistic regression.

| Causal Configuration | Logistic Regression |
|---|---|
| RCT-50 | 0.501±0.005 |
| RCT-5 | 0.497±0.011 |
| OBS-CPS | 0.661±0.007 |
| OBS-UConf | 0.548±0.007 |
| OBS-NoPos | 0.820±0.005 |
| OBS-CPS-InfC | 0.661±0.007 |
| OBS-UConf-InfC | 0.548±0.007 |
| OBS-NoPos-InfC | 0.820±0.005 |

### F.6.3 BASE SURVIVAL LEARNER EVALUATION

See Table 25, 26, 27 for time-dependent concordance index on different base survival learners by the base survival learners for S-, T-, matching-learners. We report the mean and standard deviation across 10 experimental repeats with different random splits.

Table 25: Survival S-Learner concordance index

| Survival Scenario | Causal Configuration | Base Regression Model | | |
|---|---|---|---|---|
| | | RSF | DeepSurv | DeepHit |
| A | RCT-50 | 0.568±0.008 | **0.595±0.003** | 0.557±0.007 |
| | RCT-5 | 0.551±0.008 | **0.580±0.004** | 0.558±0.006 |
| | OBS-CPS | 0.565±0.004 | **0.596±0.004** | 0.567±0.008 |
| | OBS-UConf | 0.556±0.005 | **0.587±0.006** | 0.558±0.010 |
| | OBS-NoPos | 0.565±0.009 | **0.594±0.004** | 0.553±0.006 |
| | OBS-CPS-InfC | 0.563±0.005 | **0.597±0.004** | 0.546±0.010 |
| | OBS-UConf-InfC | 0.557±0.006 | **0.585±0.006** | 0.538±0.008 |
| | OBS-NoPos-InfC | 0.562±0.006 | **0.591±0.003** | 0.539±0.008 |
| | Total Win | 0 | 80 | 0 |
| B | RCT-50 | 0.640±0.003 | **0.645±0.004** | **0.645±0.004** |
| | RCT-5 | 0.616±0.003 | **0.622±0.005** | 0.621±0.004 |
| | OBS-CPS | 0.631±0.005 | **0.632±0.003** | 0.631±0.003 |
| | OBS-UConf | 0.632±0.005 | **0.634±0.005** | **0.634±0.004** |
| | OBS-NoPos | 0.650±0.003 | **0.656±0.002** | **0.656±0.002** |
| | OBS-CPS-InfC | 0.630±0.004 | **0.632±0.004** | 0.629±0.003 |
| | OBS-UConf-InfC | 0.630±0.004 | **0.633±0.005** | 0.631±0.005 |
| | OBS-NoPos-InfC | 0.649±0.003 | **0.655±0.003** | 0.654±0.003 |
| | Total Win | 10 | 50 | 20 |
| C | RCT-50 | 0.545±0.009 | **0.576±0.004** | 0.570±0.005 |
| | RCT-5 | 0.522±0.007 | **0.554±0.007** | 0.540±0.014 |
| | OBS-CPS | 0.538±0.006 | **0.573±0.005** | 0.562±0.004 |
| | OBS-UConf | 0.536±0.007 | **0.566±0.007** | 0.561±0.008 |
| | OBS-NoPos | 0.550±0.007 | **0.583±0.005** | 0.575±0.007 |
| | OBS-CPS-InfC | 0.498±0.015 | **0.558±0.026** | 0.546±0.017 |
| | OBS-UConf-InfC | 0.502±0.023 | **0.560±0.029** | 0.541±0.020 |
| | OBS-NoPos-InfC | 0.511±0.029 | **0.586±0.019** | 0.561±0.023 |
| | Total Win | 0 | 70 | 10 |
| D | RCT-50 | 0.633±0.027 | 0.676±0.021 | **0.696±0.013** |
| | RCT-5 | 0.569±0.019 | 0.626±0.017 | **0.628±0.011** |
| | OBS-CPS | 0.610±0.029 | 0.668±0.019 | **0.683±0.011** |
| | OBS-UConf | 0.634±0.027 | **0.702±0.015** | 0.696±0.018 |
| | OBS-NoPos | 0.615±0.032 | 0.678±0.016 | **0.683±0.015** |
| | OBS-CPS-InfC | 0.626±0.011 | **0.634±0.005** | 0.629±0.007 |
| | OBS-UConf-InfC | 0.639±0.005 | **0.646±0.005** | 0.643±0.007 |
| | OBS-NoPos-InfC | 0.635±0.006 | **0.644±0.006** | 0.640±0.005 |
| | Total Win | 4 | 40 | 36 |
| E | RCT-50 | 0.544±0.010 | **0.591±0.011** | 0.578±0.011 |
| | RCT-5 | 0.513±0.009 | **0.554±0.015** | 0.547±0.012 |
| | OBS-CPS | 0.538±0.013 | **0.583±0.010** | 0.566±0.018 |
| | OBS-UConf | 0.533±0.016 | **0.574±0.018** | 0.567±0.017 |
| | OBS-NoPos | 0.544±0.015 | **0.599±0.010** | 0.589±0.012 |
| | OBS-CPS-InfC | 0.482±0.041 | **0.546±0.030** | 0.538±0.028 |
| | OBS-UConf-InfC | 0.445±0.029 | **0.542±0.045** | 0.534±0.017 |
| | OBS-NoPos-InfC | 0.474±0.017 | **0.565±0.035** | 0.563±0.022 |
| | Total Win | 0 | 60 | 20 |

Table 26: Survival T-Learner concordance index

| Survival Scenario | Causal Configuration | Base Regression Model (Treated) | | | Base Regression Model (Control) | | |
|---|---|---|---|---|---|---|---|
| | | RSF | DeepSurv | DeepHit | RSF | DeepSurv | DeepHit |
| A | RCT-50 | 0.579±0.009 | **0.612±0.006** | 0.581±0.015 | 0.546±0.010 | **0.578±0.007** | 0.549±0.014 |
| | RCT-5 | 0.567±0.031 | **0.604±0.018** | 0.592±0.025 | 0.549±0.007 | **0.581±0.005** | 0.557±0.012 |
| | OBS-CPS | 0.569±0.008 | **0.603±0.009** | 0.582±0.011 | 0.546±0.006 | **0.577±0.007** | 0.548±0.009 |
| | OBS-UConf | 0.546±0.008 | **0.579±0.010** | 0.553±0.012 | 0.557±0.009 | **0.585±0.007** | 0.554±0.009 |
| | OBS-NoPos | 0.567±0.009 | **0.598±0.008** | 0.564±0.016 | 0.534±0.006 | **0.564±0.009** | 0.544±0.005 |
| | OBS-CPS-InfC | 0.569±0.006 | **0.602±0.009** | 0.559±0.018 | 0.546±0.007 | **0.578±0.006** | 0.541±0.017 |
| | OBS-UConf-InfC | 0.546±0.009 | **0.578±0.008** | 0.540±0.012 | 0.555±0.012 | **0.584±0.006** | 0.538±0.019 |
| | OBS-NoPos-InfC | 0.564±0.009 | **0.598±0.006** | 0.545±0.013 | 0.531±0.009 | **0.564±0.007** | 0.537±0.010 |
| | Total Win | 0 | 75 | 5 | 0 | 80 | 0 |
| B | RCT-50 | 0.651±0.004 | **0.656±0.004** | 0.654±0.005 | 0.610±0.005 | **0.618±0.005** | 0.616±0.005 |
| | RCT-5 | **0.628±0.017** | 0.627±0.027 | 0.609±0.022 | 0.610±0.007 | 0.619±0.004 | **0.620±0.004** |
| | OBS-CPS | 0.630±0.006 | **0.637±0.005** | 0.634±0.005 | 0.605±0.006 | **0.612±0.006** | 0.607±0.007 |
| | OBS-UConf | 0.644±0.008 | **0.648±0.006** | 0.646±0.006 | 0.598±0.005 | **0.605±0.007** | 0.602±0.008 |
| | OBS-NoPos | 0.628±0.005 | **0.636±0.004** | 0.631±0.005 | 0.593±0.008 | **0.601±0.004** | 0.600±0.007 |
| | OBS-CPS-InfC | 0.628±0.005 | **0.633±0.005** | 0.632±0.003 | 0.603±0.006 | **0.611±0.008** | 0.610±0.007 |
| | OBS-UConf-InfC | 0.642±0.004 | 0.645±0.005 | **0.646±0.005** | 0.598±0.009 | **0.605±0.004** | 0.603±0.007 |
| | OBS-NoPos-InfC | 0.624±0.007 | **0.634±0.005** | 0.628±0.005 | 0.592±0.006 | **0.602±0.004** | 0.600±0.007 |
| | Total Win | 14 | 44 | 22 | 7 | 51 | 22 |
| C | RCT-50 | 0.532±0.015 | **0.565±0.013** | 0.557±0.015 | 0.512±0.008 | **0.541±0.011** | 0.524±0.010 |
| | RCT-5 | 0.536±0.031 | 0.541±0.050 | **0.544±0.027** | 0.518±0.014 | **0.547±0.010** | 0.541±0.007 |
| | OBS-CPS | 0.536±0.008 | **0.568±0.009** | 0.555±0.011 | 0.516±0.007 | **0.540±0.011** | 0.523±0.009 |
| | OBS-UConf | 0.537±0.010 | **0.568±0.012** | 0.557±0.007 | 0.509±0.010 | **0.533±0.012** | 0.521±0.012 |
| | OBS-NoPos | 0.524±0.016 | **0.555±0.012** | 0.543±0.014 | 0.521±0.012 | **0.547±0.011** | 0.535±0.008 |
| | OBS-CPS-InfC | 0.487±0.041 | **0.550±0.032** | 0.542±0.038 | 0.497±0.018 | **0.537±0.021** | 0.522±0.033 |
| | OBS-UConf-InfC | 0.491±0.044 | **0.552±0.031** | 0.550±0.032 | 0.484±0.021 | 0.513±0.035 | **0.519±0.018** |
| | OBS-NoPos-InfC | 0.470±0.045 | **0.549±0.037** | 0.544±0.032 | 0.489±0.029 | **0.538±0.023** | 0.513±0.033 |
| | Total Win | 3 | 57 | 20 | 0 | 64 | 16 |
| D | RCT-50 | 0.646±0.038 | 0.683±0.084 | **0.727±0.024** | 0.565±0.025 | 0.614±0.018 | **0.623±0.025** |
| | RCT-5 | 0.447±0.174 | 0.412±0.158 | **0.672±0.135** | 0.573±0.019 | **0.626±0.016** | 0.625±0.015 |
| | OBS-CPS | 0.584±0.052 | 0.646±0.038 | **0.672±0.022** | 0.543±0.023 | 0.609±0.032 | **0.620±0.024** |
| | OBS-UConf | 0.655±0.038 | 0.731±0.028 | **0.745±0.012** | 0.536±0.034 | 0.588±0.036 | **0.597±0.029** |
| | OBS-NoPos | 0.668±0.051 | 0.658±0.085 | **0.773±0.024** | 0.547±0.027 | **0.593±0.019** | 0.586±0.021 |
| | OBS-CPS-InfC | 0.632±0.010 | **0.639±0.005** | 0.637±0.006 | 0.556±0.010 | **0.575±0.010** | 0.566±0.010 |
| | OBS-UConf-InfC | 0.673±0.005 | 0.675±0.007 | **0.676±0.007** | 0.549±0.011 | **0.569±0.007** | 0.556±0.008 |
| | OBS-NoPos-InfC | 0.664±0.009 | **0.671±0.007** | 0.668±0.007 | 0.549±0.007 | **0.563±0.006** | 0.553±0.009 |
| | Total Win | 5 | 25 | 50 | 2 | 49 | 29 |
| E | RCT-50 | 0.539±0.020 | **0.589±0.024** | 0.575±0.017 | 0.514±0.020 | **0.547±0.019** | 0.537±0.014 |
| | RCT-5 | 0.481±0.065 | **0.518±0.047** | 0.516±0.065 | 0.518±0.011 | **0.554±0.010** | 0.544±0.013 |
| | OBS-CPS | 0.533±0.021 | **0.574±0.022** | 0.562±0.020 | 0.508±0.018 | **0.544±0.015** | 0.535±0.014 |
| | OBS-UConf | 0.534±0.023 | **0.587±0.024** | 0.552±0.014 | 0.510±0.014 | 0.520±0.023 | **0.531±0.022** |
| | OBS-NoPos | 0.520±0.024 | 0.539±0.032 | **0.547±0.024** | 0.516±0.020 | **0.546±0.016** | 0.534±0.015 |
| | OBS-CPS-InfC | 0.485±0.047 | **0.551±0.042** | 0.520±0.034 | 0.454±0.038 | **0.515±0.045** | 0.508±0.042 |
| | OBS-UConf-InfC | 0.437±0.064 | 0.525±0.048 | **0.541±0.065** | 0.455±0.025 | 0.495±0.037 | **0.499±0.041** |
| | OBS-NoPos-InfC | 0.464±0.046 | **0.520±0.038** | 0.505±0.040 | 0.453±0.043 | 0.514±0.027 | **0.537±0.023** |
| | Total Win | 2 | 53 | 25 | 5 | 43 | 32 |

Table 27: Survival Matching-Learner concordance index

| Survival Scenario | Causal Configuration | Base Survival Model | | |
|---|---|---|---|---|
| | | RSF | DeepSurv | DeepHit |
| A | RCT-50 | 0.568±0.008 | **0.595±0.003** | 0.557±0.007 |
| | RCT-5 | 0.551±0.008 | **0.580±0.004** | 0.558±0.006 |
| | OBS-CPS | 0.556±0.005 | **0.596±0.004** | 0.567±0.008 |
| | OBS-UConf | 0.556±0.005 | **0.587±0.006** | 0.558±0.010 |
| | OBS-NoPos | 0.565±0.009 | **0.594±0.004** | 0.553±0.006 |
| | OBS-CPS-InfC | 0.563±0.005 | **0.597±0.004** | 0.546±0.010 |
| | OBS-UConf-InfC | 0.557±0.006 | **0.585±0.006** | 0.538±0.008 |
| | OBS-NoPos-InfC | 0.562±0.006 | **0.591±0.003** | 0.539±0.008 |
| | Total Win | 0 | 80 | 0 |
| B | RCT-50 | 0.640±0.003 | **0.645±0.004** | **0.645±0.004** |
| | RCT-5 | 0.616±0.003 | **0.622±0.005** | 0.621±0.004 |
| | OBS-CPS | 0.631±0.005 | **0.632±0.003** | 0.631±0.003 |
| | OBS-UConf | 0.632±0.005 | **0.634±0.005** | **0.634±0.004** |
| | OBS-NoPos | 0.650±0.003 | **0.656±0.002** | **0.656±0.002** |
| | OBS-CPS-InfC | 0.630±0.004 | **0.632±0.004** | 0.629±0.003 |
| | OBS-UConf-InfC | 0.630±0.004 | **0.633±0.005** | 0.631±0.005 |
| | OBS-NoPos-InfC | 0.649±0.003 | **0.655±0.003** | 0.654±0.003 |
| | Total Win | 9 | 50 | 21 |
| C | RCT-50 | 0.545±0.009 | **0.576±0.004** | 0.570±0.005 |
| | RCT-5 | 0.522±0.007 | **0.554±0.007** | 0.540±0.014 |
| | OBS-CPS | 0.538±0.006 | **0.573±0.005** | 0.562±0.004 |
| | OBS-UConf | 0.536±0.007 | **0.566±0.007** | 0.561±0.008 |
| | OBS-NoPos | 0.550±0.007 | **0.583±0.005** | 0.575±0.007 |
| | OBS-CPS-InfC | 0.498±0.015 | **0.558±0.026** | 0.546±0.017 |
| | OBS-UConf-InfC | 0.502±0.023 | **0.560±0.029** | 0.541±0.020 |
| | OBS-NoPos-InfC | 0.511±0.029 | **0.586±0.019** | 0.561±0.023 |
| | Total Win | 0 | 70 | 10 |
| D | RCT-50 | 0.633±0.027 | 0.676±0.021 | **0.696±0.013** |
| | RCT-5 | 0.569±0.019 | 0.626±0.017 | **0.628±0.011** |
| | OBS-CPS | 0.610±0.029 | 0.668±0.019 | **0.683±0.011** |
| | OBS-UConf | 0.634±0.027 | **0.702±0.015** | 0.696±0.018 |
| | OBS-NoPos | 0.615±0.032 | 0.678±0.016 | **0.683±0.015** |
| | OBS-CPS-InfC | 0.626±0.011 | **0.634±0.005** | 0.629±0.007 |
| | OBS-UConf-InfC | 0.639±0.005 | **0.646±0.005** | 0.643±0.007 |
| | OBS-NoPos-InfC | 0.635±0.006 | **0.644±0.006** | 0.640±0.005 |
| | Total Win | 4 | 40 | 36 |
| E | RCT-50 | 0.544±0.010 | **0.591±0.011** | 0.578±0.011 |
| | RCT-5 | 0.513±0.009 | **0.554±0.015** | 0.547±0.012 |
| | OBS-CPS | 0.538±0.013 | **0.583±0.010** | 0.566±0.018 |
| | OBS-UConf | 0.533±0.016 | **0.574±0.018** | 0.567±0.017 |
| | OBS-NoPos | 0.544±0.015 | **0.599±0.010** | 0.589±0.012 |
| | OBS-CPS-InfC | 0.482±0.041 | **0.546±0.030** | 0.538±0.028 |
| | OBS-UConf-InfC | 0.445±0.029 | **0.542±0.045** | 0.534±0.017 |
| | OBS-NoPos-InfC | 0.474±0.017 | **0.565±0.035** | 0.563±0.022 |
| | Total Win | 0 | 60 | 20 |

## F.7 CONVERGENCE RESULTS

Figure 18 presents the convergence behavior of different causal inference methods under eight configurations of assumptions, all within Scenario C which was the main focus in Section 4.1. The x-axis shows increasing training set sizes (ranging from 50 to 10,000), while the y-axis plots the root mean squared error (RMSE) of the estimated CATE on the test set. (Note that, all models are selected based on performance on the validation set).

Across all configurations, we observe general convergence trends where CATE RMSE decreases as training size increases. Among the survival methods, the T-Learner-Survival consistently converges the slowest, especially under small training sizes. This may be due to the model requiring sufficient uncensored samples per treatment arm to function effectively. Double-ML also tends to require more data to stabilize, particularly in the presence of low treatment rate or lack of positivity. The Causal Survival Forests shows slower convergence under settings with non-ignorable censoring or positivity violations, reflecting its convergence sensitivity to these assumptions despite its nonparametric structure. Overall, while standard meta-learners and tree-based methods show relatively stable convergence behavior, survival-specific adaptations appear more data-hungry and assumption-sensitive for convergence. These trends highlight the importance of choosing appropriately robust methods with respect to the dataset size in practice, especially in real-world settings where assumptions like positivity or ignorability may be compromised.

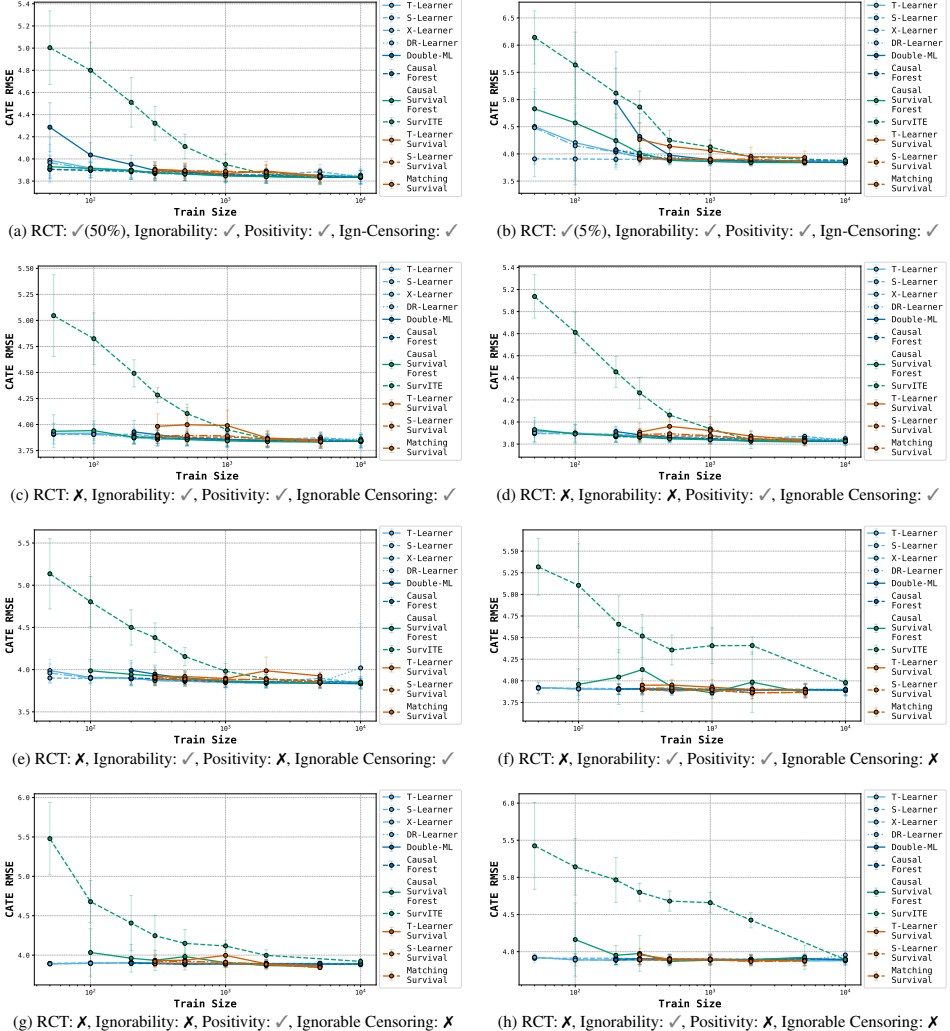

Figure 18: Convergence properties: CATE RMSE in Scenario C as number of training data increases.

# G  SEMI-SYNTHETIC DATASETS: SETUP AND ADDITIONAL RESULTS

## G.1  SEMI-SYNTHETIC DATASETS SETUP

To complement synthetic benchmarks and real-world case studies, we construct semi-synthetic datasets that pair real covariates with simulated treatment assignment and survival outcomes. This strategy preserves realistic covariate distributions and correlations while enabling controlled evaluation against known ground-truth CATEs. We consider two covariate sources: the ACTG HIV trial and MIMIC-IV ICU records. Table 28 summarizes dataset sizes, covariate counts, treatment and censoring rates, and whether the treatment-assignment and event-time mechanisms depend on covariates through linear versus non-linear functions; full generative details are provided in Appendix G.2 and G.3. Figure 19 shows Kaplan–Meier curves for event and censoring in the treatment and control groups for each semi-synthetic dataset.

Table 28: Semi-synthetic datasets overview. The last two columns summarize how treatment assignment and event-time generation depend on covariates. linear indicates dependence through a linear predictor with no quadratic or interaction terms; non-linear includes quadratic and/or interaction terms. independent indicates covariate-independent treatment assignment.

|  | Data size | No. covariates | Treatment rate | Censoring. rate | Treatment assignment mechanism | Event time mechanism |
|---|---|---|---|---|---|---|
| ACTG | 2,139 | 23 | 56.15% | 51.19% | linear | non-linear |
| MIMIC-$i$ | 25,170 | 36 | 49.92% | 88.49% | independent | linear |
| MIMIC-$ii$ | 25,170 | 36 | 49.92% | 81.65% | independent | linear |
| MIMIC-$iii$ | 25,170 | 36 | 49.92% | 74.10% | independent | linear |
| MIMIC-$iv$ | 25,170 | 36 | 49.92% | 66.34% | independent | linear |
| MIMIC-$v$ | 25,170 | 36 | 49.92% | 53.35% | independent | linear |
| MIMIC-$vi$ | 25,170 | 36 | 53.54% | 53.07% | linear | linear |
| MIMIC-$vii$ | 25,170 | 36 | 54.26% | 52.66% | linear | non-linear |
| MIMIC-$viii$ | 25,170 | 36 | 50.94% | 53.26% | non-linear | linear |
| MIMIC-$ix$ | 25,170 | 36 | 51.47% | 52.82% | non-linear | non-linear |

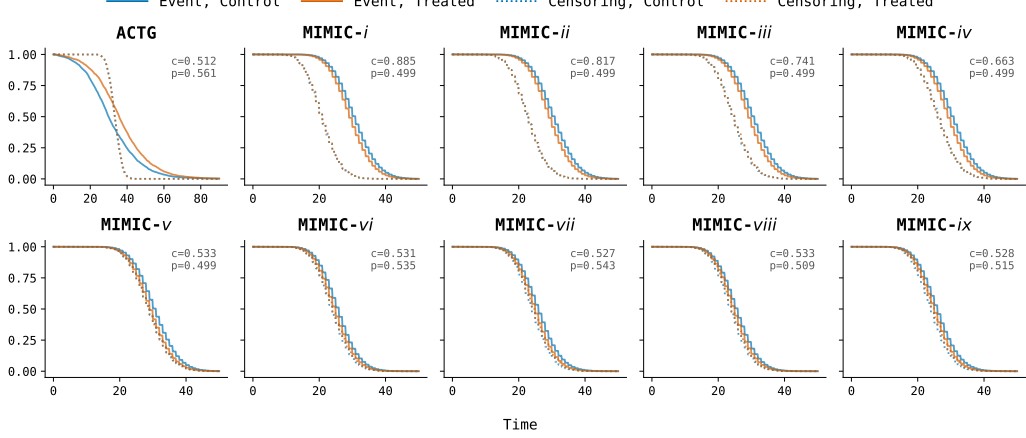

Figure 19: (Semi-synthetic datasets) Kaplan-Meier curves. Solid lines show event-time survival under control (blue) and treatment (orange); dotted lines show censoring-time survival for each arm. Each panel reports the empirical censoring rate $c$ and treatment probability $p$.

## G.2 ACTG SEMI-SYNTHETIC DATASET

The ACTG semi-synthetic dataset is derived from the ACTG 175 HIV clinical trial (Hammer et al., 1996), which contains 23 baseline covariates. Following the construction procedure of Chapfuwa et al. (2021), we simulate covariate-dependent treatment assignment and generate event times from a Gompertz–Cox model with an AFT-style censoring mechanism. This dataset exhibits realistic treatment imbalance and moderate censoring ($\sim 51\%$), providing a clinically grounded benchmark with preserved trial covariate structure.

More concretely, following Chapfuwa et al. (2021), we generate:

$$X = \text{ACTG covariates}$$

$$P(A = 1|X = x) = \frac{1}{b} \times (a + \sigma\left(\eta(\text{AGE} - \mu_{\text{AGE}} + \text{CD40} - \mu_{\text{CD40}}))\right)$$

$$U \sim \text{Uniform}(0, 1)$$

$$T_A = \frac{1}{\alpha_A} \log\left[1 - \frac{\alpha_A \log U}{\lambda_A \exp\left(x^T \beta_A\right)}\right]$$

$$\log C \sim \text{Normal}(\mu_c, \sigma_c^2)$$

$$Y = \min(T_A, C)$$

where $\sigma(\cdot)$ is the sigmoid function, $\{\beta_A, \alpha_A, \lambda_A, b, a, \eta, \mu_c, \sigma_c\}$ are hyper-parameters taken to be the same as described at `https://github.com/paidamoyo/counterfactual_survival_analysis` and $\{\mu_{\text{AGE}}, \mu_{\text{CD40}}\}$ are the means for age and CD40 respectively.

### G.3  MIMIC SEMI-SYNTHETIC DATASETS

We construct a suite of nine semi-synthetic datasets derived from MIMIC-IV ICU records (Johnson et al., 2023). All variants share the same covariates and sample size, and differ only in the treatment-assignment and event/censoring mechanisms. The suite is organized into two complementary subsets: (1) *independent assignment with varying censoring severity* (MIMIC-$i$–$v$), which isolates robustness to censoring under randomized treatment; and (2) *confounded assignment with varying functional form* (MIMIC-$vi$–$ix$), which evaluates robustness to observed confounding and to non-linear (interaction-based) event and censoring mechanisms while keeping covariates fixed.

We extract 36 covariates spanning laboratory test abnormalities (e.g., creatinine, glucose, hemoglobin), demographic features (e.g., age, sex, race, marital status), and admission descriptors (e.g., admission type, recurrent admissions, night admission). Table 29 summarizes covariate statistics; Table 30 reports demographic and categorical distributions; and Figure 20 shows correlations among covariates.

Table 29: Summary statistics of MIMIC semi-synthetic covariates. Reported values are mean $\pm$ standard deviation. Physiological covariates are coded as indicators for abnormal values where mean reflects prevalence of abnormality.

| Covariate | Mean $\pm$ Std | Covariate | Mean $\pm$ Std |
|---|---|---|---|
| Sodium | $0.12 \pm 0.32$ | Admission age | $61.39 \pm 17.97$ |
| Potassium | $0.08 \pm 0.28$ | Sex:Male | $0.51 \pm 0.50$ |
| Chloride | $0.19 \pm 0.39$ | Race:White | $0.70 \pm 0.46$ |
| Bicarbonate | $0.24 \pm 0.43$ | Race:Black | $0.14 \pm 0.35$ |
| Anion gap | $0.09 \pm 0.29$ | Race:Hispanic | $0.05 \pm 0.22$ |
| Creatinine | $0.28 \pm 0.45$ | Race:Other | $0.07 \pm 0.25$ |
| Urea nitrogen | $0.40 \pm 0.49$ | Insurance:Medicare | $0.42 \pm 0.49$ |
| Glucose | $0.65 \pm 0.48$ | Insurance:Other | $0.52 \pm 0.50$ |
| Calcium total | $0.29 \pm 0.45$ | Marital status:Married | $0.45 \pm 0.50$ |
| Magnesium | $0.09 \pm 0.28$ | Marital status:Single | $0.33 \pm 0.47$ |
| Phosphate | $0.28 \pm 0.45$ | Marital status:Widowed | $0.14 \pm 0.34$ |
| Hemoglobin | $0.73 \pm 0.44$ | Direct emergency:Yes | $0.11 \pm 0.31$ |
| Hematocrit | $0.69 \pm 0.46$ | Night admission:Yes | $0.54 \pm 0.50$ |
| MCV | $0.20 \pm 0.40$ | Previous admission this month: Yes | $0.08 \pm 0.27$ |
| MCH | $0.26 \pm 0.44$ | Admissions number:2 | $0.16 \pm 0.37$ |
| MCHC | $0.31 \pm 0.46$ | Admissions number:3+ | $0.22 \pm 0.42$ |
| Platelet count | $0.29 \pm 0.45$ | | |
| RDW | $0.29 \pm 0.45$ | | |
| White blood cells | $0.40 \pm 0.49$ | | |
| Red blood cells | $0.76 \pm 0.43$ | | |

We now introduce some shared notation in all of the MIMIC semi-synthetic datasets. Let $X_{1:5}$ denote the first five binary covariates corresponding to abnormal laboratory values (*Anion gap*, *Bicarbonate*, *Calcium total*, *Chloride*, *Creatinine*), and let $X_{36}$ denote the standardized *Admission age*. We define the abnormal-lab burden as

$$S = \sum_{j=1}^{5} X_j.$$

Across all MIMIC variants, we generate potential event times $T(0), T(1)$ and a censoring time $C$, and observe

$$T = WT(1) + (1 - W)T(0), \quad Y = \min(T, C), \quad \delta = \mathbb{1}\{T \leq C\}.$$

Table 30: Demographic and categorical distributions in MIMIC semi-synthetic datasets. Reported values are proportions.

| Demographics | |
|---|---|
| **Variable** | **Proportion** |
| **Sex** | |
| Male | 0.512 |
| Female | 0.488 |
| **Race** | |
| White | 0.699 |
| Black | 0.141 |
| Other | 0.066 |
| Hispanic | 0.053 |
| Asian | 0.041 |
| **Insurance** | |
| Other | 0.522 |
| Medicare | 0.421 |
| Medicaid | 0.057 |
| **Marital status** | |
| Married | 0.449 |
| Single | 0.334 |
| Widowed | 0.136 |
| Divorced | 0.081 |

| Admission-related | |
|---|---|
| **Variable** | **Proportion** |
| **Direct emergency** | |
| Yes | 0.110 |
| No | 0.890 |
| **Night admission** | |
| Yes | 0.539 |
| No | 0.461 |
| **Previous admission this month** | |
| Yes | 0.081 |
| No | 0.919 |
| **Admissions number** | |
| 1 | 0.615 |
| 2 | 0.164 |
| 3+ | 0.222 |

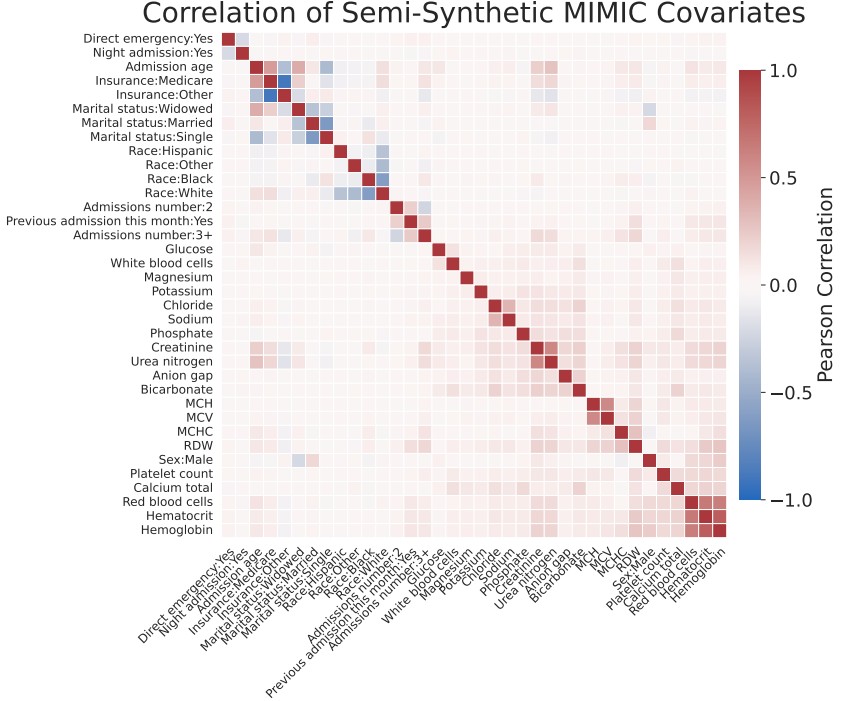

Figure 20: Correlation heatmap of the 36 semi-synthetic MIMIC covariates. Variables include demographic features, admission descriptors, insurance and marital status indicators, and laboratory measurements. Most correlations are weak to moderate, with stronger dependencies visible among related laboratory values (e.g., hematocrit, hemoglobin, and red blood cell count).

### G.3.1 INDEPENDENT-ASSIGNMENT, VARYING CENSORING (MIMIC-$i$–$v$)

This subset follows the event-time construction of Meir et al. (2025) and varies censoring severity to span moderate to extreme censoring regimes.

**Treatment assignment.** Treatment is assigned independently of covariates:

$$W \sim \text{Bernoulli}(0.5),$$

yielding balanced treatment groups with $W \perp X$.

**Potential outcomes.** Potential event times are Poisson with means depending on $S$ and $X_{36}$:

$$T(0) \sim \text{Poisson}(30 + 0.75S + 0.75X_{36}),$$
$$T(1) \sim \text{Poisson}(30 + 0.75X_{36} - 0.45).$$

We define the conditional average treatment effect (CATE) as $\tau(x) = \mathbb{E}[T(1) - T(0) \mid X = x]$ and record unit-level ground truth CATE as $T(1) - T(0)$.

**Censoring.** Censoring is independent and varies across dataset variants:

$$C \sim \text{Poisson}(\lambda_c),$$

where $\lambda_c \in \{21, 23, 24.7, 26.5, 29\}$ controls censoring severity for MIMIC-$i$–$v$, yielding censoring rates from approximately 53% to 88%.

### G.3.2 CONFOUNDED ASSIGNMENT, VARYING FUNCTIONAL FORM (MIMIC-$vi$–$ix$)

This subset reuses the same covariate backbone but introduces covariate-dependent treatment assignment (observed confounding) with a propensity score and varies whether the event-time and censoring mechanisms are linear or non-linear in covariates. The four variants correspond to the following combinations:

- MIMIC-$vi$: Propensity score=`linear`, event/censoring time=`linear`;
- MIMIC-$vii$: Propensity score=`linear`, event/censoring time=`non-linear`;
- MIMIC-$viii$: Propensity score=`non-linear`, event/censoring time=`linear`;
- MIMIC-$ix$: Propensity score=`non-linear`, event/censoring time=`non-linear`.

**Covariate-dependent treatment assignment (propensity score).** We consider two propensity-score families:

- **Linear (no interactions or quadratic terms).**

$$\eta(x) = \alpha_0 + \alpha_1 X_{36} + \alpha_2 S,$$
$$e(x) = \Pr(W = 1 \mid X = x) = \sigma(\eta(x)), \quad W \mid X \sim \text{Bernoulli}(e(X)),$$

  with $(a_0, a_1, a_2) = (-0.25, 0.8, 0.4)$;

- **Non-linear (quadratic + interaction).**

$$\eta(x) = \beta_0 + \beta_1 X_{36} + \beta_2 S + \beta_3 X_{36}^2 + \beta_4 X_{36} S,$$
$$e(x) = \Pr(W = 1 \mid X = x) = \sigma(\eta(x)), \quad W \mid X \sim \text{Bernoulli}(e(X)),$$

  with $(b_0, b_1, b_2, b_3, b_4) = (-0.2, 0.8, 0.4, -0.3, 0.5)$.

where $\sigma(\cdot)$ is the sigmoid function. We clip $e(x)$ into $[0.05, 0.95]$ to avoid near-deterministic treatment assignment and preserve overlap. Note that the coefficients are chosen (by checking the realized $\mathbb{E}[W]$) so that the treatment rate remains close to $0.5$ while inducing confounding through $X_{36}$ and $S$.

**Event-time and censoring mechanisms.** Potential outcomes and censoring are defined through Poisson means with an identity link and are clipped below at $1$ to ensure positivity.

- **Linear (no interactions or quadratic terms).**

$$\mu_0(x) = \psi_{00} + \psi_{01}S + \psi_{02}X_{36},$$
$$\mu_1(x) = \psi_{10} + \psi_{11}S + \psi_{12}X_{36},$$

with $(\psi_{00}, \psi_{01}, \psi_{02}) = (25.0,\ 0.75,\ 0.75)$ and $(\psi_{10}, \psi_{11}, \psi_{12}) = (25.0,\ 0.0,\ 0.75)$, and

$$T(0) \sim \text{Poisson}(\max\{1, \mu_0(X)\}), \quad T(1) \sim \text{Poisson}(\max\{1, \mu_1(X)\}).$$

Censoring is

$$\lambda_c(x) = \omega_0 + \omega_1 S + \omega_2 X_{36}, \quad C \sim \text{Poisson}(\max\{1, \lambda_c(X)\}).$$

with $(\omega_0, \omega_1, \omega_2) = (24.0,\ 0.2,\ 0.2)$.

- **Non-linear (quadratic + interaction).**

$$\mu_0(x) = \psi_{00} + \psi_{01}S + \psi_{02}X_{36} + \psi_{03}X_{36}^2 + \psi_{04}X_{36}S,$$
$$\mu_1(x) = \psi_{10} + \psi_{11}S + \psi_{12}X_{36} + \psi_{13}X_{36}^2 + \psi_{14}X_{36}S,$$

with $(\psi_{00}, \psi_{01}, \psi_{02}, \psi_{03}, \psi_{04}) = (25.0,\ 0.75,\ 0.75,\ 0.3,\ 0.5)$ and $(\psi_{10}, \psi_{11}, \psi_{12}, \psi_{13}, \psi_{14}) = (25.0,\ 0.0,\ 0.75,\ 0.2,\ 0.4)$, and

$$T(0) \sim \text{Poisson}(\max\{1, \mu_0(X)\}), \quad T(1) \sim \text{Poisson}(\max\{1, \mu_1(X)\}).$$

Censoring is

$$\lambda_c(x) = \omega_0 + \omega_1 S + \omega_2 X_{36} + \omega_3 X_{36}^2 + \omega_4 X_{36}S,$$
$$C \sim \text{Poisson}(\max\{1, \lambda_c(X)\}).$$

with $(\omega_0, \omega_1, \omega_2, \omega_3, \omega_4) = (24.0,\ 0.2,\ 0.2,\ 0.3,\ 0.4)$

These mechanisms allow survival outcomes and censoring to vary non-linearly with baseline severity (lab abnormalities) and age, thereby creating heterogeneous and more realistic treatment effects.

**Fixed-horizon survival probabilities.** Because event times are Poisson-distributed, the conditional survival probability for arm $w \in \{0, 1\}$ at discrete horizon $t$ is

$$S_w(t \mid X) = \Pr(T(w) > t \mid X) = 1 - \sum_{k=0}^{\lfloor t \rfloor} \frac{e^{-\mu_w(X)}\mu_w(X)^k}{k!}.$$

For each dataset, we compute unit-level ground-truth survival probabilities at horizons corresponding to the empirical 25th, 50th, and 75th percentiles of the realized event-time distribution, and use these when evaluating survival-probability CATE estimands.

Across all MIMIC variants, the sample size is $N = 25{,}170$ and the covariate distributions in Tables 29 and 30 are identical. The subset MIMIC-$i$–$v$ spans censoring rates from approximately 53% to 88% under covariate-independent treatment assignment ($\Pr(W{=}1) \approx 0.5$), isolating the effect of censoring severity. The subset MIMIC-$vi$–$ix$ keeps censoring moderate (around 53% in our instantiation) while introducing covariate-dependent treatment assignment and varying whether event-time and censoring mechanisms are linear or non-linear in covariates, providing complementary stress tests for both confounding and misspecified or non-linear hazard/censoring relationships.

### G.4 SEMI-SYNTHETIC RESULTS: FULL MIMIC SUITE AND ADDITIONAL ESTIMANDS

This section extends the semi-synthetic evaluation in Section 4.2. First, we complete coverage of the full MIMIC semi-synthetic suite by reporting RMST-based CATE RMSE on MIMIC-$vi$–$ix$ under the same primary estimand used in the main paper (RMST evaluated at the maximum observed time $T_{\max}$). Second, we report results for additional estimands that probe time-horizon sensitivity: horizon-specific survival-probability CATEs and RMST evaluated at a shorter horizon (the median event time $T_{\mathrm{med}}$).

#### G.4.1 PRIMARY ESTIMAND: RMST AT $T_{\max}$ (FULL MIMIC SUITE)

The main paper reports RMST-based CATE RMSE on ACTG and the censoring-severity subset MIMIC-$i$–$v$ (Table 3). Table 31 reports CATE RMSE on MIMIC-$vi$–$ix$ using the same estimand as in the main paper: RMST evaluated at the maximum observed time in each dataset. These variants reuse the same covariates as MIMIC-$i$–$v$ but introduce covariate-dependent treatment assignment and/or non-linear event-time and censoring mechanisms (Appendix G.3.2). Across MIMIC-$i$–$ix$, mean RMSE values typically lie in a narrow band, so practical differences are often reflected more strongly in stability (variance across repetitions) than in average error.

**Dataset-dependent performance patterns.** The ACTG dataset (Table 3) exhibits clearer separation among methods, whereas the MIMIC variants (Tables 3 and 31) show substantially tighter clustering. This reinforces that method rankings can depend on covariate structure and data-generating mechanisms: approaches that perform well in trial-like settings (ACTG) need not be the strongest in high-dimensional EHR-like settings (MIMIC), and vice versa.

**Censoring-gradient and stability under MIMIC-$i$–$v$.** The censoring sweep MIMIC-$i$–$v$ (53%–88% censoring) provides a granular view of censoring sensitivity. Several methods maintain stable mean RMSE as censoring increases, but standard deviation can change markedly for some learners under extreme censoring (Table 3). As a result, stability considerations (e.g., standard deviation across repeats) can be as important as mean RMSE when operating in highly censored regimes typical of EHR studies.

**Robustness across mechanism complexity (MIMIC-$vi$–$ix$).** Moving from MIMIC-$i$–$v$ to the mechanism-complexity subset MIMIC-$vi$–$ix$ does not qualitatively change the overall picture: multiple methods remain competitive and the top-performing group (Causal Survival Forests and S-Learner-Survival) is often tightly clustered (Table 31). This suggests that the primary conclusions drawn from MIMIC-$i$–$v$ are not driven solely by covariate-independent treatment assignment; rather, they persist under observed confounding and non-linear outcome/censoring mechanisms within the same covariate support.

#### G.4.2 ADDITIONAL ESTIMAND: HORIZON-SPECIFIC SURVIVAL-PROBABILITY CATES

In addition to RMST-based CATEs, we consider treatment effects defined on the survival function at a fixed horizon. Let $S_i(w; h) = \Pr(T_i(w) > h)$ denote the potential-outcome survival probability for unit $i$ under treatment $w \in \{0, 1\}$ at horizon $h$. In this case, the transformation $y(\cdot)$ in equation 1 is replaced by

$$y\big(T_i(w)\big) := S_i(w; h).$$

The corresponding CATE is:

$$\tau_h(x) := \mathbb{E}\big[\, S_i(1; h) - S_i(0; h) \mid X_i = x \,\big]. \tag{6}$$

We evaluate $\tau_h(x)$ at three dataset-specific horizons given by the empirical 25th, 50th, and 75th percentiles of the realized event-time distribution.

We evaluate $\tau_h(x)$ at three dataset-specific horizons given by the empirical 25th, 50th, and 75th percentiles of the realized event-time distribution. We report results for methods that explicitly model conditional survival functions (direct-survival CATE methods and survival meta-learners). Our outcome-imputation baselines impute event times (or RMST) rather than estimating a full conditional survival curve, and therefore are not included for this estimand. Tables 32–34 summarize RMSE across the MIMIC semi-synthetic datasets for the 25th, 50th, and 75th percentile horizons, respectively.

Table 31: CATE RMSE (mean $\pm$ std over 10 repeats) on MIMIC-$vi$–$ix$, using RMST evaluated at the maximum observed time $T_{\max}$ of each dataset as the estimand. Best two methods per dataset are **bolded**.

| Method Family | MIMIC-$vi$ | MIMIC-$vii$ | MIMIC-$viii$ | MIMIC-$ix$ |
|---|---|---|---|---|
| *Outcome Imputation Methods* | | | | |
| T-Learner | $7.184 \pm 0.052$ | $7.374 \pm 0.067$ | $7.220 \pm 0.046$ | $7.354 \pm 0.048$ |
| S-Learner | $7.176 \pm 0.048$ | $7.308 \pm 0.068$ | $7.197 \pm 0.049$ | $7.275 \pm 0.061$ |
| X-Learner | $7.182 \pm 0.053$ | $7.318 \pm 0.069$ | $7.203 \pm 0.045$ | $7.273 \pm 0.044$ |
| DR-Learner | $7.145 \pm 0.054$ | $7.295 \pm 0.061$ | $7.167 \pm 0.046$ | $7.263 \pm 0.045$ |
| Double-ML | $\mathbf{7.127 \pm 0.051}$ | $\mathbf{7.259 \pm 0.072}$ | $\mathbf{7.147 \pm 0.049}$ | $\mathbf{7.226 \pm 0.056}$ |
| Causal Forest | $7.142 \pm 0.052$ | $7.288 \pm 0.068$ | $7.162 \pm 0.047$ | $7.247 \pm 0.043$ |
| *Direct-Survival CATE Methods* | | | | |
| Causal Survival Forests | $\mathbf{7.123 \pm 0.048}$ | $\mathbf{7.281 \pm 0.064}$ | $\mathbf{7.149 \pm 0.045}$ | $\mathbf{7.227 \pm 0.054}$ |
| SurvITE | $7.169 \pm 0.042$ | $7.286 \pm 0.059$ | $7.184 \pm 0.043$ | $7.265 \pm 0.055$ |
| *Survival Meta-Learners* | | | | |
| T-Learner-Survival | $7.168 \pm 0.055$ | $7.332 \pm 0.071$ | $7.188 \pm 0.040$ | $7.358 \pm 0.047$ |
| S-Learner-Survival | $7.138 \pm 0.048$ | $7.285 \pm 0.064$ | $7.163 \pm 0.041$ | $7.239 \pm 0.055$ |
| Matching Survival | $7.163 \pm 0.050$ | $7.340 \pm 0.062$ | $7.199 \pm 0.043$ | $7.297 \pm 0.050$ |

Table 32: CATE RMSE (mean $\pm$ std over 10 repeats) on MIMIC semi-synthetic datasets, using survival probability at 25th quantile event time of each dataset as the estimand. Best method per dataset is **bolded**.

| Method Family | MIMIC-$i$ | MIMIC-$ii$ | MIMIC-$iii$ | MIMIC-$iv$ | MIMIC-$v$ |
|---|---|---|---|---|---|
| *Direct-Survival CATE Methods* | | | | | |
| Causal Survival Forests | $\mathbf{0.051 \pm 0.003}$ | $\mathbf{0.045 \pm 0.005}$ | $\mathbf{0.039 \pm 0.002}$ | $\mathbf{0.037 \pm 0.003}$ | $\mathbf{0.034 \pm 0.002}$ |
| SurvITE | $0.066 \pm 0.009$ | $0.060 \pm 0.004$ | $0.057 \pm 0.008$ | $0.055 \pm 0.005$ | $0.055 \pm 0.005$ |
| *Survival Meta-Learners* | | | | | |
| T-Learner-Survival | $0.083 \pm 0.011$ | $0.077 \pm 0.040$ | $0.049 \pm 0.008$ | $0.047 \pm 0.007$ | $0.045 \pm 0.005$ |
| S-Learner-Survival | $0.053 \pm 0.006$ | $0.050 \pm 0.006$ | $0.053 \pm 0.010$ | $0.051 \pm 0.004$ | $0.046 \pm 0.004$ |
| Matching Survival | $0.061 \pm 0.006$ | $0.058 \pm 0.005$ | $0.058 \pm 0.008$ | $0.055 \pm 0.004$ | $0.050 \pm 0.003$ |

| Method Family | MIMIC-$vi$ | MIMIC-$vii$ | MIMIC-$viii$ | MIMIC-$ix$ |
|---|---|---|---|---|
| *Direct-Survival CATE Methods* | | | | |
| Causal Survival Forests | $\mathbf{0.044 \pm 0.003}$ | $\mathbf{0.035 \pm 0.003}$ | $\mathbf{0.038 \pm 0.005}$ | $\mathbf{0.041 \pm 0.003}$ |
| SurvITE | $0.070 \pm 0.011$ | $0.066 \pm 0.011$ | $0.068 \pm 0.007$ | $0.076 \pm 0.009$ |
| *Survival Meta-Learners* | | | | |
| T-Learner-Survival | $0.057 \pm 0.007$ | $0.068 \pm 0.016$ | $0.074 \pm 0.014$ | $0.069 \pm 0.010$ |
| S-Learner-Survival | $0.051 \pm 0.005$ | $0.049 \pm 0.006$ | $0.054 \pm 0.008$ | $0.055 \pm 0.005$ |
| Matching Survival | $0.060 \pm 0.005$ | $0.062 \pm 0.004$ | $0.067 \pm 0.005$ | $0.071 \pm 0.002$ |

**Horizon effects.** Across MIMIC variants, performance gaps are generally more pronounced at earlier horizons (25th percentile) and become more compressed at later horizons (75th percentile). This is consistent with the intuition that early-horizon survival probabilities can be more sensitive to misspecification and estimation error, whereas later-horizon estimates may be less discriminative across methods in these settings.

**Method family patterns.** Across horizons, Causal Survival Forests tends to achieve the lowest RMSE among the survival CATE estimators, while survival meta-learners remain competitive but display method-specific sensitivity. In particular, S-Learner-Survival is typically among the most stable meta-learners across horizons and variants, whereas Matching Survival shows larger degradation for later-horizon survival probabilities in several variants (Tables 32–34). SurvITE is consistently less accurate than Causal Survival Forests under this estimand, with the gap more salient at earlier horizons.

Table 33: CATE RMSE (mean $\pm$ std over 10 repeats) on MIMIC semi-synthetic datasets, using survival probability at 50th quantile event time of each dataset as the estimand. Best method per dataset is **bolded**.

| Method Family | MIMIC-$i$ | MIMIC-$ii$ | MIMIC-$iii$ | MIMIC-$iv$ | MIMIC-$v$ |
|---|---|---|---|---|---|
| *Direct-Survival CATE Methods* | | | | | |
| Causal Survival Forests | $0.086 \pm 0.008$ | $0.077 \pm 0.008$ | $\mathbf{0.059 \pm 0.003}$ | $\mathbf{0.063 \pm 0.005}$ | $\mathbf{0.051 \pm 0.002}$ |
| SurvITE | $0.086 \pm 0.011$ | $0.081 \pm 0.009$ | $0.065 \pm 0.010$ | $0.070 \pm 0.006$ | $0.066 \pm 0.007$ |
| *Survival Meta-Learners* | | | | | |
| T-Learner-Survival | $0.144 \pm 0.006$ | $0.096 \pm 0.018$ | $0.092 \pm 0.018$ | $0.068 \pm 0.011$ | $0.065 \pm 0.009$ |
| S-Learner-Survival | $\mathbf{0.069 \pm 0.008}$ | $\mathbf{0.064 \pm 0.008}$ | $0.061 \pm 0.007$ | $0.064 \pm 0.005$ | $0.058 \pm 0.005$ |
| Matching Survival | $0.083 \pm 0.007$ | $0.078 \pm 0.007$ | $0.073 \pm 0.007$ | $0.072 \pm 0.004$ | $0.066 \pm 0.005$ |

| Method Family | MIMIC-$vi$ | MIMIC-$vii$ | MIMIC-$viii$ | MIMIC-$ix$ |
|---|---|---|---|---|
| *Direct-Survival CATE Methods* | | | | |
| Causal Survival Forests | $\mathbf{0.052 \pm 0.005}$ | $\mathbf{0.044 \pm 0.006}$ | $\mathbf{0.054 \pm 0.005}$ | $\mathbf{0.054 \pm 0.004}$ |
| SurvITE | $0.076 \pm 0.012$ | $0.071 \pm 0.010$ | $0.079 \pm 0.009$ | $0.081 \pm 0.009$ |
| *Survival Meta-Learners* | | | | |
| T-Learner-Survival | $0.084 \pm 0.013$ | $0.098 \pm 0.028$ | $0.093 \pm 0.020$ | $0.091 \pm 0.014$ |
| S-Learner-Survival | $0.063 \pm 0.007$ | $0.059 \pm 0.009$ | $0.065 \pm 0.010$ | $0.066 \pm 0.006$ |
| Matching Survival | $0.077 \pm 0.007$ | $0.081 \pm 0.006$ | $0.085 \pm 0.007$ | $0.091 \pm 0.002$ |

Table 34: CATE RMSE (mean $\pm$ std over 10 repeats) on MIMIC semi-synthetic datasets, using survival probability at 75th quantile event time of each dataset as the estimand. Best method per dataset is **bolded**.

| Method Family | MIMIC-$i$ | MIMIC-$ii$ | MIMIC-$iii$ | MIMIC-$iv$ | MIMIC-$v$ |
|---|---|---|---|---|---|
| *Direct-Survival CATE Methods* | | | | | |
| Causal Survival Forests | $0.104 \pm 0.020$ | $0.088 \pm 0.012$ | $0.066 \pm 0.007$ | $0.054 \pm 0.006$ | $0.052 \pm 0.003$ |
| SurvITE | $0.093 \pm 0.020$ | $0.065 \pm 0.010$ | $0.064 \pm 0.010$ | $0.055 \pm 0.005$ | $0.059 \pm 0.007$ |
| *Survival Meta-Learners* | | | | | |
| T-Learner-Survival | $0.137 \pm 0.041$ | $0.097 \pm 0.009$ | $0.085 \pm 0.014$ | $0.065 \pm 0.012$ | $0.059 \pm 0.007$ |
| S-Learner-Survival | $\mathbf{0.061 \pm 0.008}$ | $\mathbf{0.056 \pm 0.007}$ | $\mathbf{0.052 \pm 0.010}$ | $\mathbf{0.049 \pm 0.003}$ | $\mathbf{0.046 \pm 0.006}$ |
| Matching Survival | $0.077 \pm 0.008$ | $0.071 \pm 0.008$ | $0.065 \pm 0.009$ | $0.060 \pm 0.003$ | $0.055 \pm 0.006$ |

| Method Family | MIMIC-$vi$ | MIMIC-$vii$ | MIMIC-$viii$ | MIMIC-$ix$ |
|---|---|---|---|---|
| *Direct-Survival CATE Methods* | | | | |
| Causal Survival Forests | $0.053 \pm 0.004$ | $\mathbf{0.050 \pm 0.002}$ | $\mathbf{0.047 \pm 0.004}$ | $\mathbf{0.056 \pm 0.005}$ |
| SurvITE | $0.066 \pm 0.013$ | $0.058 \pm 0.008$ | $0.059 \pm 0.003$ | $0.066 \pm 0.009$ |
| *Survival Meta-Learners* | | | | |
| T-Learner-Survival | $0.086 \pm 0.013$ | $0.078 \pm 0.012$ | $0.067 \pm 0.011$ | $0.088 \pm 0.014$ |
| S-Learner-Survival | $\mathbf{0.052 \pm 0.007}$ | $0.054 \pm 0.009$ | $0.051 \pm 0.008$ | $0.059 \pm 0.006$ |
| Matching Survival | $0.065 \pm 0.008$ | $0.079 \pm 0.005$ | $0.072 \pm 0.006$ | $0.087 \pm 0.002$ |

Overall, the methods' relative performance appears stable across the 25th, 50th, and 75th percentile horizons, with no major reversals as $h$ changes.

### G.4.3 ADDITIONAL ESTIMAND: RMST HORIZON SENSITIVITY

The main paper reports RMST-based CATEs evaluated at a large horizon $h = T_{\max}$ (the maximum observed time in each dataset). To probe horizon sensitivity, we additionally evaluate RMST at a shorter horizon $h = T_{\mathrm{med}}$, the median of the realized event-time distribution in each dataset. Table 35 compares CATE RMSE under both horizons across the full MIMIC suite.

**Robustness to horizon choice.** Shortening the horizon typically reduces absolute RMSE (as expected for a less variable target), but the relative behavior of method families is largely preserved:

methods that are competitive at $T_{\max}$ tend to remain competitive at $T_{\mathrm{med}}$, and large ranking reversals are uncommon. Differences are more apparent in the stability of certain learners (e.g., matching-based survival approaches), which remain more sensitive to horizon choice (Table 35).

### G.4.4  DETAILED ANALYSIS AND PRACTICAL IMPLICATIONS

This subsection provides a consolidated interpretation of the semi-synthetic results across datasets and estimands (Tables 3, 31, 32–34, and 35), focusing on stability characteristics and implications for method selection.

**No universally best method across realistic data structures.**  Across ACTG and the MIMIC suite, no single method dominates uniformly. ACTG resembles a trial-like setting with moderate dimensionality and covariate-dependent assignment, where flexible doubly robust estimators can perform strongly (Table 3). In contrast, the high-dimensional, EHR-like MIMIC variants yield tighter performance bands, and survival-oriented approaches are frequently competitive (Tables 3 and 31). This highlights a limitation of relying on a single data structure: rankings can shift when covariate support, censoring regime, and assignment mechanisms change.

**Mean performance versus stability.**  In MIMIC, mean RMSE values are often close, so variability across repetitions provides an additional signal for method selection. Under extreme censoring (MIMIC-$i$–$ii$), some approaches exhibit noticeably higher variability, whereas others remain stable across censoring levels (Table 3). When performance differences are small, stability can be the more actionable differentiator.

**Estimand dependence and time-horizon effects.**  Comparing RMST-based CATEs to horizon-specific survival-probability CATEs illustrates that estimand choice can change how clearly methods separate. Survival-probability CATEs at earlier horizons tend to be more discriminative (Tables 32–34), while RMST-based targets can compress differences by integrating over time. The RMST horizon sensitivity analysis further indicates that shortening the horizon changes the scale of errors but rarely overturns broad conclusions (Table 35).

**Practical guidance.**  Taken together, the semi-synthetic results suggest the following heuristics. (i) In trial-like settings with moderate dimensionality and covariate-dependent assignment (e.g., ACTG), flexible causal estimators can provide strong accuracy (Table 3). (ii) In highly censored, high-dimensional EHR-like settings (e.g., MIMIC), survival-oriented estimators and stable meta-learner variants are often competitive, and stability under censoring becomes particularly important (Tables 3 and 31). (iii) When multiple methods fall within a tight RMSE band, secondary considerations—stability, interpretability, and computational cost—may dominate method choice.

Table 35: CATE RMSE (mean ± std over 10 repeats) on MIMIC semi-synthetic datasets, comparing RMST estimands at different horizons $h = T_{\max}$ (maximum observed time) and $h = T_{\mathrm{med}}$ (median event time). Best two methods per dataset are **bolded**.

| Method Family | MIMIC-$i$ $h=T_{\max}$ | MIMIC-$i$ $h=T_{\mathrm{med}}$ | MIMIC-$ii$ $h=T_{\max}$ | MIMIC-$ii$ $h=T_{\mathrm{med}}$ | MIMIC-$iii$ $h=T_{\max}$ | MIMIC-$iii$ $h=T_{\mathrm{med}}$ | MIMIC-$iv$ $h=T_{\max}$ | MIMIC-$iv$ $h=T_{\mathrm{med}}$ | MIMIC-$v$ $h=T_{\max}$ | MIMIC-$v$ $h=T_{\mathrm{med}}$ |
|---|---|---|---|---|---|---|---|---|---|---|
| *Outcome Imputation Methods* | | | | | | | | | | |
| T-Learner | 7.964 ± 0.046 | 4.367 ± 0.038 | **7.912 ± 0.046** | **4.363 ± 0.040** | 7.915 ± 0.043 | **4.361 ± 0.039** | 7.912 ± 0.043 | 4.361 ± 0.039 | 7.908 ± 0.043 | 4.360 ± 0.039 |
| S-Learner | 7.977 ± 0.044 | 4.387 ± 0.040 | 7.968 ± 0.047 | 4.390 ± 0.042 | 7.956 ± 0.050 | 4.387 ± 0.039 | 7.959 ± 0.046 | 4.387 ± 0.039 | 7.958 ± 0.048 | 4.387 ± 0.039 |
| X-Learner | 7.964 ± 0.046 | 4.367 ± 0.038 | **7.912 ± 0.046** | **4.363 ± 0.040** | 7.915 ± 0.043 | **4.361 ± 0.039** | 7.912 ± 0.043 | 4.361 ± 0.039 | 7.908 ± 0.043 | 4.360 ± 0.039 |
| DR-Learner | 7.964 ± 0.046 | 4.367 ± 0.038 | **7.912 ± 0.047** | **4.363 ± 0.040** | 7.911 ± 0.043 | **4.361 ± 0.039** | 7.911 ± 0.043 | 4.361 ± 0.039 | 7.909 ± 0.043 | 4.360 ± 0.039 |
| Double-ML | 7.954 ± 0.047 | **4.365 ± 0.040** | 7.936 ± 0.045 | **4.357 ± 0.041** | 7.919 ± 0.044 | **4.355 ± 0.040** | 7.917 ± 0.046 | **4.356 ± 0.040** | **7.891 ± 0.050** | **4.354 ± 0.040** |
| Causal Forest | 7.967 ± 0.045 | 4.371 ± 0.039 | 7.949 ± 0.044 | 4.365 ± 0.041 | 7.934 ± 0.043 | 4.365 ± 0.039 | 7.931 ± 0.047 | 4.364 ± 0.040 | 7.909 ± 0.044 | 4.363 ± 0.037 |
| *Direct-Survival CATE Methods* | | | | | | | | | | |
| Causal Survival Forests | 7.963 ± 0.057 | 4.376 ± 0.035 | 7.942 ± 0.039 | 4.363 ± 0.038 | 7.929 ± 0.037 | 4.364 ± 0.037 | 7.911 ± 0.051 | **4.357 ± 0.036** | **7.893 ± 0.042** | **4.357 ± 0.035** |
| SurvITE | **7.931 ± 0.050** | 4.381 ± 0.035 | **7.908 ± 0.065** | 4.377 ± 0.038 | **7.906 ± 0.071** | 4.375 ± 0.040 | 7.907 ± 0.058 | 4.374 ± 0.038 | 7.906 ± 0.066 | 4.376 ± 0.037 |
| *Survival Meta-Learners* | | | | | | | | | | |
| T-Learner-Survival | 8.007 ± 0.075 | 4.391 ± 0.040 | 7.980 ± 0.233 | 4.477 ± 0.340 | 7.911 ± 0.054 | 4.364 ± 0.034 | **7.902 ± 0.042** | 4.361 ± 0.038 | 7.902 ± 0.046 | 4.360 ± 0.036 |
| S-Learner-Survival | **7.921 ± 0.044** | **4.362 ± 0.039** | **7.912 ± 0.052** | **4.363 ± 0.036** | **7.900 ± 0.045** | **4.361 ± 0.038** | **7.901 ± 0.046** | 4.361 ± 0.037 | 7.897 ± 0.042 | 4.358 ± 0.038 |
| Matching Survival | 7.949 ± 0.043 | 4.603 ± 0.140 | 7.935 ± 0.053 | 4.600 ± 0.070 | 7.920 ± 0.047 | 4.652 ± 0.065 | 7.921 ± 0.046 | 4.714 ± 0.086 | 7.912 ± 0.042 | 4.735 ± 0.058 |

| Method Family | MIMIC-$vi$ $h=T_{\max}$ | MIMIC-$vi$ $h=T_{\mathrm{med}}$ | MIMIC-$vii$ $h=T_{\max}$ | MIMIC-$vii$ $h=T_{\mathrm{med}}$ | MIMIC-$viii$ $h=T_{\max}$ | MIMIC-$viii$ $h=T_{\mathrm{med}}$ | MIMIC-$ix$ $h=T_{\max}$ | MIMIC-$ix$ $h=T_{\mathrm{med}}$ |
|---|---|---|---|---|---|---|---|---|
| *Outcome Imputation Methods* | | | | | | | | |
| T-Learner | 7.184 ± 0.052 | 3.865 ± 0.036 | 7.374 ± 0.067 | 3.772 ± 0.027 | 7.220 ± 0.046 | 3.866 ± 0.031 | 7.354 ± 0.048 | 3.769 ± 0.032 |
| S-Learner | 7.176 ± 0.048 | 3.873 ± 0.037 | 7.308 ± 0.068 | 3.770 ± 0.026 | 7.197 ± 0.049 | 3.854 ± 0.032 | 7.275 ± 0.061 | 3.747 ± 0.034 |
| X-Learner | 7.182 ± 0.053 | 3.865 ± 0.036 | 7.318 ± 0.069 | 3.772 ± 0.027 | 7.203 ± 0.045 | 3.866 ± 0.031 | 7.273 ± 0.044 | 3.769 ± 0.032 |
| DR-Learner | 7.145 ± 0.054 | **3.854 ± 0.035** | 7.295 ± 0.061 | 3.751 ± 0.026 | 7.167 ± 0.046 | 3.846 ± 0.032 | 7.263 ± 0.045 | 3.737 ± 0.034 |
| Double-ML | **7.127 ± 0.051** | 3.855 ± 0.038 | **7.259 ± 0.072** | **3.743 ± 0.028** | **7.147 ± 0.049** | 3.849 ± 0.035 | **7.226 ± 0.056** | **3.732 ± 0.033** |
| Causal Forest | 7.142 ± 0.052 | 3.860 ± 0.035 | 7.288 ± 0.068 | 3.753 ± 0.026 | 7.162 ± 0.047 | 3.852 ± 0.033 | 7.247 ± 0.043 | 3.739 ± 0.029 |
| *Direct-Survival CATE Methods* | | | | | | | | |
| Causal Survival Forests | **7.123 ± 0.048** | **3.850 ± 0.032** | **7.281 ± 0.064** | **3.740 ± 0.025** | **7.149 ± 0.045** | **3.839 ± 0.031** | **7.227 ± 0.054** | **3.725 ± 0.032** |
| SurvITE | 7.169 ± 0.042 | 3.878 ± 0.034 | 7.286 ± 0.059 | 3.766 ± 0.025 | 7.184 ± 0.043 | 3.859 ± 0.029 | 7.265 ± 0.055 | 3.755 ± 0.031 |
| *Survival Meta-Learners* | | | | | | | | |
| T-Learner-Survival | 7.168 ± 0.055 | 3.860 ± 0.039 | 7.332 ± 0.071 | 3.761 ± 0.031 | 7.188 ± 0.040 | 3.861 ± 0.025 | 7.358 ± 0.047 | 3.762 ± 0.026 |
| S-Learner-Survival | 7.138 ± 0.048 | 3.856 ± 0.031 | 7.285 ± 0.064 | 3.751 ± 0.024 | 7.163 ± 0.041 | **3.845 ± 0.030** | 7.239 ± 0.055 | 3.735 ± 0.031 |
| Matching Survival | 7.163 ± 0.050 | 4.238 ± 0.038 | 7.340 ± 0.062 | 4.233 ± 0.039 | 7.199 ± 0.043 | 4.255 ± 0.028 | 7.297 ± 0.050 | 4.283 ± 0.063 |

## H    REAL-WORLD DATASETS: SETUP AND ADDITIONAL RESULTS

We evaluate our benchmark on two real-world datasets: the Twins dataset (with known ground truth) and the ACTG 175 HIV clinical trial dataset (without known ground truth). This section provides detailed descriptions of data preprocessing and additional experimental results.

### H.1    TWINS DATASET

The Twins dataset is derived from all births in the USA between 1989-1991 (Almond et al., 2005) focusing on twin births. Following Curth et al. (2021a), we artificially create a binary treatment where $W = 1$ ($W = 0$) denotes being born the heavier (lighter) twin. The outcome of interest is the time-to-mortality (in days) of each twin in their first year, administratively censored at $t = 365$ days. Since we have records for both twins, we treat their time-to-event outcomes as two potential outcomes $\tau(1)$ and $\tau(0)$ with respect to the treatment assignment of being born heavier. While the Twins dataset is a widely used benchmark (Louizos et al., 2017; Du et al., 2021; Curth et al., 2021a; Curth & Van der Schaar, 2021; Curth et al., 2021b), we note that treating twins as perfect counterfactuals at best is an approximation. The "ground-truth" relies on the assumption that the unobserved potential outcome of one twin is identical to the observed of their sibling, which in reality may not fully capture genetic or environmental heterogeneity.

We obtained 30 features (43 feature dimensions after one-hot encoding categorical features) for each twin relating to the parents, pregnancy, and birth characteristics including marital status, race, residence, number of previous births, pregnancy risk factors, quality of care during pregnancy, and number of gestation weeks prior to birth. We select only twins weighing less than 2kg and without missing features, resulting in more than 11,000 twin pairs.

To create an observational time-to-event dataset with known ground truth, we follow the semi-synthetic experimental design from Curth et al. (2021a). The treatment assignment is given by $W|x \sim \text{Bernoulli}(\sigma(\beta_1^\top x + e))$ where $\beta_1 \sim \text{Uniform}(-0.1, 0.1)^{43 \times 1}$ and $e \sim \mathcal{N}(0, 1^2)$. The time-to-censoring is given by $C \sim \text{Exp}(100 \cdot \sigma(\beta_2^\top x))$ where $\beta_2 \sim \mathcal{N}(0, 1^2)$. This results in a treatment rate of 68.1% and a censoring rate of 84.8%.

We split the data 50/25/25 for training/validation/testing samples and repeat all the experiments 10 times with different random splits. CATE RMSE are reported on the testing sets. In Section 4.3, we display the CATE RMSE with horizon $h = 30$ days. Here, we show CATE RMSE results for the Twins dataset with horizon $h = 180$ days in Figure 21, and we can see it indicates similar results as $h = 30$.

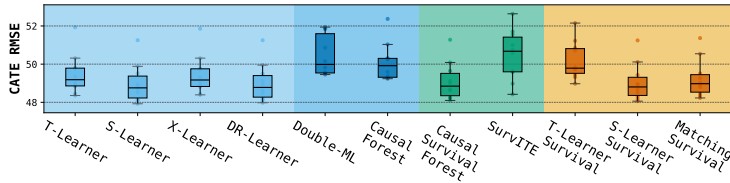

Figure 21: CATE RMSE for twin birth data using different estimator families with $h = 180$ days across 10 experimental runs.

## H.2 ACTG 175 HIV CLINICAL TRIAL DATASET

We use data from the AIDS Clinical Trials Group Protocol 175 (ACTG 175) (Hammer et al., 1996), a double-blind, randomized controlled trial that compared four treatment regimens in adults infected with HIV type I: monotherapy with zidovudine (ZDV), monotherapy with didanosine (ddI), combination therapy with ZDV and ddI, or combination therapy with ZDV and zalcitabine (Zal). The publicly available dataset[3] includes 2,139 HIV-infected patients randomized into four groups with assigned treatments: ZDV, ZDV+ddI, ZDV+Zal, and ddI. An event occurrence was defined as the first of either a decline in CD4 cell count, an event indicating AIDS progression, or death.

Following Meir et al. (2025), after fetching raw data from the UCI Machine Learning Repository, we change the resolution from days to months and add synthetic censoring based on a Bernoulli distribution with parameter $p = 0.6 + 0.25 \cdot Z30$, where $Z30$ is a feature that is available in the data and indicates whether a patient started taking ZDV prior to the assigned treatment, and it is not included in the covariates for CATE estimation. We conduct three pairwise comparisons with ZDV as the baseline treatment ($W = 0$): ZDV vs. ZDV+ddI (HIV1), ZDV vs. ZDV+Zal (HIV2), and ZDV vs. ddI (HIV3). The baseline censoring rate is less than 15% for different treatment groups. After applying the censoring injection procedure from Meir et al. (2025), increasing censoring rates to over 90%. For each treatment group, we establish baseline CATE estimates by running Causal Survival Forests 10 times and averaging the estimated conditional average treatment effects. Since there are many variants of outcome imputation and survival meta-learner families due to different imputation and base learner options, for display purposes in the HIV dataset results, we use a model selection criterion based on closeness (CATE RMSE) to estimation by Causal Survival Forests. We have looked at the results using other variants of same CATE estimator as well, and similar trends are observed.

In Section 4.3, we display the comparisons of CATE estimates between baseline and high-censoring conditions for group HIV1. Here we display the same sets of results for HIV2 and HIV3 groups in Figure 22, 23. Consistent patterns emerge across all three treatment comparisons: Causal Survival Forests produces estimates that cluster tightly around their baseline CATE estimations on data before additional censoring injection; outcome imputation methods show higher variation in baseline estimates but more concentrated predictions under high censoring, and survival meta-learners display substantial deviations from the 45-degree line, indicating sensitivity to censoring conditions. The consistency of these patterns across different treatment pairs reinforces the robustness of our findings regarding how different estimator families respond to increased censoring.

---

[3]https://archive.ics.uci.edu/dataset/890/aids+clinical+trials+group+study+175

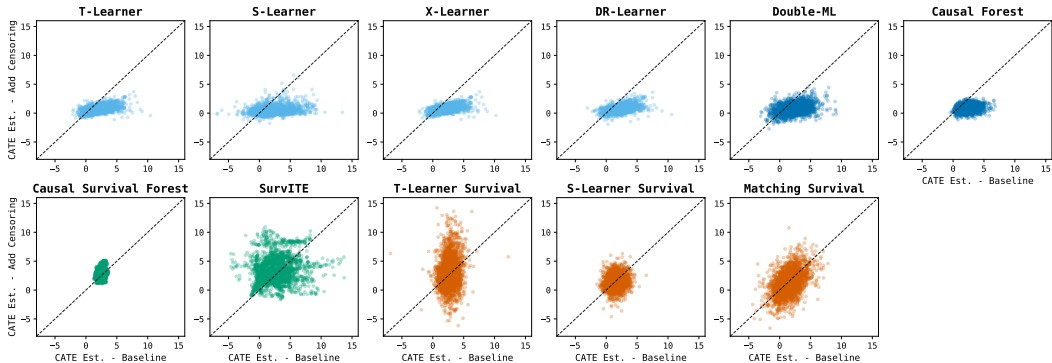

Figure 22: CATE Estimation comparison between baseline and high-censoring conditions under ZDV vs. ZDV+Zal treatments (HIV2). Each point represents an individual patient in test sets, with the dashed diagonal line indicating perfect consistency between baseline CATE estimation and that with the additional censoring injected.

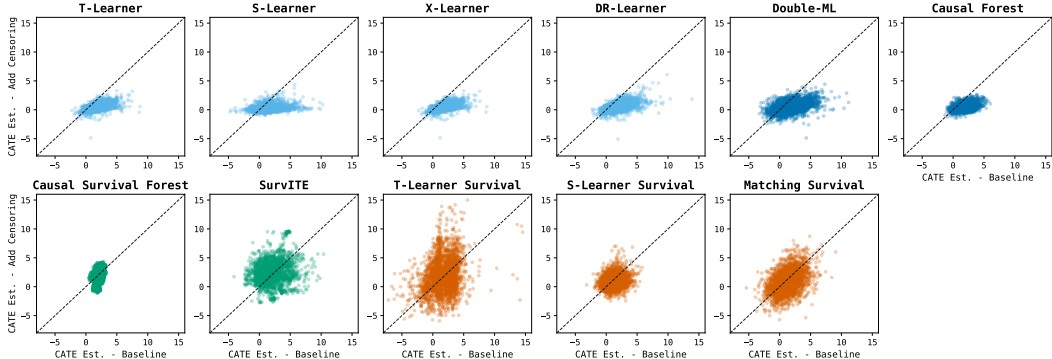

Figure 23: CATE Estimation comparison between baseline and high-censoring conditions under ZDV vs. ddI treatments (HIV3). Each point represents an individual patient in test sets, with the dashed diagonal line indicating perfect consistency between baseline CATE estimation and that with the additional censoring injected.

# I  ADDITIONAL INFORMATIVE CENSORING VIA UNOBSERVED CONFOUNDING

In the main paper, we model informative censoring by making censoring times stochastically dependent on event times, reflecting realistic scenarios where patients with shorter expected survival may drop out earlier. Here we complement this setting with an alternative mechanism where the ignorable censoring assumption is violated due to unobserved confounding. This extension demonstrates the flexibility of our modular data generation framework.

**Data generation process.**  We follow the same covariate generation procedure as in our synthetic datasets: observed covariates $X \sim \text{Uniform}(0, 1)^5$ and an unobserved covariate $U \sim \text{Uniform}(0, 1)$. Treatment assignment follows the `OBS-UConf` configuration, where $U$ enters into both treatment assignment and outcome generation but remains unobserved during estimation.

We focus on survival Scenario `C` (Poisson hazards with medium censoring). Event times and censoring times are generated as follows, where $w \in \{0, 1\}$ is the treatment indicator:

$$\lambda(w) = X_2^2 + X_3 + 6 + 2\left(\sqrt{0.3 \cdot X_1 + 0.7 \cdot U} - 0.3\right) \cdot w + \epsilon, \tag{7}$$

$$T(w) \sim \text{Poisson}(\lambda(w)), \tag{8}$$

$$C = \begin{cases} \infty & \text{if } U \le 0.6, \\ 1 + \mathbb{1}(X_4 < 0.5) & \text{otherwise,} \end{cases} \tag{9}$$

where $\epsilon \sim \mathcal{N}(0, 0.1)$ adds stochastic variation. The censoring distribution thus depends directly on the unobserved variable $U$, creating dependence between censoring and survival that cannot be explained away by the observed $X$ alone.

**Summary statistics**  Similar to the other synthetic datasets, we include up to 50,000 samples with treatment assigned according to an observational study mechanism. The treatment rate is 53.9%, the censoring rate is 39.7% (driven by $U$), and the population-level ATE is 0.7737 (computed from the 50,000 samples by averaging the CATEs). This setup mirrors real-world contexts such as clinical trials with dropout patterns influenced by latent health status.

**Experimental results**  We evaluated representative estimators from all three method families. Figure 24 reports CATE RMSE and ATE bias (mean ± standard error) across 10 random splits.

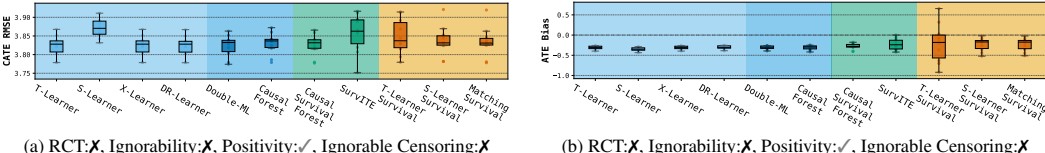

(a) RCT:✗, Ignorability:✗, Positivity:✓, Ignorable Censoring:✗     (b) RCT:✗, Ignorability:✗, Positivity:✓, Ignorable Censoring:✗

Figure 24: CATE RMSE (left) and ATE bias (right) under informative censoring induced by unobserved confounding.

The results indicate that Causal Survival Forests and survival meta-learners with matching tend to perform best under this setting, consistent with findings from the main synthetic datasets.

**Extensibility to other settings.**  Here we illustrate one case: `OBS-UConf` combined with Scenario `C`. However, the same mechanism can be straightforwardly extended to other causal configurations (e.g., randomized trials with imbalance) and survival scenarios (e.g., AFT or Cox models). We leave systematic exploration of these additional combinations for future work, but their ease of inclusion highlights the flexibility of SURVHTE-BENCH to accommodate alternative censoring mechanisms.

