# OpenReview forum: "SurvHTE-Bench: A Benchmark for Heterogeneous Treatment Effect Estimation in Survival Analysis"
_ICLR.cc/2026/Conference — ICLR 2026 Poster_

### Official Review · Reviewer_m4FZ · 2025-10-27

**Soundness:** 2
**Presentation:** 2
**Contribution:** 3
**Rating:** 4
**Confidence:** 3

**Summary:**

This paper is a benchmark comparing many methods for time-dependent conditional average treatment estimation, which is key in survival analysis.

**Strengths:**

This paper tackles the important evaluation of the plural methods existing for HTE with survival outcomes. The benchmark evaluates methods under a grid of relevant synthetic scenarios in a structured way, and on real-world datasets.

**Weaknesses:**

While the paper presents a valuable and ambitious benchmarking effort, I found it somewhat difficult to read due to repetition, particularly concerning the repeated description of the three families of survival HTE estimators. Streamlining these sections could make the paper more concise and readable.

A major omission in the related work is the lack of reference to existing benchmarks for survival ATE, such as [1]. This recent contribution on causal inference for survival outcomes and meta-learning estimators is highly relevant to the present study. Including and situating the current benchmark within that context would strengthen the paper’s positioning.

In Table 1, the RCT/Observational distinction is presented as an assumption being held or violated, whereas it is in fact a study design choice rather than an assumption. I would recommend removing that column or clearly separating design factors from causal assumptions.

Regarding the simulation scenarios, the rationale for comparing RCTs with 50% versus 5% treated participants is not entirely clear. Since randomization guarantees ignorability regardless of treatment imbalance, it is not obvious what this variation adds conceptually beyond a variance stress test. Similarly, it is surprising that there is no RCT scenario with non-ignorable censoring, given that informative censoring can occur even in randomized studies.

There is also a typo at line 227, where “2,500 points” is reported twice.

Conceptually, the description of informative censoring is somewhat unclear. The paper defines A4 as conditional independence between censoring and event time given covariates and treatment, but it remains ambiguous how this assumption is violated across scenarios (e.g., whether censoring depends on T directly, on unobserved confounders, or both). In the causal configurations labeled A–D, several aspects appear to change simultaneously, making it difficult to isolate the effect of each violation. Similarly, in Figure 2, it is confusing that the paper refers to “Scenario C” while ignorable and non-ignorable censoring seem to coexist.

Figure 1 is not very easy to read; in addition to being small, it may not fully respect the ICLR template in terms of spacing between the plot and caption. Moreover, I am not convinced that evaluating estimators’ average performance across all scenarios is particularly informative. It would be much more insightful to emphasize “which method works best under which conditions.” In particular, comparing IPCW-based estimators outside of the informative censoring (InfC) settings does not seem meaningful, since those estimators are specifically designed for that violation.

From a results standpoint, Figure 2 shows relatively small differences in RMSE across estimators, suggesting that performance gaps may not be statistically significant—perhaps because ten repetitions are insufficient for stable estimates.

Overall, I find that a great deal of work was put into this paper, but more attention is needed to the presentation and analysis of results, so that the article can better answer questions such as “which method works best for a given scenario?” Knowing which method performs best on average is less informative, since some assumptions are testable and practitioners often operate under known or partially known conditions.

Finally, while the benchmark is well implemented, it remains somewhat descriptive. The paper would benefit from a stronger analytical component—focusing less on global rankings and more on interpretable insights about which methods are most reliable under specific types of assumption violations. This would make the contribution more actionable for practitioners.

Also, I do not really see a large difference in RMSEs in Figure 2 across estimators—perhaps ten repeats is not sufficient for clear differentiation.

Finally, I am willing to increase my grade if the authors answer my concerns, especially concerning the "when does each method works best" issue.

[1] Voinot, Charlotte, et al. “Causal survival analysis, Estimation of the Average Treatment Effect (ATE): Practical Recommendations.” arXiv preprint arXiv:2501.05836 (2025).

**Questions:**

Why is there no RCT scenario with non-ignorable (informative) censoring? Since informative censoring can occur even in randomized studies, such a configuration would seem important for completeness.

Would it be possible to present the results more clearly by scenario, explicitly showing which methods perform best under each assumption violation (e.g., lack of ignorability, positivity, or informative censoring)? This would make the benchmark’s insights much easier to interpret.

What is the motivation for including IPCW-based estimators in settings where censoring is already ignorable? Their inclusion in those cases seems less informative, as IPCW adjustments are designed specifically for informative censoring scenarios.

How do you interpret the strong performance of DeepSurv-based meta-learners compared to Double-ML in high-censoring settings?

Could you explain why Double-ML performs relatively poorly on the Twins dataset, despite ranking highly in synthetic experiments? Does this reflect differences in sample size, feature distribution, or unmeasured confounding?

Could you include more references to existing papers that explicitly state which methods they use for survival CATE estimation? This would help situate your benchmark within the current methodological landscape.

Would it be possible to provide clearer results by scenario, highlighting which methods work best under which assumption violations?

In Figure 1, DeepSurv-based methods achieve overall strongest RMSE performance across many scenarios, likely due to their overparameterization and resulting robustness to model misspecification. How do these methods perform in simpler, well-specified settings such as the Cox proportional hazards scenario? Do they still outperform other approaches when the underlying model assumptions hold?

---

> ### Author Response · Authors · 2025-11-22
> **Response to Reviewer m4FZ [1/3]**
>
> We thank the reviewer for the careful reading and constructive feedback. Below provide our response for each concern and question.
>
> ---
>
>
> ### [W1: Repetition / readability]
> Thank you for pointing this out. Our intent was to first provide a unified taxonomy and literature review of survival HTE estimators (Section 2), and then give a benchmark-specific instantiation of that taxonomy, i.e., the exact estimators we implemented and evaluated (Section 3). We believe both sections are necessary for a benchmark paper: Section 2 establishes the conceptual landscape, while Section 3 specifies the concrete methods being compared.
>
> At the same time, we agree that some repetition can be reduced. To address this, we have shortened the overview in Section 1. Specifically, we revised the second paragraph to avoid re-enumerating method families there, and instead provide a concise pointer to the broader literature, now reading:
>
> > Recent years have seen a growing set of causal survival methods (Chapfuwa et al., 2021; Curth
> et al., 2021a; Cui et al., 2023; Bo et al., 2024; Noroozizadeh et al., 2025; Xu et al., 2024; Meir
> et al., 2025).
>
> This replaces the earlier version that repeated the family-level breakdown already detailed in Sections 2–3. We hope this streamlining improves readability while keeping the taxonomy and benchmark setup clear.
>
> ### [W2: Missing related benchmark work]
> Thank you for pointing us to Voinot et al. (2025). That benchmark focuses on survival ATE, whereas our benchmark targets survival CATE/HTE, but the underlying causal-survival themes (estimands, censoring, identification assumptions, etc.) are highly aligned. We will situate our contribution relative to theirs, emphasizing the overlap in goals and clarifying the additional challenges introduced by heterogeneity.
>
> We have now added the following text in the revised draft to cite related benchmarks in Section 1:
> > While there is a growing benchmarking literature for treatment-effect heterogeneity in fully observed outcomes (e.g., Crabb´e et al. (2022); Shimoni et al. (2018); Kapkic¸ et al. (2024)) and recent benchmarks for survival ATE estimation (e.g., Voinot et al. (2025)), to our knowledge, there is not yet any benchmark for survival HTE estimation under right-censoring. This missing piece motivates our focus on heterogeneous effects in censored time-to-event data.
>
> ### [W3: Table 1 mixes design vs assumptions]
> We appreciate this clarification. You are right that the RCT vs. observational distinction reflects a study-design choice rather than an “assumption violation”. We will keep these factors in Table 1, but adjust the notation and wording: in caption of Table 1, instead of
> > ✓ = assumption holds, ✗ = violated
> we have changed it to be
> > ✓=held, ✗= not held
> to indicate whether a given setting/design factor or identifying condition is present vs. absent in that configuration. We have also made sure there’s no text in our paper indicating observational study is a type of “violation”.
>
> ### [W4: Rationale for treatment imbalance in RCT + missing InfC RCT scenario]
> We varied treatment rates in RCT settings primarily as a stress test for estimators whose variance depends on treatment split. In particular, meta-learners such as T-learners (and survival T-learners) fit models separately on treated vs. control groups; severe imbalance can materially affect stability even when ignorability holds.
> We also agree that an RCT setting with informative (non-ignorable) censoring is realistic and important. We apologize for not including it in the initial suite. In our initial draft discussion part, we have acknowledged that our synthetic dataset do not cover all possible complexities. We are now explicitly adding the missing configuration RCT setting with informative censoring there.
>
> > First, the synthetic datasets include numerous scenarios representing common real-world violations, however, they do not encompass all possible complexities, for example, RCT setting with informative censoring or varying degrees of severity in assumption violations.
>
> Given the rebuttal time constraints, while we may not be able to fully re-benchmark all methods on this setting before final submission, we do plan to add this setting to the synthetic generator with code and datasets being released publicly for future research reference as part of the benchmark for completeness and future extensions.
>
> ### [W5: Typos]
> Thank you for pointing this out. We apologize for the confusion: the repeated “2,500 points” refers to the validation and test set sizes (with 5,000 for training). The validation set is used for model selection.

---

> > ### Author Response · Authors · 2025-11-22
> > **Response to Reviewer m4FZ [2/3]**
> >
> > ### [W6: Informative censoring description unclear / multiple changes per config]
> > We appreciate this comment and will clarify the presentation. In our main benchmark, informative censoring is induced by making censoring times stochastically dependent on event times, i.e., violating ignorability through direct dependence between C and T. We additionally include an alternative InfC mechanism driven by unobserved factors in Appendix I, to show extensibility. This content appears in the “The 8 causal configurations” paragraph in Section 3 in our initial submission.
> >
> > We want to clarify: Survival Scenarios A–E specify the event/censoring time distributions (and thus different censoring rates), while Causal Configurations (A–D / 1–8 in the paper) specify the causal assumptions and study design (RCT vs OBS, ignorability/positivity, ignorable vs informative censoring). A single survival scenario can be paired with multiple causal configurations.
> >
> > ### [W7: interpretable insights “when does each method work best?”]
> > Thanks for this strong suggestion; we fully agree that the most valuable outcome of a benchmark is clarity on which methods work best under which conditions and is exactly what our benchmark aims for. Figure 1 is intended only as a high-level overview under space constraints.
> > Importantly, we do not rely only on global averages. In Appendix F we already report rankings by survival scenario (Figure 6) and by causal configuration (Figure 7), providing condition-specific comparisons.  Section 4 further distills these into actionable takeaways
> >
> > Building on that foundation, we have now added **win-rate analyses** in Appendix F.1–F.3 (Top-1/Top-3/Top-5 percentage) to complement the Borda rankings. Specifically, we have added new sets of tables (Table 15 in Appendix F.1 across all experimental configurations, Table 16 in Appendix F.2 for win-rate performance at each survival scenario variation, and Table 17 in Appendix F.3 for win-rate performance at each causal assumption violation) reporting the percentage of times each method family appears in the Top-1, Top-3, and Top-5 positions according to both CATE RMSE and ATE Bias. This provides a clearer view of how often a method is “best” versus “average”. The accompanying text added to these sections (Appendices in F.1, F.2, and F.3) also help give an overall view of the strength-areas of each method family for different assumption violations and setup.
> >
> > Here we are copying the newly added Table 15, but we encourage the reviewer to also look at Table 16 and 17 and *also the text added* in Appendix F.2 and F.3.
> >
> > | Method Family           | CATE RMSE Top-1 | CATE RMSE Top-3 | CATE RMSE Top-5 | ATE Bias Top-1 | ATE Bias Top-3 | ATE Bias Top-5 |
> > |----|----:|------:|----:|---:|---:|----:|
> > | *Outcome Imputation Methods* |   |   |   |  |  |  |
> > | T-Learner               |  0.0 | 0.0 | 0.0 | 0.0 | 17.5 | 25.0 |
> > | S-Learner               |  0.0 | 2.5 | 12.5 | 5.0 | 7.5 | 27.5 |
> > | X-Learner               |  0.0 | 0.0 |  0.0 | 2.5 | 7.5 | 17.5 |
> > | DR-Learner              | 0.0 | 0.0 | 0.0 | 0.0 | 10.0 | 37.5 |
> > | Double-ML               | 27.5 | 62.5 | 85.0 | 2.5 | 15.0 | 37.5 |
> > | Causal Forest           | 2.5 | 40.0 | 52.5 | 2.5 | 27.5 | 40.0 |
> > | *Direct-Survival Methods*    |  | |  |  | | |
> > | Causal Survival Forest  | 35.0 |  67.5 |  82.5 | 52.5 |  75.0 |  82.5 |
> > | SurvITE                 |   25.0 | 37.5 | 45.0 | 15.0 | 37.5 | 55.0 |
> > | *Survival Meta-Learners*     | | | | | | |
> > | T-Learner-Survival      | 0.0 | 10.0 | 40.0 | 12.5 | 30.0 | 40.0 |
> > | S-Learner-Survival      | 10.0 | 45.0 | 97.5 | 5.0 | 30.0 | 65.0 |
> > | Matching-Survival       |  0.0 | 35.0 | 85.0 | 2.5 | 42.5 | 72.5 |
> >
> >
> > We have revised the main-text discussion to foreground these win-rate findings and point readers directly to Appendix F.1–F.3, so that practitioners can readily map observed dataset conditions (especially censoring severity and survival dynamics) to the families that are most reliable in those regimes.
> >
> > ### [W8: Small RMSE gaps / only 10 repeats]
> > Thank you for raising this. We used boxplots to summarize variability across runs because the benchmark produces many results across datasets and configurations; listing all means/stds in tables would be difficult to parse. While the shared y-axis and relatively wide inter-quartile ranges can make gaps look small at first glance, there are still meaningful variations across methods. For instance, in Figure 2(a), the median RMSE of the T-Learner is clearly lower than that of the S-Learner, and similar median shifts appear in other settings. We agree that enforcing a common y-axis to fit all panels may visually compress differences. In the revised paper draft, we have polished the figure (also with results of newly added baseline SurvITE included) to make these variations more salient—for example, we have slightly increased plot height and used zoomed-in y-ranges where appropriate (see Figure 2 and Figure 8-17).

---

> > > ### Author Response · Authors · 2025-11-22
> > > **Response to Reviewer m4FZ [3/3]**
> > >
> > > ### [Q1: Why no RCT + informative censoring scenario?]
> > > Addressed in W4. We agree that this is an important configuration and will add it to the data generator.
> > >
> > > ### [Q2/Q7: Can results be presented by scenario / “which method works best when”?]
> > > Addressed in W7. Per-scenario and per-configuration rankings are already in Appendix F; discussions are in Section 4/Appendix F; new “win-rate” insights added.
> > >
> > > ### [Q3: Motivation for IPCW methods under ignorable censoring?]
> > > We indeed agree that IPCW is most conceptually motivated under informative censoring.
> > > In our taxonomy, IPCW appears as one imputation strategy within the outcome-imputation family. Even when censoring is ignorable, IPCW can be viewed as a principled way to reconstruct full-data pseudo-outcomes from right-censored observations, enabling standard CATE learners (e.g., Double-ML) to operate on an imputed dataset.
> > >
> > > ### [Q4: Why do DeepSurv meta-learners outperform Double-ML under high censoring?]
> > > We suspect that the following is happening: Double-ML in our benchmark is imputation-based. It first imputes censored event times (i.e., any data point with a censored time has its time replaced by an imputed survival time that could potentially be inaccurate) and then applies regression-style causal learning, so its accuracy naturally degrades as censoring increases. In contrast, DeepSurv-based survival meta-learners directly account for censoring in the likelihood and can still learn under heavy right-censoring, so that they could be more stable in that regime.
> > >
> > > ### [Q5: Why does Double-ML underperform on Twins vs synthetic?]
> > > Twins is a high-censoring semi-synthetic dataset (censoring rate ≈ 85%). Consistent with Q4, imputation-based Double-ML is more sensitive to such high censoring. This pattern mirrors our synthetic results: Double-ML is competitive in lower-censoring scenarios but degrades in the highest-censoring ones, which is one of the main messages we aim to convey.
> > >
> > > ### [Q6: Add more references for survival CATE methods?]
> > > We are not sure if we understand the reviewer’s comment accurately but we try to provide more context in the survival CATE methods:
> > >
> > > The survival-CATE literature is rapidly evolving, and we have expanded the related-work section in the Introduction to better reflect the current landscape while clarifying how existing methods map to our three-family taxonomy. Concretely, as already summarized in our paper, prior survival HTE estimators fall into: (i) **outcome-imputation approaches**, which first handle censoring via strategies such as pseudo-observations, margin-based imputation, or IPCW-type reweighting, and then apply standard CATE learners including causal forests, Double-ML, and meta-learners such as S-, T-, X-, and DR-learners; (ii) **direct-survival CATE models**, which estimate CATEs directly from time-to-event data, including targeted learning / TMLE, tree-based direct estimators, SurvITE, Bayesian survival-causal models, and causal survival forests; and (iii) **survival meta-learners**, which adapt S-/T-/matching-learner frameworks to survival outcomes via survival base models such as Random Survival Forests, DeepSurv, or DeepHit. We will make these connections more explicit in the revision and add additional recent references in each family to situate SurvHTE-Bench within ongoing methodological developments.
> > >
> > > ### [Q8: Do DeepSurv methods still win in well-specified Cox settings?]
> > > DeepSurv is a neural network extension of the Cox PH model so that under a standard Cox setting, we would expect DeepSurv to work well (namely when DeepSurv's base neural network is just a single linear layer, it becomes the standard Cox PH model).
> > >
> > > ---
> > >
> > > We sincerely appreciate the reviewer’s detailed feedback. We believe the revisions above will make the benchmark clearer, more actionable, and better positioned in the literature.

---

### Official Review · Reviewer_FuYz · 2025-10-27

**Soundness:** 3
**Presentation:** 3
**Contribution:** 3
**Rating:** 4
**Confidence:** 4

**Summary:**

This paper introduces a comprehensive benchmark designed to evaluate methods estimating heterogeneous treatment effects (HTEs) from (right-censored) survival data. Estimating HTEs in survival analysis is vital for applications such as precision medicine but remains challenging due to censoring, unobserved counterfactuals, and complex modeling assumptions. This paper addresses these challenges by providing a unified categorization of survival HTE methods into three families—outcome imputation, direct-survival CATE models, and survival meta-learners—along with modular implementations of their variants. It also includes synthetic datasets with known ground-truth HTEs that systematically vary causal assumptions (e.g., randomization, confounding, positivity, informative censoring) and survival dynamics (e.g., Cox, AFT, Poisson models with varying censoring rates). Additionally, it offers multiple semi-synthetic datasets and two real-world datasets.

**Strengths:**

-	Addresses a Critical Need: The paper tackles a significant gap in the causal inference and survival analysis literature -- the lack of standardized evaluation practices for HTE estimation with survival data.
-	The benchmark incorporates multiple data types (synthetic, semi-synthetic, real). The synthetic data explores a wide range of crucial factors: different causal assumption violations (i.e., confounding, positivity, informative censoring) and commonly used survival/censoring distributions (Cox, AFT, Poisson).

**Weaknesses:**

Major comments:

-	Limited synthetic data generation processes: The synthetic datasets are restricted to three survival time generation mechanisms—Cox, AFT, and Poisson—which may not capture more complex, non-parametric scenarios. For instance, SurvITE defines discrete hazards and samples time-to-event or censoring outcomes based on those hazards. Also, the validity and representativeness of such synthetic data remain unclear. In particular, how treatment effects, selection biases, and covariate shifts influence the generated time-to-event outcomes is not well articulated, as the datasets are primarily described in terms of their underlying hazard functions.
- Lack of actionable insights from the benchmark: The benchmark results do not provide sufficient insight into how model design decisions should be made based on the findings. As a result, the analysis feels largely descriptive rather than being guided by principled intuitions or hypotheses about model behavior. The paper could better highlight how specific model characteristics or data properties influence performance, offering clearer guidance for model selection and application.
-	The paper introduces a category of "Direct-Survival CATE Models", but, deep survival models that directly estimate the conditional average treatment effect (CATE) (such as SurvITE) on survival outcomes are not included.
-	Although the paper claims to provide a unified framework for handling various survival models, it does not address models that directly predict time-to-event outcomes (e.g., Chafuwa et al., 2021). It is unclear how such models would be integrated into the proposed benchmark.
-	Estimates of hazard or survival functions can vary substantially depending on the chosen time horizon, yet the benchmark does not clearly specify or analyze how this dependency affects the evaluation results.

Minor comments:

-	Visualizations (e.g., conditional hazard and survival function plots) for synthetic datasets under different causal configurations would help clarify and validate the underlying data generation processes.

**Questions:**

-	If ground-truth hazard functions are available, it would be more meaningful to evaluate HTE performance with respect to the hazard functions themselves. Since many of the evaluated methods are capable of directly estimating conditional hazards from data, it is unclear why these performance results are not reported in the paper.
-	Regarding Weakness 2, it remains unclear how the benchmark results can be practically leveraged when applying the methods to new datasets or models.

---

> ### Author Response · Authors · 2025-11-22
> **Response to Reviewer FuYz [1/4]**
>
> We thank the reviewer for the thoughtful and constructive feedback. Below we respond point-by-point and describe concrete changes we have made (or will make) in the revision.
>
> ---
>
> ### [W1: Limited synthetic generation processes / representativeness]
> Our synthetic suite currently instantiates three widely used survival mechanisms, Cox PH, AFT, and Poisson hazards, because they are the most common testbeds in causal survival work and they let us generate controlled, ground-truth CATEs while systematically varying assumption violations. Regarding how treatment effects and biases influence outcomes, we believe these are already encoded via our causal configurations: several settings make treatment assignment depend on covariates (introducing confounding), and we additionally vary overlap/positivity and informative censoring, which together determine the resulting counterfactual hazards and survival curves. We acknowledge that the current generators do not span all possible survival-time settings, for example, more flexible non-parametric or discrete-hazard processes, and we are saying this limitation explicitly in the discussion, positioning such cases as natural future extensions supported by the benchmark’s modular design.

---

> > ### Author Response · Authors · 2025-11-22
> > **Response to Reviewer FuYz [2/4]**
> >
> > ### [W2: Lack of actionable insights]
> > We have provided key takeaways and actionable insights in Section 4 and a few high-level summary includes: (1) **censoring rate is one of the primary drivers of method choice**: in low-censoring settings, outcome-imputation approaches such as X-Learner and Double-ML lead, while as censoring increases, survival meta-learners and Causal Survival Forest progressively rise to the top.  (2) **assumption violations shift the relative strengths of families in predictable ways**: under ignorability violations or informative censoring, survival meta-learners remain comparatively robust, and Double-ML is the only imputation method that stays competitive; under positivity violations alone, sophisticated imputation methods (X-Learner/Double-ML) outperform survival meta-learners, but when multiple violations co-occur, survival meta-learners regain the advantage. (3) **within-family robustness differs under treatment imbalance**: when treatment is highly imbalanced, methods relying on separate treated/control fits (e.g., T-Learner-Survival) degrade sharply, which is an expected behavior, given their reliance on splitting the sample by treatment, which amplifies variance and instability under sparse treated groups.
> >
> > **Newly added “Win-Rate” Metrics:** To further capture effect sizes and dominance more transparently, we have added new sets of tables (Table 15 in Appendix F.1 across all experimental configurations, Table 16 in Appendix F.2 for win-rate performance at each survival scenario variation, and Table 17 in Appendix F.3 for win-rate performance at each causal assumption violation) reporting the percentage of times each method family appears in the Top-1, Top-3, and Top-5 positions according to both CATE RMSE and ATE Bias. This provides a clearer view of how often a method is “best” versus “average”. The accompanying text added to these sections (Appendices in F.1, F.2, and F.3) also help give an overall view of the strength-areas of each method family for different assumption violations and setup.
> >
> > Here we are copying the newly added Table 15, but we encourage the reviewer to also look at Table 16 and 17 and also the text added.
> > | Method Family           | CATE RMSE Top-1 | CATE RMSE Top-3 | CATE RMSE Top-5 | ATE Bias Top-1 | ATE Bias Top-3 | ATE Bias Top-5 |
> > |-------------------------|----------------:|-----------------:|-----------------:|---------------:|---------------:|---------------:|
> > | *Outcome Imputation Methods* |                 |                  |                  |                |                |                |
> > | T-Learner               |             0.0 |              0.0 |              0.0 |            0.0 |           17.5 |           25.0 |
> > | S-Learner               |             0.0 |              2.5 |             12.5 |            5.0 |            7.5 |           27.5 |
> > | X-Learner               |             0.0 |              0.0 |              0.0 |            2.5 |            7.5 |           17.5 |
> > | DR-Learner              |             0.0 |              0.0 |              0.0 |            0.0 |           10.0 |           37.5 |
> > | Double-ML               |            27.5 |             62.5 |             85.0 |            2.5 |           15.0 |           37.5 |
> > | Causal Forest           |             2.5 |             40.0 |             52.5 |            2.5 |           27.5 |           40.0 |
> > | *Direct-Survival Methods*    |                 |                  |                  |                |                |                |
> > | Causal Survival Forest  |            35.0 |             67.5 |             82.5 |           52.5 |           75.0 |           82.5 |
> > | SurvITE                 |            25.0 |             37.5 |             45.0 |           15.0 |           37.5 |           55.0 |
> > | *Survival Meta-Learners*     |                 |                  |                  |                |                |                |
> > | T-Learner-Survival      |             0.0 |             10.0 |             40.0 |           12.5 |           30.0 |           40.0 |
> > | S-Learner-Survival      |            10.0 |             45.0 |             97.5 |            5.0 |           30.0 |           65.0 |
> > | Matching-Survival       |             0.0 |             35.0 |             85.0 |            2.5 |           42.5 |           72.5 |

---

> > > ### Author Response · Authors · 2025-11-22
> > > **Response to Reviewer FuYz [3/4]**
> > >
> > > ### [W3,5: Missing deep direct-survival CATE models; horizon dependence]
> > > - We **have added SurvITE as a representative deep direct-survival CATE baseline** in our benchmark (with a PyTorch implementation for reproducibility). We report its results on the synthetic and semi-synthetic datasets in the revised paper draft (Figure 1-2, Table 3, and Figure 6-7, Figure 8-17, Table 15-17, and Table 31-35 in Appendix). We find SurvITE
> > > - Separately, while the current paper focuses on RMST (with a large horizon) as the main estimand, we now **add horizon-specific survival-probability CATEs** (e.g., at multiple quantiles of the event-time distribution) and **RMST at median event time** to directly analyze time-horizon sensitivity . Please refer to our response to Reviewer yGg8 (the part *Q1. RMST time horizon varying across datasets; need sensitivity analysis and alternative estimands*) for results using RMST based on a different time horizon (median event time) as well as survival probability at multiple horizons (i.e., 25/50/75-percentile of event times).  We also encourage reviewers to take a look at the newly added *Appendix ​​G.3 Additional Experiment Results and Estimands* in our revised paper draft, which includes more details.
> > >
> > > ### [W4: Where do Chapfuwa et al. (2021)–style models fit?]
> > > - These methods directly model counterfactual survival outcomes without an imputation stage or a separate meta-learner wrapper, so they fall under our Direct-Survival CATE Models family. We will clarify this mapping in Section 2 / Appendix D and note how such models plug into our direct-survival evaluation API.
> > >
> > > ### [W6: Add synthetic visualizations]
> > > - Thanks for the great suggestion. In the revised paper draft, we are adding Kaplan-Meier curves (Figure 5 in Appendix A for synthetic datasets and Figure 19 in Appendix G for semi-synthetic datasets) of the survival and censoring distributions for treated and control groups, respectively, to make the generators and the effects of each causal violation more transparent.

---

> > > > ### Author Response · Authors · 2025-11-22
> > > > **Response to Reviewer FuYz [4/4]**
> > > >
> > > > ### [Q1: Why not evaluate against ground-truth hazards?]
> > > > We chose RMST-CATE as the primary estimand because it is widely adopted in causal survival work and yields an absolute, clinically interpretable treatment effect (time gained/lost) within a clinically meaningful horizon. More broadly, we **intentionally avoid making hazard-based estimands** (especially hazard ratios) the main benchmark target. Traditional causal survival analysis often expresses effects as hazard ratios under proportional-hazards–type assumptions, but a substantial literature has pointed out that hazard contrasts can be difficult to interpret causally and clinically, and may obscure absolute benefits or harms—even in randomized studies ([1–5]). In contrast, RMST-type estimands summarize counterfactual time-to-event differences directly, do not hinge on proportional hazards, and provide a holistic measure of treatment impact over time rather than at a single instant.
> > > >
> > > > That said, we agree hazards are important modeling objects. To provide a complementary, time-local view that is closely tied to hazards while retaining interpretability, we are adding horizon-specific survival-probability CATEs at multiple standardized horizons (25/50/75-percentiles of event times). This strengthens the benchmark’s coverage of time-horizon dependence without relying on hazard-ratio targets.
> > > >
> > > > *References:*
> > > >
> > > > [1] Hernán, Miguel A. "The hazards of hazard ratios." Epidemiology 21.1 (2010): 13-15.
> > > >
> > > > [2] Aalen, Odd O., Richard J. Cook, and Kjetil Røysland. "Does Cox analysis of a randomized survival study yield a causal treatment effect?." Lifetime data analysis 21 (2015): 579-593.
> > > >
> > > > [3] Martinussen, Torben, Stijn Vansteelandt, and Per Kragh Andersen. "Subtleties in the interpretation of hazard contrasts." Lifetime Data Analysis 26 (2020): 833-855.
> > > >
> > > > [4] Axelrod, Rachel, and Daniel Nevo. "A sensitivity analysis approach for the causal hazard ratio in randomized and observational studies." Biometrics 79.3 (2023): 2743-2756.
> > > >
> > > > [5] Fay, Michael P., and Fan Li. "Causal interpretation of the hazard ratio in randomized clinical trials." Clinical Trials 21.5 (2024): 623-635.
> > > >
> > > >
> > > > ### [Q2: How should users leverage results on new datasets/models?]
> > > > We believe the actionable insights already provided in Section 4 (and highlighted in our response to W2 above) offer practical guidance for applying these methods to new datasets—for example, by linking observable dataset properties such as censoring severity and potential assumption violations to which method families/variants are likely to be reliable—and they also surface clear gaps that motivate how future models should be developed. In addition, the benchmark is explicitly designed to be extensible: it modularizes data loading, estimators, and evaluation, so users can readily plug in new datasets or new direct/imputation/meta-learner methods and evaluate them under the same unified protocol. We have also refactored the codebase (already updated in the anonymized repo) to introduce three clear base classes, with a step-by-step guide for integrating new estimators, one for each method family (`OutcomeImputationBase`, `DirectSurvivalCATEBase`, and `SurvivalMetaLearnerBase`) that can be inherited to implement new methods. To further support extensions to new settings, we have also added a `DataGeneration` base class that can be adapted to incorporate custom data-generating mechanisms (e.g., alternative hazard functions or more complex treatment policies) while still leveraging our standardized evaluation pipeline.
> > > >
> > > > ---
> > > >
> > > > Thanks again for the detailed feedback. We think these clarifications and additions will substantially improve both coverage and practical value.

---

### Official Review · Reviewer_qN3T · 2025-10-29

**Soundness:** 4
**Presentation:** 3
**Contribution:** 3
**Rating:** 6
**Confidence:** 4

**Summary:**

This work introduces a benchmark for evaluating estimators of the CATE in right-censored survival settings. The benchmark considers several violations or near violations of the assumptions that these methods commonly rely on (overlap, ignorability, etc.). The authors then evaluate a variety of existing learners for survival functions, which they classify into 3 categories: outcome imputation, direct methods, and meta-learners.

**Strengths:**

To my knowledge, this is the first large-scale benchmark for CATE estimation in the context of right-censored outcomes.

The benchmark covers a wide mix of datasets, including semi-synthetic and real ones.

Reports a comprehensive set of metrics.

Provides reproducible code.

**Weaknesses:**

The provided README in the anonymized GitHub doesn't make it clear how to add new learners to the benchmark. For this benchmark to be widely adopted, this should be described (and made as simple as possible).

Minor point, but TMLE can't generally be used to estimate the CATE. The parameter isn't smooth enough for TMLE to be used to estimate it. An exception to this occurs when measuring effect modification with respect to a discrete summary of the baseline covariates (e.g., Stitelman et al., 2010), but that doesn't seem to be your main case of interest.

**Questions:**

The proposed methods seem to require the set of covariates you're conditioning on in the CATE to be the same as the ones to be used to satisfy the ignorability assumptions. This is somewhat restrictive, as it means that using this CATE in the future requires measuring all of X, which may be infeasible. For non-survival CATE estimation, people get around this by instead considering the CATE conditional on a user-specified subvector V of X. The identifiability assumptions remain unchanged, and many methods (e.g., DR-learner) also work in these cases. Can your benchmark be used in these settings as well?

Would it be easy to extend your benchmark to other types of coarsened data - e.g., left-truncated data?

---

> ### Author Response · Authors · 2025-11-22
> **Response to Reviewer qN3T**
>
> We thank the reviewer for recognizing the novelty, breadth of datasets, comprehensive metrics, and strong reproducibility. Below we respond to all points.
>
> ---
>
> ### W1. README does not clearly explain how to add new learners
> We thank the reviewer for identifying this barrier to adoption. To ensure the benchmark is easily extensible, we have refactored the codebase (and updated the anonymized repo) to include three explicit base classes---one for each method family (`OutcomeImputationBase`, `DirectSurvivalCATEBase`, and `SurvivalMetaLearnerBase`) that can be extended for new methods. We have updated the main `README.md` with a step-by-step tutorial on how to inherit from these classes to integrate new estimators.
>
> Furthermore, to support the extension of the benchmark to new settings, we have also provided a `DataGeneration` base class. This allows researchers to easily inject custom data-generating procedures (for instance, introducing different hazard functions or complex treatment policies, while still utilizing our standardized evaluation pipeline).
>
> ### W2. TMLE generally cannot estimate the CATE except in special discrete-modifier cases
>
> Thank you for pointing out this important nuance. You are correct that standard TMLE theory faces challenges with the smoothness required for estimating continuous CATE functions, making it more suitable for subgroup analyses or defined summary measures. We have updated our *Related Work* section to reflect this distinction accurately and to mention other closely related settings that are outside the scope of our setting of interest:
>
> > While our benchmark focuses on static treatments under selection on observables, related work addresses Heterogeneous Treatment Effects in alternative settings. This includes Instrumental Variable approaches for survival (Tchetgen et al., 2015), dynamic treatment regimes (Rudolph et al., 2022; Bates et al., 2022; Rudolph et al., 2023; Cho et al., 2023), and Bayesian machine learning approaches (Chen et al., 2024). Additionally, Targeted Maximum Likelihood Estimation (TMLE) (Stitelman & van der Laan, 2010; Stitelman et al., 2011) offers robust estimation for survival parameters, though primarily for subgroup effects rather than continuous CATE functions.
>
> ### Q1. Conditioning set: Can benchmark support CATE conditional on a subvector V instead of full X?
> Yes, the benchmark can technically support this setting. The data generation process in SurvHTE-Bench yields the full covariate set $X$ required for ignorability. Because the evaluation pipeline is decoupled from the data generation, a user can straightforwardly implement this scenario by partitioning $X$ into the confounding set (used for nuisance parameter estimation, such as propensity scores) and the subvector $V$ (used for the final CATE model).
>
> For example, when using a DR-Learner within our framework, one could pass the full $X$ to the internal propensity and outcome regression models to satisfy ignorability, while restricting the final stage regressor (which estimates the pseudo-outcome) to only condition on $V$. This requires no changes to the data generation code, only to the configuration of the specific learner being evaluated.
>
> ### Q2. Can the benchmark be extended to other coarsened data types, e.g., left-truncated data?
> Thank you for pointing this out. Extending the benchmark to other data types involves two components: data generation and estimator compatibility.
>
> On the *data generation* side, the extension is straightforward: our modular `DataGeneration` class can be easily adapted to simulate left-truncation (by sampling entry times) or interval censoring. However, the primary bottleneck lies in *estimator compatibility*. The vast majority of our implemented methods as well as the off-the-shelf open-source implementations for survival analysis (including the deep learning and forest-based baselines used in this paper) are hard-coded to handle right-censoring only. Benchmarking on left-truncated data would effectively require re-implementing or significantly modifying the core logic of dozens of baseline methods, which is why we limited the current scope to the most prevalent setting of right-censored outcomes.
>
> ---
>
> We again thank the reviewer for their thoughtful and constructive feedback. If there are any remaining concerns or clarifications that would strengthen the paper, we would be very glad to address them. We hope that the revisions above fully resolve the reviewer’s questions and accurately reflect the contributions and scope of our work.

---

### Official Review · Reviewer_Yomt · 2025-10-31

**Soundness:** 4
**Presentation:** 4
**Contribution:** 3
**Rating:** 6
**Confidence:** 4

**Summary:**

The authors propose a benchmark for evaluating models which estimate a heterogeneous treatment effect from survival data. The benchmark consists of 40 synthetic, 6 semi-synthetic, and 2 real datasets. The synthetic and semi-synthetic datasets include ground truth HTEs which realistically violate standard assumptions in causal inference to different degrees. One of the two real datasets, Twins, is conducted so that one twin receives treatment and one does not; as close to a ground truth as can be obtained in reality. The other dataset is from an HIV clinical trial. The authors also implement 52 methods for survival HTE estimation and compare their efficacy on the benchmark. They find that there is no single dominant method, with the best method frequently depending on which causal inference assumptions are violated. Based on their observations, they provide guidelines suggesting which method or class of methods to use in a given scenario.

**Strengths:**

The paper is clearly written and easy to follow.

The proposed benchmark is thorough and does an excellent job of covering many different scenarios of practical relevance for HTE estimation with survival data. The motivation for the construction of the different synthetic datasets (in particular, tying each one to a combination of the "causal configuration" and survival analysis setup) makes it clear exactly what is being tested by each synthetic dataset. The semi-synthetic and real datasets are also constructed sensibly. The semi-synthetic data tests a method's ability to cope with more realistic covariate distributions (as the fully synthetic datasets only use uniformly distributed covariates) while still maintaining access to a ground truth CATE. For the real data, the Twins dataset seems to be the closest possible thing to having a ground truth measurement with real data. Modulating the censoring pattern on the HIV clinical trial dataset is an effective method for testing the robustness of each method to censoring on real data without a ground truth. I found Fig. 4 related to this dataset to be particularly interesting, as it shows that *none* of the existing methods have completely desirable behavior in the case of increasing censoring on real-world data. The CATE either changes drastically under increased censoring, or (as in the case of the CSF) the CATE estimate is simply very insensitive to the covariates.

In addition to providing a consolidated benchmark for HTE estimation with survival data--already useful in its own right to standardize the evaluation of these methods--the authors make good use of their benchmark to derive practical recommendations for the strenghts and weaknesses of existing methods. Access to these results as well as the collected implementations of a large number of relevant baselines should be of great use to other researchers working in this area.

**Weaknesses:**

While the types of CI assumption violations considered are extensive, the ground truth CATEs used for the synthetic data generation (discussed in Appendix A.3) are somewhat limited. In particular, the functional form of the ground truth consists of mostly linear components, with a couple of instances of quadratic terms, square root terms, or threshold discontinuities. Of particular note is that there are no interaction terms between covariates. Examining the effect of more severe nonlinearities, as well as interactions between the covariates, would strengthen the baselines. There are also a large number of presumably hand-picked constants used to define the ground truth. Some justification for how these particular constants were chosen, or else results showing that the conclusions of the paper are robust to the particular choice of constants, would also help greatly. I believe these are the most pressing issues and I would happily raise my score for the paper if they were addressed.

The CATE is defined to be the difference of some function of the survival time with and without the binary treatment. This is a reasonable design choice which is common in causal inference settings. However, in practical settings such as the analysis of clinical trials, treatment is often measured as a change in the underlying hazard function, and it is not immediately clear if this quantity can be represented in the form of equation (1).

Lastly, as the authors acknowledge, the evaluation is limited to binary, static treatments and static covariates, with time-varying or more complicated treatment variables left for future work. While this is a limitation, I agree with the authors that the scope they consider is already of great value.

Typos:
1. Lines 82-83: $T_i$ and $\widetilde{T}_i$ are reversed.

**Questions:**

How were the coefficients chosen for the synthetic data generation? How robust are your conclusions to variations in these coefficients?

---

> ### Author Response · Authors · 2025-11-22
> **Response to Reviewer Yomt [1/3]**
>
> We thank the reviewer for their positive evaluation of clarity, thoroughness, and strong benchmark construction. We are also particularly encouraged by the reviewer’s assessment that the benchmark covers scenarios of practical relevance and that the results will be of great use to researchers.
>
> Below, we address the specific weaknesses and questions raised, particularly focusing on the complexity of the data generation and the justification for parameter choices.
>
> ---
>
> ### W1. Limited ground-truth CATE complexity: mostly linear functions, missing interactions
> We thank the reviewer for this suggestion. We agree that while the initial synthetic data was designed to be simple to isolate specific assumption violations, testing against complex, non-linear dependencies is crucial for a complete evaluation.
>
> Based on this feedback, we have expanded the benchmark by **adding four new semi-synthetic MIMIC configurations**. These configurations systematically vary the covariate dependence mechanisms for both treatment assignment and outcomes, specifically introducing interaction terms and non-linearities:
>
> (Note that our original 5 MIMIC cases while varying censoring rate only considered randomized treatment assignment and the outcome was linearly dependent on the covariates. In the newly added datasets, these degrees of freedom are also perturbed).
>
> 1.  *MIMIC-$vi$:* Linear dependence for Treatment / Linear dependence for Outcome
> 2.  *MIMIC-$vii$:* Linear dependence for Treatment / **Non-linear dependence for Outcome (using interaction terms)**
> 3.  *MIMIC-$viii$:* **Non-Linear dependence for Treatment** / Linear dependence for Outcome
> 4.  *MIMIC-$ix$:* **Non-Linear dependence for Treatment** / **Non-linear dependence for Outcome**
>
> **Data Generation Details:**
>
> We let $X\_{36}$ denote standardized admission age, and define the abnormal-lab burden $S=\sum\_{j=1}^{5}X\_j$, where $X\_{1:5}$ are abnormal laboratory indicators.
>
> * **Non-linear Treatment Assignment:** We use a quadratic assignment mechanism with interactions:
>
>     $$\eta(x)=\beta\_0+\beta\_1X_{36}+\beta\_2S+\beta\_3X\_{36}^2+\beta\_4X\_{36}S,\quad e(x)=\mathrm{logit}^{-1}(\eta(x)).$$
>
> * **Non-linear Outcome:** We generate potential event times $T(w)$ from Poisson models where the mean parameter includes quadratic ($X\_{36}^2$) and interaction ($X\_{36}S$) terms, allowing survival outcomes to vary non-linearly with baseline severity and age.
>
> Full definitions and coefficients are provided in the revised **Appendix G**. These additions push the semi-synthetic suite of datasets to a substantially higher level of complexity to rigorously stress-test estimators.
>
> **Results on running our causal survival methods on these new datasets:**
>
> The results for these new datasets are reported in **Appendix G.3**. As shown in the table below, **Causal Survival Forest** consistently outperforms other baselines, maintaining low error rates even under the complex non-linearities of MIMIC-$ix$. Among survival meta-learners, S-Learner Survival and Matching Survival show stability regarding non-linear treatment assignment but exhibit slight performance degradation when the outcome depends non-linearly on covariates.
>
> | **Method Family**        | **MIMIC-$vi$** | **MIMIC-$vii$** | **MIMIC-$viii$** | **MIMIC-$ix$** |
> |--------------------------|------------------------|--------------------------|--------------------------|----------------------------|
> | *Outcome Imputation Methods*    |                 |                          |                          |                            |
> | **T-Learner** | 7.184 ± 0.052 | 7.374 ± 0.067 | 7.220 ± 0.046 | 7.354 ± 0.048 |
> | **S-Learner** | 7.176 ± 0.048 | 7.308 ± 0.068 | 7.197 ± 0.049 | 7.275 ± 0.061 |
> | **X-Learner** | 7.182 ± 0.053 | 7.318 ± 0.069 | 7.203 ± 0.045 | 7.273 ± 0.044 |
> | **DR-Learner**| 7.145 ± 0.054 | 7.295 ± 0.061 | 7.167 ± 0.046 | 7.263 ± 0.045 |
> | **Double-ML** | 7.127 ± 0.051 | 7.259 ± 0.072 | 7.147 ± 0.049 | 7.226 ± 0.056 |
> | **Causal Forest** | 7.142 ± 0.052 | 7.288 ± 0.068 | 7.162 ± 0.047 | 7.247 ± 0.043 |
> | *Direct-Survival Methods*    |     |     |         |                       |
> | **Causal Survival Forest** | 7.123 ± 0.048          | 7.281 ± 0.064            | 7.149 ± 0.045            | 7.227 ± 0.054              |
> | **SurvITE**                        | 7.243 ± 0.154          | 7.378 ± 0.112    | 7.268 ± 0.117             | 7.347 ± 0.078              |
> | *Survival Meta-Learners*    |      |       |     |      |
> | **T-Learner Survival**      | 7.465 ± 0.364   | 7.487 ± 0.179      | 7.266 ± 0.053      | 7.465 ± 0.254              |
> | **S-Learner Survival**     | 7.183 ± 0.051   | 7.345 ± 0.066      | 7.198 ± 0.042      | 7.283 ± 0.049              |
> | **Matching Survival**      | 7.219 ± 0.060   | 7.393 ± 0.074      | 7.240 ± 0.043       | 7.357 ± 0.046              |
>
> ~~Note: Imputation-based methods are currently running for these new datasets; final results will be updated in the paper.~~ *Updated now!*

---

> > ### Author Response · Authors · 2025-11-22
> > **Response to Reviewer Yomt [2/3]**
> >
> > ### W2.  & Q1. Hand-picked constants in synthetic data generation process; need justification or robustness
> >
> > This is a critical point regarding the experimental design. The coefficients in our synthetic data generation were chosen to achieve specific, distinct regimes of "difficulty" to ensure the benchmark provided enough "dynamic range" in how well different methods performed. The constants in our generators follow the synthetic setups in Cui et al. (2023) and Meir et al. (2025), and were set with the practical goal of spanning distinct, interpretable regimes, such as balanced vs. sparse treatment prevalence, low/medium/high censoring levels, and meaningful degrees of assumption stress. Our simulation design follows the strategies in Cui et al. (2023) and Meir et al. (2025), with targeted adjustments to (i) separate causal configurations from survival scenarios, and (ii) introduce informative censoring in a modular way. This lets us vary one axis (e.g., censoring mechanism or confounding strength) while holding others fixed, which prior simulations by Cui et al. and Meir et al. did not systematically provide.
> >
> > At a high level, we tune constants to achieve intended difficulty levels:
> >
> > - **Censoring regimes via event-time and censoring-time scales.**
> > For example, Scenarios B vs. D both use AFT event-time models but with different intercepts and treatment coefficients.
> > In Scenario B we have:
> >
> > $$
> > \log T(w) = -1.85 - 0.8 \cdot 1(X\_{1} < 0.5) + 0.7\sqrt{X\_{2}} + 0.2X\_{3}  + [0.7 - 0.4 \cdot 1(X\_{1} < 0.5) - 0.4\sqrt{X\_{2}}] \cdot W + \epsilon + \eta,
> > $$
> >
> > whereas in Scenario D we have:
> >
> > $$
> > \log T(w) = 0.3 - 0.5 \cdot 1(X\_{1} < 0.5) + 0.5\sqrt{X\_{2}} + 0.2X\_{3}  + [1 - 0.8 \cdot 1(X\_{1} < 0.5) - 0.8\sqrt{X\_{2}}] \cdot W + \epsilon + \eta
> > $$
> >
> > By shifting the constant term (e.g., -1.85 in B vs. 0.3 in D), we increase typical event times in D; combined with the corresponding censoring models, this yields higher censoring in D than B. Intuitively, larger event-time scales and/or smaller censoring-time scales increase censoring, and vice versa. Similar intercept adjustments are used across scenarios to create low/medium/high censoring settings while keeping covariate functional forms comparable.
> >
> >
> > - **Treatment prevalence and confounding strength via propensity-score calibration.**
> > Outside `RCT-5` (where we intentionally set $\mathbb{E}[W]\approx 0.05$), we aim for balanced prevalence near 0.5 so that differences across causal configurations reflect assumption changes rather than trivial prevalence shifts. For instance, in OBS with observed confounding (`OBS-CPS`), we have the propensity score function being:
> > $$
> > e(X) = \frac{1 + \text{Beta}(X\_{1}; 2, 4)}{4},
> > $$
> > versus in OBS with unobserved confounding (`OBS-Uconf`), the propensity score function is:
> > $$
> > e(X, U) = \frac{1 + \text{Beta}(0.3X\_{1} + 0.7U\_{1}; 2, 4)}{4},
> > $$
> > We choose weights in the propensity score so that both yield similar treatment rates, while `OBS-UConf` introduces dependence on the latent $U\_1$. Specifically, in `OBS-CPS`, only $X\_1$ shows up in the equation with the weight being 1; in `OBS-UConf`, both $X\_1$ and $U\_1$ are in the equation with weight 0.3 and 0.7 and they sum to one to preserve the input range and prevalence comparability. Adjusting these weights would make confounding weaker/stronger; we fix them to yield a meaningful but controlled violation.
> >
> >
> > - **Effect magnitudes via treatment coefficients.**
> > The coefficients on W (or W-interactions) control the size and heterogeneity of treatment effects. For simplicity and comparability, we keep these within a moderate range across scenarios rather than expanding effect sizes in parallel with censoring/assumption changes.
> >
> >
> > We have added a short clarifying paragraph to the end of Appendix A to make these calibration goals explicit (treatment rate targets, censoring-rate tiers, and how constants control each). We acknowledge that these choices are not unique and that alternative parameterizations could also be reasonable; our aim is not to claim an optimal or exhaustive simulation, but to provide a principled and reproducible starting point that covers a wide range of survival HTE settings.
> >
> > Additionally, to improve transparency and demonstrate robustness, we have added *Kaplan-Meier curves* for the survival and censoring distributions for treated and control groups to Appendix A. These plots show that our chosen constants produce sensible survival patterns (e.g., capturing features like crossing versus proportional hazards) without introducing any unexpected quirks.

---

> > > ### Author Response · Authors · 2025-11-22
> > > **Response to Reviewer Yomt [3/3]**
> > >
> > > ### W3. Interpretation: CATE defined as difference in function of survival time vs hazard-based effects
> > > The reviewer raises a valid point that clinical trials often utilize Hazard Ratios (HR). However, we deliberately chose Restricted Mean Survival Time (RMST) and survival probabilities as our estimands for several methodological and practical reasons:
> > >
> > > 1.  HRs are most natural under the Proportional Hazards (PH) assumption. However, our benchmark includes non-PH settings and datasets where PH is violated. In non-PH settings, the HR varies over time, making a single scalar “treatment effect” mathematically ambiguous and difficult to define as a ground truth [1, 2]. RMST and survival probability differences remain valid and interpretable estimands regardless of whether the PH assumption holds.
> > > 2.  Hazard ratios can be difficult to interpret in the causal inference setting, particularly regarding the absolute magnitude of benefit [3]. RMST provides a direct measure of (e.g., time to event delayed), which is an absolute metric often more actionable for decision-making.
> > > 3.  In the potential outcomes framework, effects are naturally defined on the outcomes themselves (the event times $T(1)$ vs $T(0)$) rather than the rates.
> > >
> > >
> > > Due to this, while HRs have been common in the past, since RMST allows for a unified evaluation metric across diverse survival distributions where proportional hazards may not hold, in our benchmark we mainly focus on the latter.
> > >
> > > We should also note that in response to other reviewers, we have also added CATE based on survival probability evaluated at various time horizons (see Appendix G.3.2 for survival-probability-based CATEs at the 25th, 50th, and 70th event-time percentiles as horizons). Given the relation between hazard and survival probability, these probability-based CATEs provide a conceptually comparable (but more interpretable) alternative to hazard-based effects while remaining well-defined under non-PH.
> > >
> > > *References:*
> > >
> > > [1] Aalen, Odd O., Richard J. Cook, and Kjetil Røysland. "Does Cox analysis of a randomized survival study yield a causal treatment effect?." Lifetime data analysis 21 (2015): 579-593.
> > >
> > > [2] Martinussen, Torben, Stijn Vansteelandt, and Per Kragh Andersen. "Subtleties in the interpretation of hazard contrasts." Lifetime Data Analysis 26 (2020): 833-855.
> > >
> > > [3] Hernán, Miguel A. "The hazards of hazard ratios." Epidemiology 21.1 (2010): 13-15.
> > >
> > > ### W4. Scope limitation: binary, static treatments and static covariates
> > > We appreciate the reviewer's agreement that the current scope is already of great value. We agree that time-varying treatments are an important frontier. To explicitly acknowledge the path toward more complex evaluations, we have expanded our **Discussion/Limitations** section as follows:
> > >
> > > > We recognize that in real-world applications, assumption violations often exist on a continuum of severity. Future extensions of our benchmark could incorporate graded sensitivity analyses, such as varying the magnitude of unmeasured confounding (e.g., via Rosenbaum's $\Gamma$) or the degree of overlap violation. This would allow for a more granular “dose-response” analysis to pinpoint the exact thresholds at which specific estimators break down.
> > >
> > > ### Typo1. Lines 82–83: variables reversed
> > > Thanks for pointing this out. This has been fixed.
> > >
> > > ---
> > >
> > > We sincerely appreciate the reviewer’s thoughtful and constructive feedback! If there are any remaining concerns or any additional details that would help the reviewer feel confident increasing their score, we would be more than happy to address them.

---

### Official Review · Reviewer_yGg8 · 2025-10-31

**Soundness:** 2
**Presentation:** 3
**Contribution:** 2
**Rating:** 4
**Confidence:** 3

**Summary:**

The authors propose survHTE-Bench, a benchmark for estimating heterogeneous treatment effects (HTE) with right-censored survival outcomes. The paper summaries 52 estimators across three families, and evaluate them on 40 synthetic datasets, 6 semi-synthetic datasets, 2 real datasets. Results are reported through Borda count rankings, RMSE metrics.

**Strengths:**

1. The paper is well written with a clear scope. The benchmark topic is important and interesting in the study of treatment effect on health science datasets.

2. The paper considers a large set of HTE estimators, the evaluations are done on multiple settings.

3. The paper includes a very complete reproducibilities resources.

**Weaknesses:**

1. A complete benchmark on HTE with survival datasets is surely needed in the community. There are some benchmark papers on HTE (not on survival setting), including [Crabbé, J., et al. 2022], [Shimoni, Y., et al. 2018], [Kapkiç, A. et al. 2024] and others. However, extending the benchmark on HTE from complete datasets to right-censored datasets can be a weak improvement. There can be overlaps among those benchmarks. I believe a comprehensive benchmark named by survHTE-Bench should include methods and datasets with all types of censoring.

Crabbé, J., Curth, A., Bica, I., & Van Der Schaar, M. (2022). Benchmarking heterogeneous treatment effect models through the lens of interpretability. Advances in Neural Information Processing Systems, 35, 12295-12309.

Shimoni, Y., Yanover, C., Karavani, E., & Goldschmnidt, Y. (2018). Benchmarking framework for performance-evaluation of causal inference analysis. arXiv preprint arXiv:1802.05046.

Kapkiç, A., Mandal, P., Wan, S., Sheth, P., Gorantla, A., Choi, Y., ... & Candan, K. S. (2024, October). Introducing causalbench: A flexible benchmark framework for causal analysis and machine learning. In Proceedings of the 33rd ACM International Conference on Information and Knowledge Management (pp. 5220-5224).

2. Synthetic data design: The data generation process is too simple and not realistic, especially in health science. For example, all five covariates are **independently** generated from Uniform(0,1); treatment assignment is simple and have weak dependence on covariates. The benefits of using simulation studies is we can include all possible settings with ground truth, therefore, the data generation should include more settings, such as low/high dimensional, unbalanced cases, correlated features, causal related features, various tail behaviors, non-smooth hazards, treatment policy with feedbacks and so on.

3. Semi-synthetic data design: I have the similar comments as in weakness 2. The treatment assignment is independent and Bernoulli(0.5). Censoring times are independently drawn from Poisson($\lambda_c$), $\lambda_c$ is constant.  The potential outcomes are Poisson with means linearly depends either on $(S,X_{36})$ or $X_{36}$. Semi-synthetic datasets are combinations of real data with reasonable synthetic data, the simple setting on treatment assignment and censoring generation can be non-realistic.

4. Synthetic data design: The violation severity is binary. Authors acknowledge violations are binary (present or absent). But realistic applications require graded severities (e.g., overlap quantified by min propensity, MNAR censoring strength). It would be great if you can provide dose response grids of violation magnitude.

5. I think Twins “ground truth” for survival is debatable. The Twins setting is typically used for counterfactual outcomes like birthweight; “ground truth” for survival times under treatment and control is not fully inherent and is partly constructed (randomized treatment assignment and censoring). The authors should justify why observed twin outcomes constitute true counterfactual survival outcomes and clarify how censoring was constructed (84.8% is extremely high).

6. The benchmark considers multiple assumptions and considers the cases when the assumptions hold or not (binary). However, there are cases when certain assumption is not simply satisfied or not satisfied. In application, we also want sensitivity analysis, for example, if those assumptions are a little bit wrong, how much would the conclusions change? There are methods for such sensitivity analysis, for example, Rosenbaum’s $\Gamma& for unmeasured cofounding.

7. Methods: The benchmark includes three families of models. This is good. However, there are models that are closely related, but are not included. For example, TMLE, IV-based survival HTE, Dynamic treatment cases, Bayesian survival HTE. Here I list several references.

Stitelman, O. M., Wester, C. W., De Gruttola, V., & van der Laan, M. J. (2011). Targeted maximum likelihood estimation of effect modification parameters in survival analysis. The international journal of biostatistics, 7(1), 19.

Tchetgen, E. J. T., Walter, S., Vansteelandt, S., Martinussen, T., & Glymour, M. (2015). Instrumental variable estimation in a survival context. Epidemiology, 26(3), 402-410.

Cho, H., Holloway, S. T., Couper, D. J., & Kosorok, M. R. (2023). Multi-stage optimal dynamic treatment regimes for survival outcomes with dependent censoring. Biometrika, 110(2), 395-410.

Chen, X., Harhay, M. O., Tong, G., & Li, F. (2024). A Bayesian machine learning approach for estimating heterogeneous survivor causal effects: applications to a critical care trial. The annals of applied statistics, 18(1), 350.

8. Borda count ranking averages hide effect sizes and are sensitive to the method set and dataset composition. It would be better to add uncertainty quantification.

9. For a benchmark paper, the evaluation of using only RMSE is restricted. Quantile based treatment effect or other rank based treatment effect would be robust in certain cases. For  survival data, fixed time survival probability difference can be meaningful in clinic study. These alternative considerations would improve the novelty of the benchmark.

**Questions:**

1. The benchmark defines CATE in terms of RMST up to a time horizon $h>0$ time horizon $h$. If I am correct, $h$ is the maximum observed time per dataset. If this is the case, $h$ varies across datasets (and splits for some methods), so different runs are literally evaluating different estimates. When censoring is heavy or tails differ, this changes both identifiability and difficulty in a way unrelated to the method, it can confound any cross dataset ranking. Then, there should be a sensitivity report with respect to $h$ and include additional estimates such as median survival. Please clarify this.

2.	The benchmark mentions serval references within the direct-survival CATE models in section 2. but  why you only include and evaluate causal survival forest in this family in your experiments? Am I missing the results of other models, for example SurvITE in [Alicia C., et al. 2021 NeurIPS]?

---

> ### Author Response · Authors · 2025-11-22
> **Response to Reviewer yGg8 [1/11]**
>
> We thank the reviewer for their positive remarks regarding the clarity of writing, importance of the benchmark topic, breadth of estimators, and reproducibility. Below we provide point-by-point responses.
>
> ---
>
> ### W1. Benchmark novelty relative to prior HTE benchmarks (Crabbe 2022, Shimoni 2018, Kapkic 2024)
> Thank you for this thoughtful comment and for pointing to prior HTE benchmarks. We fully agree that general HTE benchmarking has a strong foundation, including interpretability-focused benchmarks (e.g., Crabbé et al., 2022), broad evaluation frameworks (Shimoni et al., 2018), and platform-style benchmarking infrastructures (Kapkiç et al., 2024).  However, our benchmark addresses a distinct and substantially harder setting that these works do not cover: **heterogeneous treatment effects under right-censored time-to-event outcomes**. Extending HTE evaluation from complete outcomes to survival data is not a minor patch—right censoring fundamentally changes (i) what is observed, (ii) what must be estimated (survival curves/hazards over time rather than scalar outcomes), and (iii) how identification assumptions interact with learning (e.g., censoring mechanisms, time horizons, and survival-model misspecification). To our knowledge, none of the existing HTE benchmarks provide (a) survival-specific estimands with known ground truth CATEs, (b) systematic causal-survival assumption violations (confounding/positivity/informative censoring) within survival settings, or (c) a unified implementation and comparison of survival-CATE method families.
>
> More concretely, the cited benchmarks primarily rely on semi-synthetic complete-outcome setups and focus on building general evaluation scaffolds, with limited survival-specific modeling or result-driven insights.  In contrast, our contribution is survival-CATE-centric: we (1) **organize the disparate causal-survival HTE literature into three method families** and provide modular implementations across families; (2) **propose a survival-aware synthetic generator that varies both survival dynamics and causal assumptions**, enabling controlled ground-truth HTEs under right censoring; and (3) **run large-scale comparisons to extract setting-specific insights** about when different survival-HTE approaches work best.
>
> We have now added the following text in the revised draft to cite related benchmarks in Section 1:
> > While there is a growing benchmarking literature for treatment-effect heterogeneity in fully observed outcomes (e.g., Crabb´e et al. (2022); Shimoni et al. (2018); Kapkic¸ et al. (2024)) and recent benchmarks for survival ATE estimation (e.g., Voinot et al. (2025)), to our knowledge, there is not yet any benchmark for survival HTE estimation under right-censoring. This missing piece motivates our focus on heterogeneous effects in censored time-to-event data.
>
> We also appreciate the suggestion that a comprehensive SurvHTE benchmark should cover “all types of censoring”. We agree this is an important long-term goal. At the same time, covering right-censoring alone already requires modeling survival processes, counterfactuals over time, and multiple censoring/causal violations jointly, and, to our knowledge, no prior benchmark has done this systematically.  We therefore view SurvHTE as a first initiative focused on the dominant and most practically relevant censoring regime in biomedical and observational studies (right censoring), while keeping the framework modular so future work can add left/interval censoring, different kinds of truncation, and/or competing risks as extensions.

---

> > ### Author Response · Authors · 2025-11-22
> > **Response to Reviewer yGg8 [2/11]**
> >
> > ### W2. Synthetic data design: too simple and not realistic
> > First off, our benchmark includes semi-synthetic and real data experiments in addition to the purely synthetic data experiments precisely because we understand that synthetic data alone could be considered too simplistic. That said, the synthetic data part of our benchmark serves an important purpose. Especially since right now there is no existing comprehensive benchmark on survival HTE approaches and, on the theory side, the community's understanding of when and why these methods work is extremely limited (known theoretical results here make a lot of assumptions that are commonly violated in real data), controlled experiments–even if they are simple–can help the community better understand whether methods that have been developed are breaking down in unexpected ways.
> >
> > To this end, the primary goal of our synthetic datasets is to provide a systematic testbed where specific causal assumptions (e.g. ignorability, positivity, ignorable censoring) are violated in isolation. This controlled environment allows us and future users of our dataset and benchmark paper to pinpoint exactly where a method fails and to then try to answer why (something that becomes impossible in highly complex “realistic” simulations where you have all sorts of non-smooth hazards, feedback loops, and tail behaviours occurring simultaneously). Furthermore, a subset of our synthetic data experiments are based on existing synthetic data experiments established in other causal survival analysis papers (e.g., the ones by Cui et al. (2023) and Meir et al. (2025) that are already cited in our paper).
> >
> > Of course, synthetic data alone is not enough, and we agree that complexity is essential for evaluating real-world utility. Our benchmark already includes settings that address the reviewer’s concerns regarding dimensionality and correlation:
> >
> > 1. **High-dimensionality:** We acknowledge that the number of covariates we use are moderate, however it is comparable to, or exceeds, the feature space sizes found in most recent causal survival literature. For instance the AIDS dataset used by methods like Causal Survival Forest [1] and MISTR [2] contains only 12 covariates, the survival meta-learner methods used by [3] utilize 17, and SurvITE[4] has 39. In our benchmark, while the synthetic data is low-dimensional to facilitate the analysis mentioned earlier, our semi-synthetic and real suite includes higher dimensional covariates: ACTG (23 covariates), 9 different MIMIC datasets (36 covariates), Twins dataset (39 covariates after preprocessing). We leave the development of semi-synthetic datasets with significantly higher dimensionality (e.g., using omics data from [5]) for future work and extensions of our current benchmark.
> >
> > 2. **Correlated Features:** Specifically for the MIMIC datasets, the covariates preserve the complex correction structure of real electronic health records (EHR). We explicitly visualize these correlations in Figure 18 of the Appendix.
> >
> > Regarding "unbalanced cases, various tail behaviors, non-smooth hazards, and treatment policy with feedbacks": While we acknowledge these are valuable variations, they represent an expanding scope that goes beyond the foundational causal survival assumptions we aim to test. However, to support the community in exploring these directions, we have designed our codebase to be modular. We have now provided a `DataGeneration` base class in the updated (anonymized) repo that allows researchers to easily inject custom data generating procedures for different hazard functions or treatment policies while utilizing our standardized evaluation pipeline.
> >
> > *References:*
> >
> > [1] Cui, Yifan, et al. "Estimating heterogeneous treatment effects with right-censored data via causal survival forests." Journal of the Royal Statistical Society Series B: Statistical Methodology 85.2 (2023): 179-211.
> >
> > [2] Meir, Tomer, Uri Shalit, and Malka Gorfine. "Heterogeneous Treatment Effect in Time-to-Event Outcomes: Harnessing Censored Data with Recursively Imputed Trees." arXiv preprint arXiv:2502.01575 (2025).
> >
> > [3] Noroozizadeh, Shahriar, et al. "The Impact of Medication Non-adherence on Adverse Outcomes: Evidence from Schizophrenia Patients via Survival Analysis." Conference on Health, Inference, and Learning (2025): 573–609.
> >
> > [4] Curth, Alicia, Changhee Lee, and Mihaela van der Schaar. "Survite: Learning heterogeneous treatment effects from time-to-event data." Advances in Neural Information Processing Systems 34 (2021): 26740-26753.
> >
> > [5] Drysdale, Erik. "SurvSet: An open-source time-to-event dataset repository." arXiv preprint arXiv:2203.03094 (2022).

---

> > > ### Author Response · Authors · 2025-11-22
> > > **Response to Reviewer yGg8 [3/11]**
> > >
> > > ### W3. Semi-synthetic data design: similar concerns as W2; simple treatment and censoring
> > > While our initial semi-synthetic design prioritized high-dimensional and correlated features with varying censoring rates, we agree with the reviewer that by increasing the complexity of the data-generating mechanisms we can enhance the benchmark’s realism.
> > >
> > > First, we clarify that the ACTG semi-synthetic dataset used in our submission does not use a simple Bernoulli treatment assignment. Instead, the treatment is dependent on the covariates. We have now added the explicit generation equations to Appendix G.1.1 to clarify this dependency (also updated in the paper):
> > >
> > > >
> > > > We simulate potential outcomes according to a Gompertz-Cox distribution with selection bias from a simple logistic model for $P(A=1| X=x )$ and AFT-based censoring mechanism.  Below is our generative scheme:
> > > >
> > > >
> > > > $$X = \text{ACTG covariates}$$
> > > > $$P(A=1|X=x) = \frac{1}{b} \times \left(a + \sigma\left( \eta ({\rm AGE} - \mu_{\rm AGE} + {\rm CD40} - \mu_{\rm CD40}) \right) \right)$$
> > > > $$ U  \sim {\rm Uniform} (0, 1 )$$
> > > > $$T_A  =  \frac{1}{\alpha_A} \log \left[1 - \frac{\alpha_A \log U}{ \lambda_A  \exp\left( x ^T  \beta_A\right)  }  \right]$$
> > > > $$\log C  \sim {\rm Normal} (\mu_c, \sigma_c^2)$$
> > > > $$Y = \min(T_A, C)$$
> > > >
> > > > where $\{ \beta_A, \alpha_A, \lambda_A, b, a, \eta, \mu_c, \sigma_c \}$ are hyper-parameters and $ \{\mu_{\rm AGE},  \mu_{\rm CD40}\}$ are the means for age and CD40 respectively.
> > > >
> > >
> > > Second, to directly address the reviewer’s concern regarding the simplicity of the MIMIC setup, we have now expanded the benchmark by **adding four new semi-synthetic MIMIC configurations**. These configurations systematically vary the covariate dependence mechanisms to treatment assignment and outcome:
> > > 1. *MIMIC-$vi$:* Linear dependence to Treatment / Linear dependence to Outcome
> > > 2. *MIMIC-$vii$:* Linear dependence to Treatment / Non-linear dependence to Outcome (using interaction terms)
> > > 3. *MIMIC-$viii$:* Non-Linear dependence to Treatment / Linear dependence to Outcome
> > > 4. *MIMIC-$ix$:* Non-Linear dependence to Treatment / Non-linear dependence to Outcome
> > >
> > > We provide the equation to generate treatment assignment and event/censoring time below.
> > >
> > > **Notation shared by MIMIC-$vi$–$ix$.** Let ($X_{36}$) denote standardized admission age, and define the abnormal-lab burden
> > > $$
> > > S=\sum_{j=1}^{5}X_j,\qquad S\in{0,1,\dots,5},
> > > $$
> > > where $X_{1:5}$ are five abnormal laboratory indicators. Thus, $S$ counts the number of abnormal labs at baseline, while $X_{36}$ captures age on a standardized scale.
> > >
> > > **Covariate-dependent treatment assignment.** We consider two propensity score families:
> > >
> > > - **Linear assignment (no interactions):**
> > > $$
> > >   \eta(x)=\alpha_0+\alpha_1X_{36}+\alpha_2S,\quad
> > >   e(x)=\Pr(W=1\mid X=x)=\sigma(\eta(x)).
> > > $$
> > >
> > > - **Non-linear assignment (quadratic + interaction):**
> > > $$
> > >   \eta(x)=\beta_0+\beta_1X_{36}+\beta_2S+\beta_3X_{36}^2+\beta_4 X_{36}S,\quad
> > >   e(x)=\Pr(W=1\mid X=x)=\mathrm{logit}^{-1}(\eta(x)).
> > > $$
> > >
> > >   We clip $e(x)$ to $[0.05,0.95]$ to preserve overlap, and choose coefficients so treatment prevalence remains close to $0.5$ while inducing meaningful confounding through $X_{36}$ and $S$.
> > >
> > > **Event-time and censoring mechanisms.** Potential event times and censoring times are generated from Poisson models (identity link), with means depending on $X_{36}$ and $S$ and clipped below at 1 to ensure positivity:
> > > $$
> > > T(0)\sim \mathrm{Poisson}(\mu_0(X)),\quad
> > > T(1)\sim \mathrm{Poisson}(\mu_1(X)),\quad
> > > C\sim \mathrm{Poisson}(\lambda_c(X)).
> > > $$
> > > We use two mean-structure families: a **linear** dependence on $(X_{36},S)$, and a **non-linear** dependence that additionally includes $X_{36}^2$ and the interaction $X_{36}S$. These mechanisms allow survival outcomes and censoring to vary non-linearly with baseline severity (abnormal labs) and age, yielding more heterogeneous and realistic treatment effects.
> > >
> > > Full definitions and coefficient choices are provided in **Appendix G** (new text highlighted in blue). We believe these additions push the semi-synthetic suite to a substantially higher level of complexity and enable a more rigorous evaluation of estimators’ robustness to complex confounding and heterogeneous survival effects.

---

> ### Author Response · Authors · 2025-11-22
> **Response to Reviewer yGg8 [4/11]**
>
> ### W3. New Semi-synthetic data design results *[continued]*
>
> The results provided below for these new datasets are reported in Appendix G.3.1 and show that that Causal Survival Forest consistently outperforms other baselines, maintaining the lowest error rates even as we introduce non-linear dependencies and interaction terms in MIMIC-$ix$. Among survival meta-learners, S-Learner Survival and Matching Survival show stability with respect to non-linear treatment assignment, but exhibit slight performance degradation when the outcome depends non-linearly on the covariates.
>
> | **Method Family**        | **MIMIC-$vi$** | **MIMIC-$vii$** | **MIMIC-$viii$** | **MIMIC-$ix$** |
> |--------------------------|------------------------|--------------------------|--------------------------|----------------------------|
> | *Outcome Imputation Methods*    |                 |                          |                          |                            |
> | **T-Learner** | 7.184 ± 0.052 | 7.374 ± 0.067 | 7.220 ± 0.046 | 7.354 ± 0.048 |
> | **S-Learner** | 7.176 ± 0.048 | 7.308 ± 0.068 | 7.197 ± 0.049 | 7.275 ± 0.061 |
> | **X-Learner** | 7.182 ± 0.053 | 7.318 ± 0.069 | 7.203 ± 0.045 | 7.273 ± 0.044 |
> | **DR-Learner**| 7.145 ± 0.054 | 7.295 ± 0.061 | 7.167 ± 0.046 | 7.263 ± 0.045 |
> | **Double-ML** | 7.127 ± 0.051 | 7.259 ± 0.072 | 7.147 ± 0.049 | 7.226 ± 0.056 |
> | **Causal Forest** | 7.142 ± 0.052 | 7.288 ± 0.068 | 7.162 ± 0.047 | 7.247 ± 0.043 |
> | *Direct-Survival Methods*    |                 |                            |                               |                                  |
> | **Causal Survival Forest** | 7.123 ± 0.048          | 7.281 ± 0.064            | 7.149 ± 0.045            | 7.227 ± 0.054              |
> | **SurvITE**                        | 7.243 ± 0.154          | 7.378 ± 0.112            | 7.268 ± 0.117             | 7.347 ± 0.078              |
> | *Survival Meta-Learners*    |                 |                            |                               |                                  |
> | **T-Learner Survival**        | 7.465 ± 0.364          | 7.487 ± 0.179            | 7.266 ± 0.053            | 7.465 ± 0.254              |
> | **S-Learner Survival**       | 7.183 ± 0.051          | 7.345 ± 0.066            | 7.198 ± 0.042            | 7.283 ± 0.049              |
> | **Matching Survival**        | 7.219 ± 0.060          | 7.393 ± 0.074            | 7.240 ± 0.043            | 7.357 ± 0.046              |
>
> ~~Note: Our imputation-based methods are still running for these new datasets. We will update both here and in the paper as soon as we get our final results for those methods on these new datasets.~~ *Updated now!*

---

> > ### Author Response · Authors · 2025-11-22
> > **Response to Reviewer yGg8 [5/11]**
> >
> > ### W4. Synthetic violation severity: binary rather than graded — and — W6. Assumption checks are binary
> > We agree with the reviewer that real-world violations often occur on a continuum (e.g. degrees of overlap) rather than as binary states.
> >
> > Our decision to focus on binary violations in this benchmark was strategic: we aimed to establish a clear “stress test” to determine if methods fail under explicit assumption violations before mapping the precise “dose-response” curve of failure. Our results indicate that many state-of-the-art methods struggle significantly even under these binary settings (e.g., the severe performance drops observed in `OBS-NoPos`). We believe that characterizing these fundamental failure modes is a necessary prerequisite to fine-grained sensitivity analysis.
> >
> > While we did not run full grids for causal violations to keep the scope manageable, we note that our benchmark **does** provide graded severity along the axis of censoring. Our Survival Scenarios (A through E) systematically vary censoring rates from <30% to >80%. This allows us to observe a “dose-response” in performance degradation relative to information loss, even if the causal violations themselves remain binary. We have added a discussion of graded causal violations to our **Limitations** section as an important direction for future extensions (see below).
> >
> > Regarding the binary assumption checks, we agree that sensitivity analysis (e.g., determining *how* much an assumption must be violated before conclusions change) is a critical component of applied causal analysis. However, the primary contribution of SurvHTE-Bench is to benchmark estimators under *known* structural violations to determine which methods are robust to specific failure modes (e.g., does Method X fail when ignorability is broken?).
> >
> > While we have focused on discrete violations to create a manageable and interpretable grid of scenarios, we agree that expanding this to continuous sensitivity parameters (like Rosenbaum's $\Gamma$) is a valuable direction. We have added this to our Limitations and Future Work section, noting that the current binary framework serves as the starting foundational "stress test" for survival CATE estimators.
> >
> > We have added the following to our limitations section that already covered mention binary nature of violation beforehand:
> > >
> > > We recognize that in real-world applications, assumption violations often exist on a continuum of severity. Future extensions of our benchmark, could incorporate graded sensitivity analyses, such as varying the magnitude of unmeasured confounding (e.g., via Rosenbaum's $\Gamma$) or the degree of overlap violation. This would allow for a more granular “dose-response” analysis to pinpoint the exact thresholds at which specific estimators break down.
> > >

---

> > > ### Author Response · Authors · 2025-11-22
> > > **Response to Reviewer yGg8 [6/11]**
> > >
> > > ### W5. Twins dataset: ground-truth validity and construction of censoring
> > > We acknowledge the reviewer’s point that the “ground-truth” in the Twins dataset is a construct based on the assumption that twins are counterfactuals of one another, which is indeed debatable. However, due to the fundamental problem of causal inference, real-world datasets with *any* form of ground-truth are exceptionally rare. The Twins dataset with its limitation, remains one of the few standard proxies accepted in the causal inference community for benchmarking counterfactual estimation in both standard [1,2] and survival [3–5] settings.
> > >
> > > To increase our transparency on this limitation, we have added a disclaimer regarding this limitation to the dataset description in Appendix H.1:
> > >
> > > >
> > > > While the Twins dataset is a widely used benchmark, we note that treating twins as perfect counterfactuals at the very best is an approximation. The “ground-truth” relies on the assumption that the unobserved potential outcome of one twin is identical to the observed of their sibling, which in reality may not fully capture genetic or environmental heterogeneity.
> > > >
> > >
> > > Regarding the censoring rate of 84.8% being high, this is a common condition in administrative survival data, where long-term follow-up is rare (e.g., the Surveillance, Epidemiology, and End Results (SEER) dataset appearing in [6–8] have approximately 80% censoring rate). Inclusion of this dataset in our benchmark enables our methods to be stress-tested in this common survival analysis condition as well.
> > >
> > > *References:*
> > >
> > > [1] Louizos, Christos, et al. "Causal effect inference with deep latent-variable models." Advances in neural information processing systems 30 (2017).
> > >
> > > [2] Du, Xin, et al. "Adversarial balancing-based representation learning for causal effect inference with observational data." Data Mining and Knowledge Discovery 35.4 (2021): 1713-1738.
> > >
> > > [3] Curth, Alicia, Changhee Lee, and Mihaela van der Schaar. "Survite: Learning heterogeneous treatment effects from time-to-event data." Advances in Neural Information Processing Systems 34 (2021): 26740-26753.
> > >
> > > [4] Curth, Alicia, and Mihaela Van der Schaar. "On inductive biases for heterogeneous treatment effect estimation." Advances in Neural Information Processing Systems 34 (2021): 15883-15894.
> > >
> > > [5] Curth, Alicia, et al. "Really doing great at estimating cate? a critical look at ml benchmarking practices in treatment effect estimation." Thirty-fifth conference on neural information processing systems datasets and benchmarks track (round 2). 2021.
> > >
> > > [6] Danks, Dominic, and Christopher Yau. "Derivative-based neural modelling of cumulative distribution functions for survival analysis." International Conference on Artificial Intelligence and Statistics. PMLR, 2022.
> > >
> > > [7] Jeanselme, Vincent, et al. "Neural Fine-Gray: Monotonic neural networks for competing risks." Conference on Health, Inference, and Learning. PMLR, 2023.
> > >
> > > [8] Liu, Xin, Weijia Zhang, and Min-Ling Zhang. "HACSurv: A Hierarchical Copula-Based Approach for Survival Analysis with Dependent Competing Risks." International Conference on Artificial Intelligence and Statistics. PMLR, 2025.

---

> > > > ### Author Response · Authors · 2025-11-22
> > > > **Response to Reviewer yGg8 [7/11]**
> > > >
> > > > ### W7. Missing method families (TMLE, IV-based survival HTE, Dynamic regimes, Bayesian models)
> > > > We appreciate the reviewer pointing out these related families of methods. We have prioritized methods that operate under the standard setting of static, binary treatment assignment with selection on observables, as this remains the most common use case for causal inference in survival analysis and in applied ML for healthcare.
> > > >
> > > > *  **Dynamic Treatment Regimes & IV-based HTE:** As noted in our Discussion, we consider time-varying treatment and instrumental variable settings to be outside the scope of this benchmark, which focuses on establishing a rigorous baseline for static, heterogeneous-treatment settings.
> > > > * **TMLE:** While TMLE is a common baseline for ATE, its application to continuous CATE estimation is technically non-trivial because the CATE parameter is often (as pointed out by Reviewer-qN3T) not smooth enough for standard TMLE theory to hold without modifications (e.g., binning covariates or targeting specific subgroups). We have now clarified this distinction in the text.
> > > > * **Bayesian Survival HTE:** We agree these are relevant but prioritized the most widely used frequentist and ML-based estimators for this initial benchmark release. Given the extendability of our repository, we put Bayesian Survival HTE as a high priority for future implementation.
> > > >
> > > > However, we strongly agree that the direct-survival family should be expanded. Despite the limited time of the rebuttal period, we have now implemented and added **SurvITE** (Curth et al., 2021), as a new direct-survival CATE model, to our benchmark.
> > > >
> > > > We have also updated our *Related Work* section to acknowledge the method families mentioned by the reviewer to ensure we provide a complete picture of the field.
> > > >
> > > > >
> > > > > While our benchmark focuses on static treatments under selection on observables, related work addresses Heterogeneous Treatment Effects in alternative settings. This includes Instrumental Variable approaches for survival (Tchetgen et al., 2015), dynamic treatment regimes (Rudolph et al., 2022; Bates et al., 2022; Rudolph et al., 2023; Cho et al., 2023), and Bayesian machine learning approaches (Chen et al., 2024). Additionally, Targeted Maximum Likelihood Estimation (TMLE) (Stitelman & van der Laan, 2010; Stitelman et al., 2011) offers robust estimation for survival parameters, though primarily for subgroup effects rather than continuous CATE functions.
> > > > >

---

> > > > > ### Author Response · Authors · 2025-11-22
> > > > > **Response to Reviewer yGg8 [8/11]**
> > > > >
> > > > > ### W8. Borda count ranking: hides effect sizes; sensitive to dataset and method set; need uncertainty
> > > > > We agree that Borda counts, while useful for high-level summaries, can obscure magnitude and variability. We have addressed this in three ways:
> > > > >
> > > > > 1. **Uncertainty Quantification:** We have updated all ranking figures (Figure 1, Figure 5, Figure 6) to include standard errors.
> > > > > 2.  **Statistical Significance:** We have performed pairwise statistical testing using the Wilcoxon signed-rank test (FDR-corrected at $\alpha=0.05$) to determine if rank differences are statistically significant. (insignificant pairs are connected via bands in the figures).
> > > > > 3. **”Win-Rate” Metrics:** To capture effect sizes and dominance more transparently, we have added new sets of tables (Table 15 in Appendix F.1 across all experimental configurations, Table 16 in Appendix F.2 for win-rate performance at each survival scenario variation, and Table 17 in Appendix F.3 for win-rate performance at each causal assumption violation) reporting the percentage of times each method family appears in the Top-1, Top-3, and Top-5 positions according to both CATE RMSE and ATE Bias. This provides a clearer view of how often a method is “best” versus “average”. The accompanying text added to these sections (Appendices in F.1, F.2, and F.3) also help give an overall view of the strength-areas of each method family for different assumption violations and setup.
> > > > >
> > > > > Here we are copying the newly added Table 15, but we encourage the reviewer to also look at Table 16 and 17 and also the text added.
> > > > > | Method Family           | CATE RMSE Top-1 | CATE RMSE Top-3 | CATE RMSE Top-5 | ATE Bias Top-1 | ATE Bias Top-3 | ATE Bias Top-5 |
> > > > > |-------------------------|----------------:|-----------------:|-----------------:|---------------:|---------------:|---------------:|
> > > > > | *Outcome Imputation Methods* |                 |                  |                  |                |                |                |
> > > > > | T-Learner               |             0.0 |              0.0 |              0.0 |            0.0 |           17.5 |           25.0 |
> > > > > | S-Learner               |             0.0 |              2.5 |             12.5 |            5.0 |            7.5 |           27.5 |
> > > > > | X-Learner               |             0.0 |              0.0 |              0.0 |            2.5 |            7.5 |           17.5 |
> > > > > | DR-Learner              |             0.0 |              0.0 |              0.0 |            0.0 |           10.0 |           37.5 |
> > > > > | Double-ML               |            27.5 |             62.5 |             85.0 |            2.5 |           15.0 |           37.5 |
> > > > > | Causal Forest           |             2.5 |             40.0 |             52.5 |            2.5 |           27.5 |           40.0 |
> > > > > | *Direct-Survival Methods*    |                 |                  |                  |                |                |                |
> > > > > | Causal Survival Forest  |            35.0 |             67.5 |             82.5 |           52.5 |           75.0 |           82.5 |
> > > > > | SurvITE                 |            25.0 |             37.5 |             45.0 |           15.0 |           37.5 |           55.0 |
> > > > > | *Survival Meta-Learners*     |                 |                  |                  |                |                |                |
> > > > > | T-Learner-Survival      |             0.0 |             10.0 |             40.0 |           12.5 |           30.0 |           40.0 |
> > > > > | S-Learner-Survival      |            10.0 |             45.0 |             97.5 |            5.0 |           30.0 |           65.0 |
> > > > > | Matching-Survival       |             0.0 |             35.0 |             85.0 |            2.5 |           42.5 |           72.5 |
> > > > >
> > > > >
> > > > >
> > > > > ### W9. Limited evaluation metric (RMSE); request for quantile-based TE, fixed-time survival differences
> > > > > We thank the reviewer for this suggestion to broaden the evaluation metrics.
> > > > > We would like to ask for a clarification on the “Quantile-based treatment effect” and “rank-based” treatment effects. Should we interpret this as an interest in Quantile Treatment Effects (QTE), which measure the difference in survival times at specific percentiles (e.g., median survival difference)? In this case, although we have not presented any results for median survival time CATE (for expositional purposes), our repository already is equipped with providing “median” as a target for CATE estimation.
> > > > >
> > > > > Another possible interpretation of this, would be that we should interpret this on how the predicted CATE distribution is different from the true CATE distribution, by comparing the two at different quantiles? If this is what the reviewer is interested in, we can add this in the coming days as an appendix section for our paper.

---

> > > > > > ### Author Response · Authors · 2025-11-22
> > > > > > **Response to Reviewer yGg8 [9/11]**
> > > > > >
> > > > > > ### Q1. RMST time horizon varying across datasets; need sensitivity analysis and alternative estimands
> > > > > >
> > > > > > We thank the reviewer for bringing up this point. We have now added in Appendix G.3.3, a sensitivity analysis on varying horizons for CATE estimates based on the restricted mean survival time (RMST).
> > > > > >
> > > > > > In this sensitivity analysis, we evaluate how CATE estimation based on RMST changes when varying the prediction horizon. The main results in the paper use the RMST defined up to the maximum observed event time $T_{\max}$, but here we additionally consider a shorter horizon based on the median event time $T_{\text{med}}$ in each of the new datasets. This allows us to assess whether individual method families behave differently when estimating treatment effects over longer versus shorter time spans.
> > > > > >
> > > > > > Although results of this additional experiment asked by the reviewer are not yet fully complete, the trends indicate that the relative ordering of method families remains broadly consistent across these two horizons. Rather than comparing absolute values across datasets, the focus here is on understanding which methods are more robust to horizon length, an aspect that appears stable across the configurations examined so far.
> > > > > >
> > > > > >
> > > > > > | Method Family            | MIMIC-$vi$ ($T_\max$) | MIMIC-$vi$ ($T_\text{med}$) | MIMIC-$vii$ ($T_\max$) | MIMIC-$vii$ ($T_\text{med}$) | MIMIC-$viii$ ($T_\max$) | MIMIC-$viii$ ($T_\text{med}$) | MIMIC-$ix$ ($T_\max$) | MIMIC-$ix$ ($T_\text{med}$) |
> > > > > > |-------------------------|------------------|-------------------|-------------------|--------------------|---------------------|----------------------|-------------------|--------------------|
> > > > > > | **Direct-Survival Methods** |||||||||
> > > > > > | Causal Survival Forest  | 7.123 ± 0.048    | 3.850 ± 0.032     | 7.281 ± 0.064     | 3.740 ± 0.025      | 7.149 ± 0.045       | 3.839 ± 0.031        | 7.227 ± 0.054     | 3.725 ± 0.032      |
> > > > > > | SurvITE                 | 7.243 ± 0.154    | 3.908 ± 0.067     | 7.378 ± 0.112     | 3.869 ± 0.177      | 7.268 ± 0.117       | 3.886 ± 0.057        | 7.347 ± 0.078     | 3.813 ± 0.055      |
> > > > > > | **Survival Meta-Learners** |||||||||
> > > > > > | T-Learner Survival      | 7.465 ± 0.364    | 4.314 ± 0.563     | 7.487 ± 0.179     | 3.967 ± 0.241      | 7.266 ± 0.053       | 3.904 ± 0.081        | 7.465 ± 0.254     | 4.093 ± 0.379      |
> > > > > > | S-Learner Survival      | 7.183 ± 0.051    | 3.866 ± 0.036     | 7.345 ± 0.066     | 3.760 ± 0.024      | 7.198 ± 0.042       | 3.852 ± 0.030        | 7.283 ± 0.049     | 3.742 ± 0.032      |
> > > > > > | Matching Survival       | 7.219 ± 0.060    | 5.192 ± 0.203               | 7.393 ± 0.074     | 5.358 ± 0.323                | 7.240 ± 0.043       | 5.207 ± 0.189                  | 7.357 ± 0.046     | 5.420 ± 0.205                |

---

> > > > > > > ### Author Response · Authors · 2025-11-22
> > > > > > > **Response to Reviewer yGg8 [10/11]**
> > > > > > >
> > > > > > > Furthermore, to address the reviewer’s suggestion regarding alternative estimands, we have incorporated an additional CATE metric based on the difference in survival probabilities at fixed time horizons. Let $S_i(w; h) := \Pr(T_i(w) > h)$ denote the potential survival function for unit $i$ under treatment $w \in \{0,1\}$ at a specific horizon $h$. By defining the outcome transformation as $y(T_i(w)) := S_i(w; h)$, the CATE estimand becomes the conditional difference in survival probability:
> > > > > > >
> > > > > > > $$
> > > > > > > \tau_h(x) := \mathbb{E}[ S_i(1; h) - S_i(0; h) \mid X_i=x ]
> > > > > > > $$
> > > > > > >
> > > > > > > We evaluated this quantity at three distinct horizons—the **25th, 50th, and 75th percentiles** of the empirical event-time distribution—to assess estimator performance across early, intermediate, and late stages of the survival trajectory.
> > > > > > >
> > > > > > > The full setup is provided in Appendix G.3.2 and the results on the new semi-synthetic datasets are reported in Tables 32, 33, and 34 of the revised draft. We note that this estimand is computed only for direct-survival models and survival meta-learners, as outcome-imputation methods generally provide point estimates rather than full survival curves. As shown in the summary table below, *Causal Survival Forest* remains the strongest performer across all horizons and datasets, but *SurvITE* seem to have degraded performance. Among the survival meta-learners, the *S-Learner Survival* exhibits the most stability, while *Matching Survival* generally show higher RMSE. Crucially, the relative ranking of methods remains consistent across the 25th, 50th, and 75th percentiles, suggesting that method performance is robust to the choice of time horizon $h$ in these datasets.
> > > > > > >
> > > > > > > **Table: CATE RMSE based on Survival Probability (Summary of Appendix Tables 32-34)**
> > > > > > >
> > > > > > > | Method Family            | MIMIC-$vi$ (25%) | MIMIC-$vi$ (50%) | MIMIC-$vi$ (75%) | MIMIC-$vii$ (25%) | MIMIC-$vii$ (50%) | MIMIC-$vii$ (75%) | MIMIC-$viii$ (25%) | MIMIC-$viii$ (50%) | MIMIC-$viii$ (75%) | MIMIC-$ix$ (25%) | MIMIC-$ix$ (50%) | MIMIC-$ix$ (75%) |
> > > > > > > |--------------------------|----------------|----------------|----------------|-----------------|-----------------|-----------------|------------------|------------------|------------------|----------------|----------------|----------------|
> > > > > > > | **Direct-Survival Methods** |                |                |                |                 |                 |                 |                  |                  |                  |                |                |                |
> > > > > > > | Causal Survival Forest   | 0.044          | 0.052          | 0.053          | 0.035           | 0.044           | 0.050           | 0.038            | 0.054            | 0.047            | 0.041          | 0.054          | 0.056          |
> > > > > > > | SurvITE                  | 0.108          | 0.125          | 0.094          | 0.099           | 0.109           | 0.084           | 0.107            | 0.116            | 0.099            | 0.099          | 0.116          | 0.096          |
> > > > > > > | **Survival Meta-Learners** |                |                |                |                 |                 |                 |                  |                  |                  |                |                |                |
> > > > > > > | T-Learner Survival       | 0.085          | 0.104          | 0.101          | 0.068           | 0.098           | 0.091           | 0.085            | 0.106            | 0.094            | 0.069          | 0.091          | 0.088          |
> > > > > > > | S-Learner Survival       | 0.064          | 0.086          | 0.074          | 0.065           | 0.085           | 0.078           | 0.069            | 0.090            | 0.073            | 0.067          | 0.085          | 0.082          |
> > > > > > > | Matching Survival        | 0.076          | 0.096          | 0.089          | 0.079           | 0.101           | 0.094           | 0.083            | 0.105            | 0.087            | 0.091          | 0.115          | 0.106          |

---

> > > > > > > > ### Author Response · Authors · 2025-11-22
> > > > > > > > **Response to Reviewer yGg8 [11/11]**
> > > > > > > >
> > > > > > > > ### Q2. Why only CSF included in direct-survival CATE models? Missing SurvITE and others
> > > > > > > > We appreciate this feedback. To address this, we have implemented *SurvITE* (Curth et al., 2021) and integrated it into the full benchmarking pipeline.
> > > > > > > > * **Results:** *SurvITE* results are now included in all aggregated figures and tables. As a brief summary, in our synthetic experiments, *SurvITE* attains a mid-range overall Borda rank but shows solid win-rates (often reaching Top-1 CATE RMSE and competitive ATE bias) particularly in settings with medium or high censoring and in randomized treatment configurations. This competitive performance of SurvITE is consistent with the solid performance of Causal Survival Forest that is in the same direct-survival method family.
> > > > > > > > * **Documentation:** The method description has been added to Appendix D.2, and hyperparameters are detailed in Appendix E.2.
> > > > > > > > * **Code:** The anonymized repository has also been updated to include our SurvITE implementation.
> > > > > > > >
> > > > > > > > ---
> > > > > > > >
> > > > > > > > We thank the reviewer again for the thoughtful and constructive feedback, which has helped us strengthen both the benchmark and the manuscript. We hope that the updates and clarifications provided above satisfactorily address all of the reviewer’s concerns and allow the reviewer to reconsider their score.

---

### Author Response · Authors · 2025-12-04
**Summary of Major Revisions: Addressing Conditions for Score Increases**

We thank the reviewers for their constructive feedback. Because the discussion phase has been cancelled, reviewers cannot update their scores to reflect our revisions. Also, unfortunately, none of them had responded to our rebuttals by the time the discussion phase was cancelled. **Crucially, multiple reviewers explicitly stated in their initial reviews that they would be willing to raise their scores if specific concerns were addressed.** This summary highlights how the revised paper meets those conditions and how we have improved our paper during the rebuttal phase.

---

1. **Addressed Reviewer Yomt’s Condition: Increased Data Complexity (Score 6 $\rightarrow$ potentially higher):** Reviewer Yomt stated they would "happily raise [their] score" if we addressed the limitation of linear ground-truth CATEs.
* **Action:** We expanded the benchmark by generating four new semi-synthetic datasets (**MIMIC-vi through MIMIC-ix**) that systematically introduce **non-linear dependencies and interaction terms** in both treatment assignment and outcomes.
* **Result:** These additions explicitly stress-test estimators against complex ground truths, fulfilling the reviewer's request for higher complexity. We also added Kaplan-Meier plots to justify parameter choices.

2. **Addressed Reviewer m4FZ’s Condition: Actionable Insights (Score 4 $\rightarrow$ potentially higher):** Reviewer m4FZ stated they were "willing to increase [their] grade" if we addressed the "when does each method work best" issue.
* **Action:** We moved beyond global averages by adding Win-Rate Tables (Top-1/3/5 frequency) and scenario-specific rankings in Appendix F.
* **Result:** The paper now provides clear, granular guidelines on which method families dominate under specific conditions (e.g., high censoring vs. confounding), directly answering "which method works best when."

3. **Addressed Consensus Request: Method Coverage (SurvITE):** Reviewers *FuYz* and *yGg8* noted the absence of deep direct-survival models.
* **Action:** We implemented and integrated **SurvITE** (Curth., et al. 2021) into the full benchmarking pipeline, updating all aggregate figures and tables (Section 4 and Appendices).

4. **Addressed Consensus Request: Estimands & Time Horizons:** Reviewers *yGg8* and *Yomt* requested analysis of horizon sensitivity and alternative estimands.
* **Action:** We added **Survival-Probability-based CATE** estimands evaluated at the 25th, 50th, and 75th percentiles of times.  We also included a sensitivity analysis for our restricted-mean-survival-time–based CATE at the median horizon (Appendix G.3), complementing our original results at the maximum follow-up horizon.
* **Result:** Observing the same trends as in our pre-rebuttal submission across these additional analyses confirms that our method-family based rankings are robust to the choice of time horizon and CATE estimand.

5. **Addressed Robustness, Usability & Positioning (Reviewers qN3T, yGg8, m4FZ):** Beyond the conditional requirements above, we resolved all other specific critiques to ensure the benchmark is rigorous and extensible:
* **Extensibility (Reviewer qN3T):** We refactored the codebase to include base classes and a step-by-step tutorial for adding new learners, ensuring the benchmark serves as a lasting community resource (can be viewed in the updated anonymized repo).
* **Uncertainty Quantification (Reviewer yGg8):** We added statistical significance bands to all Borda rankings as well as standard errors to each ranking figure to address concerns about rank stability.
* **Positioning (Reviewer m4FZ):** We revised the related work to explicitly situate our contribution against recent causal inference and survival ATE benchmarks.

---

**Summary:** We have executed the extensive revisions requested by the reviewers, specifically satisfying the explicit conditions set by **Yomt** and **m4FZ** for score increases, while comprehensively resolving the methodological and extensibility concerns of **yGg8**, **FuYz**, and **qN3T**. We hope that, with these extensive additions (new estimands, additional estimator, expanded ranking and win-rate analyses, and clarified data-generating processes) and the fulfillment of specific reviewer conditions, the AC finds the paper suitable for acceptance.

---

### Meta-Review · Area_Chair_4zBk · 2026-01-07

**Summary:**

Across reviews, there was broad agreement that the paper’s core contribution, a unified, reproducible benchmark for heterogeneous treatment effects (HTE/CATE) with right-censored survival outcomes, fills a real gap and is valuable to the community. Reviewers highlighted (i) breadth of evaluated estimators, (ii) coverage across synthetic, semi-synthetic, and real data, and (iii) strong reproducibility and modularity including a large unified implementation suite.

**Reviewer Concerns:**

* Benchmark realism and data-generating complexity. Multiple reviewers worried the synthetic and semi-synthetic generators were too simple (e.g., limited nonlinearity/interaction structure; simplistic treatment/censoring mechanisms; binary “violation” settings rather than graded severities). The authors properly address it by integrating configurations and methods and providing extended analysis in the revision.

* Coverage gaps in method families and estimands. Reviewers requested inclusion of deep direct-survival CATE models and more explicit analysis of time-horizon or estimand dependence (e.g., fixed-time survival probability differences, sensitivity to RMST horizon). The paper’s new win-rate framing directly addresses this by quantifying top frequencies and by scenario-level breakdowns.

* Actionability and presentation of results. A recurring critique was that global rankings (Borda averages) can be hard to interpret and may hide uncertainty or effect sizes; reviewers wanted clearer discussion on when each model works best and stronger uncertainty quantification. The arguments have been provided by the author in the rebuttal.

* Scope and positioning. Some reviewers questioned novelty relative to prior HTE benchmarks and requested clearer positioning, plus clarifications on assumptions and design choices. The authors spend effort to provide clearer positioning on the paper and frame the novelty as the comprehensive benchmark for right-censored survival HTE.

Overall speaking, while a few improvements are still possible for the benchmark and the paper, including extended methods and synthetic process, more insightful and comprehensive methodological comparisons and a few more arguments, the paper provides a concrete benchmark on HTE with survival datasets, which is, as recognized by the reviewers, surely needed in the community. The major limitation is still the scope of the paper. However, as the authors have also contributed much effort to integrate methods and configurations in the revision, making it a suitable choice to evaluate methods on survival HTE estimation, some remaining limitations can be appropriately treated as future work. Therefore, while the benchmark has the potential to be more extensive and impactful, a move above threshold for the paper is plausible.

**Reviewer Scores:**

Below I list each reviewer’s original overall rating and how I believe the major concerns would have adjusted after seeing the described revisions. I also hypothetically justify each change in scores in terms of their stated score-increase conditions.

Reviewer yGg8: 4 → 4

In the rebuttal, the authors address the concerns raised by Reviewer yGg8 including: added uncertainty quantification and win-rate reporting, broader estimands/horizon sensitivity, and inclusion of SurvITE, while major concerns such as graded violations, broader censoring types beyond right-censoring and additional methods in survival HTE remain further assessment.

Reviewer Yomt: 6 → 8

They explicitly said they’d “happily raise” if nonlinearities or interactions and coefficient justification or robustness were addressed. The revisions directly target this by adding configurations with non-linear dependencies and interaction terms, as well as justification on the parameter choice, so a modest bump to solid accept seems plausible.

Reviewer qN3T: 6 → 6 or 8

Their main weakness was the adoptability and extensibility in documentation, plus a technical TMLE nuance and questions about conditioning on sub-vectors or other coarsened data. The rebuttal describes refactoring and clearer extension interfaces, which may convert into a more confident accept.

Reviewer FuYz: 4 → 6

Their major issues lie in two aspects. (a) More extensive synthetic data generation process and clearer articulation. (b) Insufficient actionable insights. (c) The missingness of some models or methods. In the revisions, the authors made much effort to address the problem including SurvITE integration, adding new win-rate metrics and providing plausible explanation on the problems raised by the reviewer, so a meaningful upward revision is reasonable.

Reviewer m4FZ: 4 → 6

They explicitly offered a score increase if the paper answered when each method works best and improved presentation and positioning. The new win-rate tables and scenario-specific summaries provided in the rebuttal answer this question to a proper extent. The other questions have also been carefully responded by the reviewer.

---

### Decision · Program_Chairs · 2026-01-26

Accept (Poster)